# Neuro-inspired optical sensor array for high-accuracy static image recognition and dynamic trace extraction

Pei-Yu Huang[1,6], Bi-Yi Jiang[2,3,6], Hong-Ji Chen[1,6], Jia-Yi Xu[2], Kang Wang[4], Cheng-Yi Zhu[1], Xin-Yan Hu[2], Dong Li[1], Liang Zhen[5], Fei-Chi Zhou[2]✉, Jing-Kai Qin[1]✉ & Cheng-Yan Xu[1,5]✉

Neuro-inspired vision systems hold great promise to address the growing demands of mass data processing for edge computing, a distributed framework that brings computation and data storage closer to the sources of data. In addition to the capability of static image sensing and processing, the hardware implementation of a neuro-inspired vision system also requires the fulfilment of detecting and recognizing moving targets. Here, we demonstrated a neuro-inspired optical sensor based on two-dimensional $NbS_2/MoS_2$ hybrid films, which featured remarkable photo-induced conductance plasticity and low electrical energy consumption. A neuro-inspired optical sensor array with $10 \times 10$ $NbS_2/MoS_2$ phototransistors enabled highly integrated functions of sensing, memory, and contrast enhancement capabilities for static images, which benefits convolutional neural network (CNN) with a high image recognition accuracy. More importantly, in-sensor trajectory registration of moving light spots was experimentally implemented such that the post-processing could yield a high restoration accuracy. Our neuro-inspired optical sensor array could provide a fascinating platform for the implementation of high-performance artificial vision systems.

Advanced artificial vision systems are essential for the development of applications such as intelligent homes, autopilot vehicles, and video content analysis[1–3]. The common digital vision system is developed based on the complementary metal oxide semiconductor (CMOS) platform, which consists of separated image sensors for receiving visual inputs, memories for storing information, and processors for performing computation. The sequential processing paradigm of von Neumann computers faces great challenges of energy consumption, redundancy, and latency for processing the explosive growth of visual data[4–6]. By comparison, the human vision system is capable of processing visual data with high efficiency and accuracy. The retina can not only obtain high-quality visual images but also conduct low-level sensory processing such as noise suppression[7–9], image sharpening[10,11] and motion extraction[12,13], etc., thus, allowing decreased redundant visual data and improved efficiency for information processing in the human brain[14–16]. Although retina-inspired vision chips based on silicon CMOS technology have been proposed[17–21], the applications still suffer from the shortcomings of circuit complexity, distortion of analog-to-digital conversion and poor compatibility with back-end processing platforms[22–24]. Therefore, the hardware implementation of neuro-inspired image sensors, which are highly integrated with visual

[1]Sauvage Laboratory for Smart Materials, School of Materials Science and Engineering, Harbin Institute of Technology (Shenzhen), Shenzhen 518055, China. [2]School of Microelectronics, Southern University of Science and Technology, Shenzhen 518055, China. [3]Department of Applied Physics, The Hong Kong Polytechnic University, Hong Kong 999077, China. [4]Key Laboratory of MEMS of the Ministry of Education, Southeast University, Nanjing 210096, China. [5]MOE Key Laboratory of Micro-Systems and Micro-Structures Manufacturing, Harbin Institute of Technology, Harbin 150080, China. [6]These authors contributed equally: Pei-Yu Huang, Bi-Yi Jiang, and Hong-Ji Chen. ✉e-mail: zhoufc@sustech.edu.cn; jk_qin@hit.edu.cn; cy_xu@hit.edu.cn

information sensing, memory, and pre-processing functions, is urgently needed to advance artificial vision systems.

Metal oxide films[25–27], two-dimensional (2D) van der Waals (vdW) materials and related heterostructures[28–35] have attracted extensive attention due to their applications in emerging neuro-inspired optical sensors. In particular, atomically thin 2D vdW heterostructures show strong light-matter interactions, a wide spectral range of optical absorption, and a gate-tunable band structure[36–43]. They are considered ideal material candidates to develop neuro-inspired vision systems with the capabilities of visual signal sensing and image processing, which is expected to offer a highly simplified hardware system compared to the conventional retinomorphic chip based on the CMOS platform. For example, Kim et al. reported a curved neuromorphic image sensor based on $MoS_2$-poly(1,3,5-trimethyl-1,3,5-trivinyl cyclotrisiloxane) (pV3D3) heterostructures[28]. By combining with a planoconvex lens, the sensor could acquire the pre-processed image from a set of noisy optical inputs. Park et al. reported an optic-neural synaptic device based on 2D $h$-BN/$WSe_2$ heterostructures. The intriguing chrominance-dependent synaptic dynamics enabled the design of artificial vision systems for colored-mixed pattern recognition[44]. However, previously reported neuro-inspired image sensors based on emerging 2D materials are largely limited to static information detection and processing. The absence of computation for moving targets hinders the development of a fully hardware-implemented machine vision system, which monolithically integrates the functions of pre-processing both static and moving targets as biological retinas.

In this work, we report a neuro-inspired phototransistor based on 2D $NbS_2$/$MoS_2$ hybrid films, which are synthesized using a two-step salt-assisted chemical vapor deposition (CVD) approach. Due to the efficient gate-tunable charge transfer at the $NbS_2$/$MoS_2$ heterointerface, the device demonstrates pronounced persistent photoconductivity (PPC), with diversified photo-modulation conductance

plasticity, including pulse-interval dependent plasticity, pulse-number dependent plasticity, multistate optical accumulation and electrical inhibition. A neuro-inspired optical sensor array with $10 \times 10$ $NbS_2$/$MoS_2$ devices enables multiple functions including sensing, memory, and contrast enhancement for static images, which benefits convolutional neural network (CNN) with a high image recognition accuracy in artificial vision systems. More importantly, the trajectory registration of dynamic light spots was also experimentally demonstrated, providing a high restoration accuracy for complicated paths. The development of neuro-inspired optical sensors could facilitate the increasing opportunities for constructing compact and efficient artificial vision systems.

## Results

### Design and fabrication of the neuro-inspired optical sensor array

The human visual system mainly consists of the retina, optic nerve, lateral geniculate nucleus (LGN) and visual cortices. Light signals are initially converted by photoreceptor cells and then pre-processed with the assistance of nerve cells, including bipolar cells, horizontal cells, amacrine cells, and ganglion cells, thus enabling the filtration of redundant and unstructured visual data[11,45,46]. Subsequently, the LGN receives and transmits information from the retina to the cortex for the cognition of visual stimuli[47–51]. The functions in terms of noise filtering, edge enhancement, and motion detection of the retina provide the cortex with high efficiency for high-level processing of massive visual data.

As shown in Fig. 1a, inspired by the pre-processing functions of the retina, we designed an optical sensor array with 100 pixels using 2D $NbS_2$/$MoS_2$ hybrid films as the sensing materials. Due to the efficient gate-tunable charge transfer at the heterointerface, the $MoS_2$/$NbS_2$ optical sensor exhibits light tunable conductance characteristics, thus

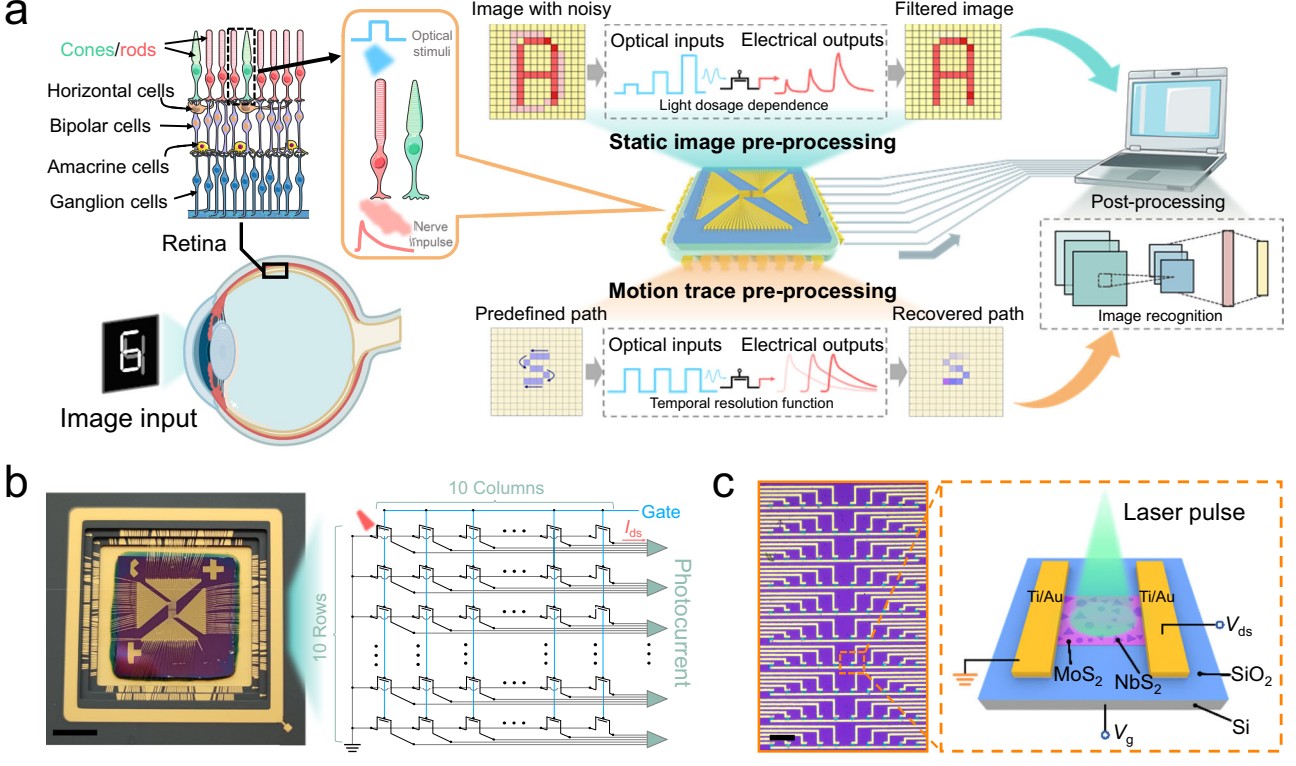

**Fig. 1 | Neuro-inspired optical sensor array based on 2D $NbS_2$/$MoS_2$ hybrid films for processing both static and moving targets. a** Schematic illustration of the neuro-inspired vision system for processing both static and moving targets. The figure was partly generated using Servier Medical Art, provided by Servier, licensed under a Creative Commons Attribution 3.0 unported license. **b** Digital image of neuro-inspired optical sensor array and corresponding circuit diagram. Scale bar in (**b**): 5 mm. **c** Low-magnification optical photograph of $10 \times 10$ device array and schematic of $NbS_2$/$MoS_2$ phototransistor. Scale bar in (**c**): 100 μm.

enabling it to perform several low-level sensory processing steps. Based on the light-dosage-dependence of photogeneration and temporally-resolved characteristic for history inputs, the neuro-inspired optical sensor can not only operate with integrated functions of sensing, memory, and contrast enhancement capabilities for static images but also enable the trajectory registration of moving light spots. Figure 1b shows a digital photographic image of the sensor array packaged in a ceramic pin grid array (PGA) carrier and corresponding equivalent circuit, in which each individual unit is connected to the corresponding voltage source as well as the current-sense amplifier for measurement. The optical sensor is fabricated in the field-effect phototransistor configuration using $SiO_2$ as the bottom-gate dielectric. All devices are patterned into rectangular shapes with a channel size of $10 \times 20$ μm. A 532 nm laser with a spot size of 10 μm² is illuminated on the channel of $NbS_2/MoS_2$ hybrid films, and the electrical properties of 100 device pixels in the array are automatically measured using a home-built testing system (Fig. 1c and Supplementary Fig. 1).

The $NbS_2/MoS_2$ phototransistor is the core cell of a neuro-inspired optical sensor. The left panel in Fig. 2a shows the optical microscopy (OM) image of an individual device. 2D hybrid films are synthesized with triangular $NbS_2$ atomic crystals randomly distributed on continuous monolayer $MoS_2$ films (Fig. 2a, middle panel). High-resolution transmission electron microscopy (HRTEM) was carried out to characterize the microstructures of $NbS_2/MoS_2$ hybrid films (right panel in Fig. 2a and Supplementary Fig. 2). From the corresponding selected-area electron diffraction (SAED) patterns, we could clearly observe two sets of well-nested hexagonal spots with the same orientation, confirming the epitaxial alignment of $NbS_2$ (200) planes with $MoS_2$ (200) planes. The results of X-ray photoelectron spectroscopy (XPS), Raman spectroscopy, and photoluminescence (PL) spectroscopy are shown in Supplementary Figs. 3–5. Notably, significant PL quenching of the $MoS_2$ monolayer occurred in the heterostructure, and similar phenomena were reported in $VS_2/MoS_2$ metal/semiconducting hybrid films[52] and graphene/$WS_2$ quasi-metal/semiconducting systems[53,54], which was indicative of the intense charge transfer of photoexcited carriers at the heterojunction.

## Light-tunable conductance characteristics of $MoS_2/NbS_2$ phototransistor

Figure 2b and Supplementary Fig. 6 exhibit the transfer ($I_{ds}$-$V_{gs}$) and output characteristics ($I_{ds}$-$V_{ds}$) of the $NbS_2/MoS_2$ phototransistor under a 532 nm laser with different incident powers. The device features typical n-type transport behavior and demonstrates a high current ratio of $10^4$ for dark and illumination conditions. The dependence of photoresponsivity ($R$) and detectivity ($D^*$) on laser power are shown in Supplementary Figs. 6–8. A maximum $R$ of 1.1 A/W and a maximum $D^*$ of $2 \times 10^{11}$ Jones are obtained at 2.4 nW μm$^{-2}$, which are ~6 and 10 times larger than those of the $MoS_2$ counterpart, respectively. The enhanced photoresponse can be effectively explained by the reduction of the contact resistance and the lowering of the barrier height with the integration of $NbS_2$, which facilitates the separation of the photon-generated charges and inhibits the recombination of the carriers[55–57].

Notably, the device shows a pronounced PPC effect at a negative gate voltage (Fig. 2c). The photocurrent rapidly increases when the light is turned on, showing a fast photo-triggered response time of 304 ms. After the light is turned off, the photocurrent slowly decays to a new equilibrium state instead of completely disappearing. The PPC state is effectively maintained for >1000 s and erased within 205 ms by applying a positive gate voltage (Supplementary Figs. 9 and 10). The

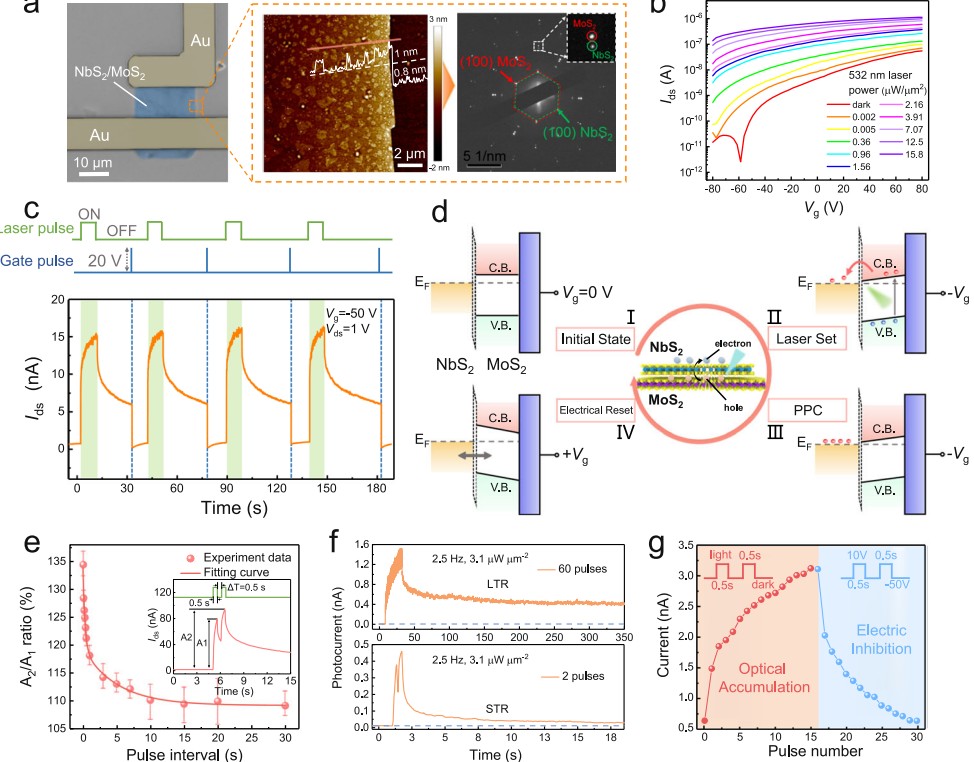

**Fig. 2 | Optoelectronic performance of $NbS_2/MoS_2$ phototransistor. a** OM image of a fabricated device. Right panels: AFM height image and SAED patterns of $NbS_2$/$MoS_2$ hybrid films. **b** Transfer curves ($I_{ds}$-$V_{gs}$) of device at $V_{ds} = 1$ V with different incident light powers. **c** Photoresponse of device to four light ON/OFF cycles at $V_{gs} = -50$ V. The green shadings indicate the duration of light pulse. **d** Schematic illustrations of the origins of the PPC and energy band diagram at different bias gate. **e** Modulation ratio of conductance as a function of interval time ($\Delta t$). The inset plots the time-resolved photocurrent of device at two successive light pulses with width of 500 ms. The error bars represent the standard deviation of 10 measurements for each pulse interval. **f** Short-term relaxation and long-term relaxation processes under 2 and 60 optical pulses, respectively. **g** Condensed plots of optical accumulation and electrical inhibition processes over 20 cycles.

working mechanism of electro-optical coordinated operation in the device is illustrated based on energy band diagram. As shown in Fig. 2d, in the initial stage (Stage I), no charge exchange occurs between $MoS_2$ and $NbS_2$ without applying a $V_g$. A large negative voltage coupled to the device gate leads to band bending at the $NbS_2$/$MoS_2$ interface. This facilitates the spatial separation of the photo-excited electron-hole pairs in $MoS_2$ and results in current enhancement. Notably, the potential barrier at the heterointerface can efficiently inhibit the electron-hole recombination in $MoS_2$, contributing to a nonvolatility of the photocurrent after light removal (Stages II and III). In the reset stage (stage IV), the PPC can be readily erased by applying a large positive $V_g$ spike, which enables an instantaneous increase in the Fermi energy and redistribution of charge carriers in the $NbS_2$/$MoS_2$ heterojunction[30,58].

The tunable memory characteristics of the $NbS_2$/$MoS_2$ phototransistor cause it to exhibit diversified photo-modulation conductance plasticity. The pulse-interval dependent plasticity was demonstrated by applying two successive light pulses with a duration of 500 ms and different interval durations at a $V_g$ value of −50 V. The amplitude of the photocurrent ($A_2$) triggered by the second light pulse was significantly higher than that ($A_1$) triggered by the first pulse. This result could be attributed to the superposition of subsequently excited carriers with residual carriers before the following pulse in a short pulse interval[59]. The modulation ratio is defined as $A_2/A_1$, and the relationship between the ratio and pulse interval time ($\Delta t$) is plotted in Fig. 2e. With the increase in $\Delta t$ from 0.02 to 30 s, the modulation ratio gradually declines from 135% to 112%. This ratio can be effectively fit by a double-exponential function:

$$\text{Modulation ratio} = \alpha_1 e^{-\Delta t/\tau_1} + \alpha_2 e^{-\Delta t/\tau_2} + C \qquad (1)$$

where $\alpha_1$, $\alpha_2$ and $C$ are the facilitation constants, and $\Delta t$, $\tau_1$, and $\tau_2$ represent the interval time in two pulses, rapid relaxation time and slow relaxation time, respectively. The two characteristic timescales, $\tau_1$ and $\tau_2$, are estimated to be 3.78 and 0.18 s, respectively (Supplementary Fig. 11). $\tau_1$ is an order of magnitude $>\tau_2$, indicating a successful stimuli process, consistent with previously reported neuro-inspired optically stimulated devices (Supplementary Table 1)[60–64]. The dynamic tunable processes of conductance plasticity were also demonstrated by changing the light intensity, duration and frequency (Supplementary Fig. 12).

The evolution of short-term relaxation (STR) and long-term relaxation (LTR) is accomplished by applying different optical pulses to the phototransistor. As shown in Fig. 2f, after applying two successive pulses with a duration of 300 ms, the photocurrent rapidly declines to the initial stage within 20 s, which conceptually refers to the STR. LTR is achieved by applying 60 successive pulses, and 30% of the peak photocurrent can be maintained even after a long decay time of 350 s. Figure 2g shows the characteristics of the optical accumulation and electrical inhibition for the channel conductance. The linear conductance change between them is evaluated by introducing the asymmetric ratio (AR)[65], which is defined by the following equation[66,67]:

$$AR = \frac{\max|G_p(n) - G_d(n)|}{G_p(15) - G_d(15)}, (n = 1 \text{ to } 15) \qquad (2)$$

where $G_p(n)$ and $G_d(n)$ are the channel conductance values after applying the nth light pulse and nth electrical spike, respectively. The mean value and standard deviation of AR for the $MoS_2$/$NbS_2$ sensor array are calculated to be 0.65 and 0.03, respectively (Supplementary Fig. 13), which are considered as being outstanding in previously reported optical neuro-inspired devices based on 2D vdW heterostructures[68–70]. The energy consumption of the electrical response for the $MoS_2$/$NbS_2$ phototransistor with different pulse durations is summarized and plotted in Supplementary Figs. 14 and 15,

and a minimum value is identified as 0.42 pJ with a pulse duration of 0.01 s. This value is comparable to those of previously reported state-of-art optical neuro-inspired devices[71–76], indicating the great potential of $NbS_2$/$MoS_2$ phototransistors for low-power optoelectronic applications (Supplementary Table 2).

## Image acquisition and pre-processing for the static images

A neuro-inspired optical sensor array was constructed based on a $10 \times 10$ matrix $NbS_2$/$MoS_2$ phototransistor pixel. The device-to-device variation was first statistically evaluated (Supplementary Fig. 16). The 100% device yield and excellent uniformity of performance metrics are crucial for the acquisition of high-quality images. Supplementary Fig. 17 demonstrates negligible crosstalk between adjacent pixels in the sensor array. To evaluate the performance of the neuro-inspired optical sensor array, an "H" shaped photomask with 18 pixels was placed on top of the array for measurement, and the photocurrent of each pixel was collected after different decay durations. As shown in Fig. 3a, image "H" was effectively recognized after repeatedly inputting optical stimuli 60 times (532 nm, 3.1 μW μm$^{-2}$, 2.5 Hz). With an increase in decay time, the measured photocurrent gradually decreased. Notably, the conductance contrast was still sufficient to identify the "H" pattern in the weight map even with a long decay duration of 300 s, indicating the achievement of steady sensing and memory of visual information based on the neuro-inspired sensor array.

The effective modulation of the conductance relaxation speed is also determined in the $NbS_2$/$MoS_2$ optical sensor by tuning the intensity of light stimulation, thus enabling the experimental implementation of image pre-processing for contrast enhancement between the target image and background. Figure 3b shows the decay characteristics of the device as trained by 100 successive light pulses with light intensities of 3.1 and 1.6 μW μm$^{-2}$. The decay rate of the photocurrent at 3.1 μW μm$^{-2}$ is much slower than that at 1.6 μW μm$^{-2}$, and a considerably increased ratio of remaining photocurrent is estimated to be 4.5 after decaying for 150 s. Similar increases in the output photocurrent ratio was also observed for input light intensity ratios of 1/0.8 and 1/0.6 (Supplementary Fig. 18). The light-tunable conductance relaxation speed enables the contrast enhancement of the input image, enabling the target letter with highlighted key features in a messy background. Here, the contrast is defined as the ratio of the collected average photocurrent between illuminated target pixels and background pixels. Correspondingly, Fig. 3c illustrates the greyscale of the pixels for the input images with a target letter "H" stimulated at 3.1 μW μm$^{-2}$ and a background letter "T" at 1.6 μW μm$^{-2}$. Clearly, the pre-processed image after repeated training demonstrates an enlarged difference between the greyscale of the pixels over the input images, thus contributing to an output image with enhanced contrast and highlighted features. These results indicate that the contrast enhancement between the target image and background can be hardware implemented based on our $NbS_2$/$MoS_2$ sensor array without using any external computing circuits.

To evaluate the recognition accuracy of the images before and after contrast enhancement, an artificial vision system including a numerical model simulating $28 \times 28$-pixel $NbS_2$/$MoS_2$ sensor array and a CNN were constructed, as shown in Fig. 3d. A 24000-sized image training set and a 4000-sized image test set with 10 categories of noisy handwritten letter images ('A' to 'J') from the Extended Modified National Institute of Standards and Technology (EMNIST) dataset were built[77]. Each image with $28 \times 28$ pixels consists of a bright informative letter, a dark background letter, and background white noise, which were pre-processed by the sensor array model and then inputted into the CNN for further high-level recognition (Supplementary Figs. 19–23). Figure 3e and Supplementary Figs. 24–26 show the performance of the artificial vision system for different combinations of input light intensities and relaxation time. The results of the output image dataset indicated that the pre-processing of the sensor array

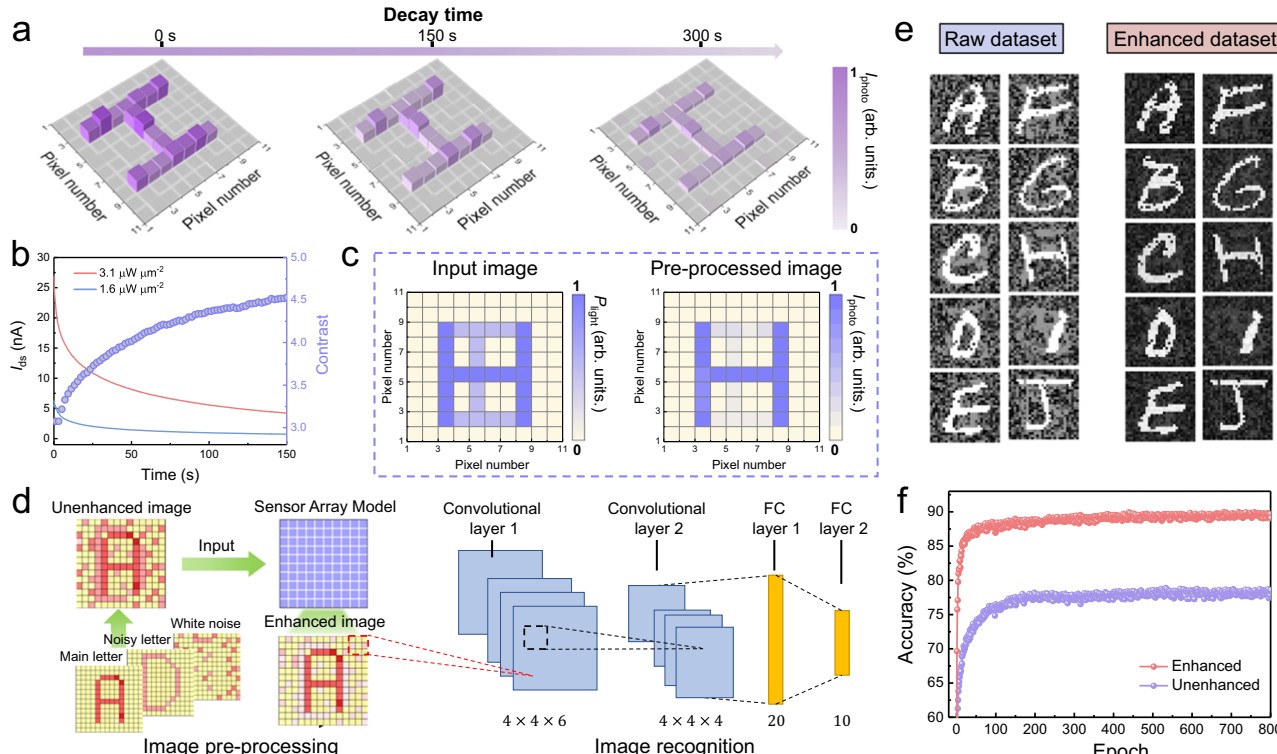

**Fig. 3 | Sensing, memory, and contrast enhancement functions of neuro-inspired optical sensor array. a** Measured training weight map of the 10 × 10 sensor array for input letter "H" with 60 successive light pulses. **b** Decay behavior of photocurrent triggered by different light intensity. **c** Pre-processed images with highlighted target letter "H". **d** Schematic diagram of constructed artificial vision system. **e** Comparison of image before and after pre-processing. **f** Recognition accuracies of the vision system with and without NbS$_2$/MoS$_2$ sensor array model for image pre-processing.

evidently reduced the noise, improved the image quality and contributed to the enhanced contrast between the target letter and background with sharpened details. For comparison, an artificial vision system without the pre-processing function was also trained and tested with the same dataset and procedure. The recognition accuracies of the two vision systems are shown in Fig. 3f. A remarkable improvement in the recognition rate was achieved from 78% to 90% after the integration of pre-processing module. Thus, with the integration of the NbS$_2$/MoS$_2$ neuro-inspired sensor array for image pre-processing, the background noise signals of input visual information could be effectively smoothed for high-accuracy static image recognition.

## Trajectory detection and recognition for motion objects

In addition to static image sensing and pre-processing, the designed NbS$_2$/MoS$_2$ neuro-inspired optical sensor array was also helpful for detecting and recognizing the trajectory of moving light spots. Front-end motion detection was generally achieved based on a CMOS vision chip, which captured visual information framed by frame at a predetermined rate (Supplementary Fig. 27). The separated memory and processing of frame data at the sensor-level or pixel level could cause unnecessarily inflating data[78,79]. A dynamic vision sensor (DVS) was also developed based on the CMOS platform, which could asynchronously output in the form of an event stream with an address-event representation (AER)[18–21,24,80]. Although DVS could be applied for real-time and high temporal resolution motion detection, the obtained raw data or event stream still needed to be successively transmitted to an external memory unit for post-processing, thus posing challenges for the hardware simplification of the vision system (Supplementary Table 3). In contrast, according to the difference in electrical output in the spatiotemporal dimension, the NbS$_2$/MoS$_2$ sensor array could derive the pre-processed information regarding the relationship between the light spot motion position and timeline (Fig. 4a). The kinetics of the time-dependent decay photocurrent enabled direct processing and generation of the spatiotemporal information within the sensor, which could be parallelly real-time memorized in each pixel. This would remarkably advance the efficiency of data processing and contribute to system simplification.

Figure 4b presents the schematic diagram of the sensor array for the implementation of trajectory registration, in which a trajectory moving with a uniform speed was predefined from pixel 0 to pixel 9. To dynamically track the trajectory change, the remaining photocurrent of each pixel along the trajectory was collected in sequence 10 times and map-plotted in **panel (i) of** Fig. 4c. The collected photocurrent of each pixel triggered earlier was lower than that of the pixel triggered later. As a result, a pre-processed trajectory containing both information from the passed pixels and the corresponding passing order was acquired. Panel (ii) of Fig. 4c summarizes and depicts the last collected current curves of all 100 pixels in the sensor array. Notably, the relatively long retention time of the current enabled the acquisition of a complete motion trace only by a single measurement after the motion was finished. Furthermore, the device could be easily reset to the initial stage by applying a global positive gate pulse, which enabled subsequent trajectory detection without interference by previous pre-processed information (panels (iii) and (iv) of Fig. 4c). Additionally, the sensor array also showed stable trajectory registration performance for the moving light spots with different combinations of velocity and light intensity (Supplementary Fig. 28).

The robustness and generalizability of the frameless paradigm with spontaneous response in the array enabled trace registration conducted in sensor terminals. To verify the detection reliability of the optical sensor array for sensing numerous trajectories, we developed an algorithm for motion recognition and set up corresponding datasets. Ten datasets, each containing 40 unique trajectories, were

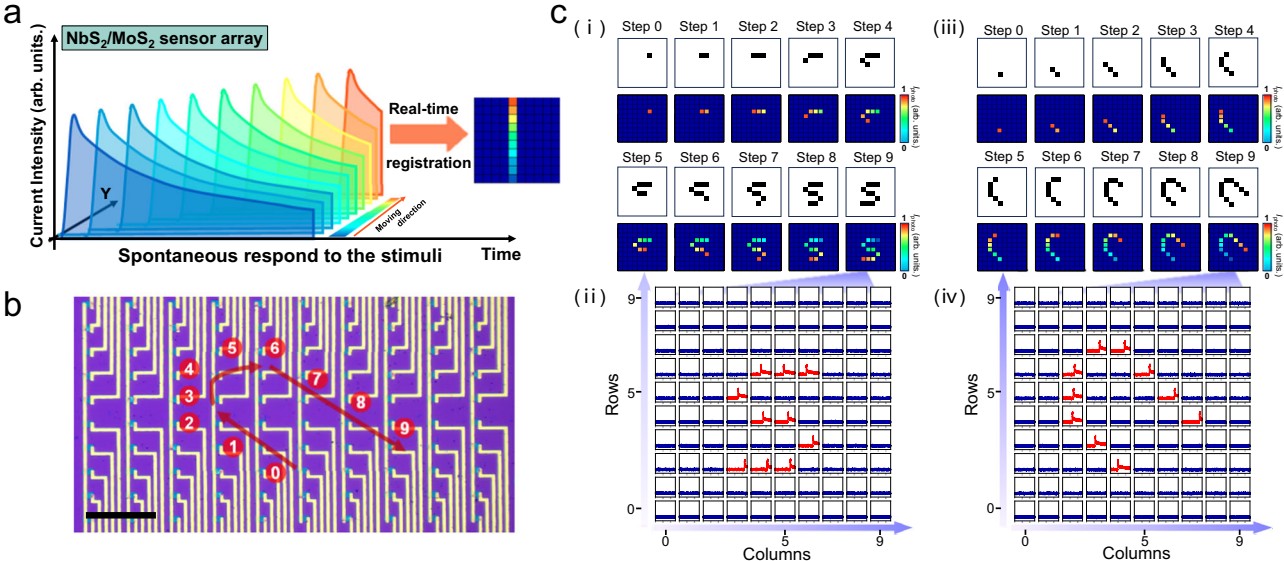

**Fig. 4 | Trajectory registration for moving light spot in neuro-inspired optical sensor array. a** Schematic illustration of in-sensor trajectories registration. **b** Predefined trace of moving targets on the sensor array. Scale bar: 200 μm. **c** (i) Acquired trajectory image in each step and (ii) corresponding current curves of all 100 pixels after 10 steps, (iii), (iv) Another trajectory image after erasure operation and corresponding current curves in each pixel. The velocity and light intensity of moving light spot are 40 μm s$^{-1}$ and 3.1 μW μm$^{-2}$, respectively.

initially generated using a random trajectory generating program. The starting point, ending point, and moving direction of the trajectory were all random, ensuring that the dataset was representative of a large population of possible trajectories. These datasets were then used to simulate the output current of the array over the entire duration of the trajectory with the consideration of device-to-device variance (Supplementary Figs. 29 and 30). Supplementary Fig. 31 shows the 12 different paths selected and their corresponding recovered trace images. These exported results were identical to the corresponding input trajectories from the dataset. Additionally, the detected accuracies of all 10 databases were above 92%, highlighting the stability of our trajectory detection scheme. The development of a NbS$_2$/MoS$_2$ neuro-inspired optical sensor array with front-end trajectory registration provides a promising strategy for sensing moving targets in emerging compact vision systems[81–83].

## Discussion

In summary, we developed a neuro-inspired optical sensor array consisting of a 10 × 10 phototransistor matrix based on NbS$_2$/MoS$_2$ hybrid films. The optical sensor array was effectively integrated with various functions including sensing, memory, and contrast enhancement, enabling a markedly enhanced recognition accuracy for static images. More importantly, in-sensor pre-processing of moving light spots was also experimentally performed, so that the post-processing yielded a high recovered accuracy of over 92% for the trajectories. The high-accuracy static image recognition and dynamic trace extraction that was highly associated with the nonlinear conductance photo-modulation in our optical sensor array remarkably improved the efficiency of data processing and contributed to system simplification. This work could facilitate the development of neuro-inspired image devices, providing opportunities for constructing compact and efficient artificial vision systems.

## Methods
### Synthesis of 2D NbS$_2$/MoS$_2$ hybrid films
Monolayer MoS$_2$ films were synthesized by atmospheric chemical vapor deposition (APCVD) using solution-based metal precursors. 0.03 M Na$_2$MoO$_4$ precursor was spin-coated on the pretreated SiO$_2$/Si substrate (15 mm × 15 mm) at 5000 rpm. A quartz boat loaded with S powder (~250 mg) was placed in the upper stream and heated to ~160 °C by a heating belt, and another quartz boat loaded with the as-prepared substrate was placed into the central zone of the tube furnace, where the temperature was ~800 °C. 50 sccm Ar was used as the carrier gas. After 5 min of CVD growth, monolayer MoS$_2$ films were obtained on the substrate.

For the synthesis of NbS$_2$/MoS$_2$ hybrid films, mixed powders of Nb$_2$O$_5$ (~480 mg) and NaCl (~80 mg) were loaded into a quartz boat at the central heating zone of the furnace, and the substrate coated with continuous MoS$_2$ films was placed upside down above the mixture. Excessive S powder (~1 g) placed at the upper stream was heated by a heating belt. The upper stream area and central zone of the furnace were heated to 160 °C and 840 °C within 40 min, respectively. The growth time was maintained for 40 min with 60 sccm Ar/H$_2$ mixed gas. After the reaction, the chamber was naturally cooled to room temperature.

### Materials characterization
The morphology of the as-grown NbS$_2$/MoS$_2$ hybrid films was characterized using OM (Zeiss, Axioscope 5), Scanning Electron Microscope (Zeiss, Quanta FEG 250 with 5 kV operation voltage), and AFM (Bruker, Icon). The binding energies of the elements were analyzed by using XPS spectroscopy (ESCALAB 250 Xi). The micro-Raman and micro-PL measurements were performed by using a confocal Raman spectrometer (Metatest corporation, Scan pro advance) with a 532 nm laser. HRTEM images and SAED patterns were collected by using an FEI Talos F200s transmission electron microscope with a 200 kV operation voltage.

### Device fabrication and measurement
The NbS$_2$/MoS$_2$ phototransistor was fabricated using laser direct writing photolithography, followed by e-beam evaporation for Ti/Au (10/50 nm) electrode deposition. A 10 × 10 image sensor array was fabricated on a SiO$_2$/Si substrate with a size of 25 × 25 mm$^2$. As shown in Supplementary Fig. 32, NbS$_2$/MoS$_2$ hybrid films were initially etched into a 10 × 10 isolated rectangle array with the assistance of photolithography and ICP etching. Electrodes and interconnections (Ti/Au: 15/75 nm) were then deposited to fabricate the MoS$_2$/NbS$_2$

phototransistor, in which the channel length and width were patterned to be $10 \times 20\,\mu m^2$, respectively. The sensor array was finally wire-bonded and electrically connected to a PGA ceramic carrier. ICP etching was performed with a pressure of 8 mTorr, 50 sccm $SF_6$, source power of 100 W, bias power of 20 W, and reaction time of 30 s. All electrical and optoelectronic characterizations were performed on a metatest scan pro laser scanning system (Metatest corporation, Scan pro advance). A 532 nm laser applied to devices was modulated by a digital modulator (Coherent, OBIS). Keithley 2612B was used to supply the source-drain bias voltage and collect data. All measurements were performed at atmospheric pressure and room temperature.

### Array measurement for the static and moving targets

Performance measurement of the sensor array for static targets was conducted with the assistance of an "H" shaped photomask. After patterning the specific letters on a glass substrate coated with AZ5214 photoresist, Ti/Au (20/100 nm) was deposited by electron-beam evaporation to block a certain area from light illumination. The active size of the photomask was consistent with that of the sensor array and could be effectively aligned according to the markers. The weight map was obtained by normalizing the photocurrent of sensors in the array, which was collected with a multichannel semiconductor analyzer integrated into the home-built testing system. To demonstrate the function of contrast enhancement, we divided the input image into two parts, including the target letter and background letter, which were separately projected on the sensor array. After training by input optical stimuli 100 times, the decaying current data of two parts are collected and map-plotted into the same image. For motion detection, a 532 nm continuous laser with different light intensities (1.6, 3.1 or 4.1 $\mu W\,\mu m^{-2}$) moving with a uniform speed along the pre-defined trace was utilized to simulate the movement of the target. In this process, the laser spot moved without stopping by using a piezoelectric displacement actuator. The moving time between adjacent pixels and effective illuminating time of each pixel for motion velocity of 20, 40 and 80 $\mu m\,s^{-1}$ were ~5 s/1 s, 2.5 s/0.5 s and 1.25 s/0.25 s, respectively.

## Data availability

The data generated in this study are provided in Source data. Extra data that support the findings of this study are available from the corresponding authors upon reasonable request. Source data are provided with this paper.

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

## Acknowledgements

This work was supported by National Natural Science Foundation of China (Nos. 52102161, 52273246, 62104091, and 51772064), National Key R&D Program of China (2022YFA1203802), Natural Science Foundation of Guangdong Province (Nos. 2021A1515012423 and 2022A1515011064), Young Innovative Talent Project Research Program of Guangdong Province (Grant No. 2021KQNCX077), Shenzhen Science and Technology Program (Nos. GXWD20201230155427003-20200805161204001, RCBS20200714114911270, RCJC20210706091950025, and KQTD20200820113045083).

## Author contributions

P.Y.H., B.Y.J. and H.J.C. contributed equally to the work. J.K.Q., F.C.Z. and C.Y.X. conceived the idea and proposed the research. P.Y.H., H.J. C. and L.Z. performed and supervised the growth experiment. P.Y.H., C.Y.Z., K.W., X.Y.H. and D.L. performed device fabrication and analyzed the experimental data. B.Y.J., J.Y.X., X.Y.H. and F.C.Z. carried out the simulation and analyzed corresponding data. J.K.Q., P.Y. H., F.C.Z. and C.Y.X. co-wrote the manuscript.

## Competing interests

The authors declare no competing interests.
