## [Peer Review File · Nature Communications]

REVIEWER COMMENTS

Reviewer #1 (Remarks to the Author):

This manuscript describes an optical sensor chip equipped with the NbS₂/MoS₂ hybrid film phototransistors. The authors claim that this chip is a retinomorph vision sensor that applies to static and dynamic object recognition. The basic optoelectrical properties of the transistor were examined experimentally, and the experimental results are presented reasonably. The slow kinetics and the flushable memory behavior of the photo-induced current could be utilized for unique applications. However, there are several flaws in the logic, explanations, and study design, at least in the current form of the manuscript. Besides, the originality of this study appears unclear and relatively weak when compared with the previous studies (reference #10-21). The detailed review comments are provided in the Adobe pdf file by the built-in comment function.

Reviewer #2 (Remarks to the Author):

The paper reports experimental results of a fabricated 10x10 array of phototransistor devices based on NbS₂/MoS₂ hybrid films. These devices exhibit retention properties of the illumination and the retention can be erased/modulated by applying voltage pulses on the device gate. Authors claim that these devices emulate biological retina and have very singular properties for filtering and motion detection.

The main shortcomings of the paper are:

Authors compare their device with standard CMOS vision sensors claiming that (page 3) “state of the art digital vision systems are developed based on CMOS platform consists of separated image sensor for receiving visual inputs, memories for storing information and processors for performing neuromorphic computation”. However, although this is a frequent approach, there are many examples of smart vision sensors published in the literature where photosensor devices are monolithically integrated with front-end retinomorph filtering and processing both in the analog and digital domain. See just a recent example

Wu, Jiangchao, et al. "A Multi-Mode CMOS Vision Sensor With On-Chip Motion Direction Detection and Simultaneous Energy Harvesting Capabilities." IEEE Sensors Journal (2022).

Or

Posch, Christoph, et al. "Retinomorph event-based vision sensors: bioinspired cameras with spiking output." Proceedings of the IEEE 102.10 (2014): 1470-1484.

In page 4, the authors claim “the trajectory tracing of dynamic objects was experimentally demonstrated in retinomorphic hardware systems for the first time”. This is a very strong claim which is not true, as many integrated CMOS vision arrays with motion detection have been published. See just one example. A. Simoni, G. Torelli, F. Maloberti, A. Sartori, S. E. Plevridis and A. N. Birbas, "A single-chip optical sensor with analog memory for motion detection," in IEEE Journal of Solid-State Circuits, vol. 30, no. 7, pp. 800-806, July 1995, doi: 10.1109/4.391119.

So the claim should be rephrased and properly put into context.

In page 12, the authors state “In conventional vision systems, the motion of objects is captured frame-by-frame based on CMOS image sensors platform. Although this is true, there are other neuromorphic CMOS vision sensors computing asynchronously temporal contrast at the pixel level. They are the so called dynamic vision sensors which are really the state of the art in CMOS for object motion detection based on vision. This technology should be referenced and used as baseline comparison.

In order to test the motion detection capabilities of the sensor a custom database of rocket trajectories is used and classified into ten groups obtaining a 92% accuracy in the classification. However, the accuracy is completely dependent on the complexity of the task. In order to evaluate the performance, it should be better explained how the dataset is acquired and how the 10 classes are determined. Furthermore, a comparison with an already existing dataset would be required for benchmarking.

Reviewer #3 (Remarks to the Author):

The manuscript of “Retinomorphic vision chip for high-accuracy in-sensor static and dynamic object recognition” written by Pei-Yu Huang et al, implemented retinomorphic sensor based on NbS₂/MoS₂, featuring remarkable photo-induced synaptic plasticity. The detection and recognition of moving object trajectory is impressive. The manuscript needs however major revision. After addressing the questions below, the manuscript may be considered for publication in this journal.

1. The authors need to show and describe “home-built testing system”. Figure 1d is not “home-built testing system”. (Figure 1d is photograph of the array and schematic of the phototransistor)
2. There are some grammar mistakes. For example: Page 6, Line 13.

3. In Fig. 2b, for the measurements under illumination, the power per unit area is quite large (16 mW). Does illumination have any irreversible on device operate? Also, provide incident optical power density.

4. Transfer and output curves of the device under illumination, and Fig. 2f were compared at different incident powers. It should be fairer to compare with equal powers.

5. Fig 2c, S8, and S9 do not clearly show PPC. After turning off the light, the current quickly approaches its initial state, which is ambiguous to call it pronounced PPC. Especially, in Figure S8, the current reaches about 87.5% of the initial current (i.e., dark current). Fig S10 better represents the PPC effect.

6. The cross-talk between adjacent pixels should be characterized. A detailed study of photocurrent mapping by scanning photo-current microscopy may be helpful.

7. Contrary to the cross-talk issue in question #6, the acquired trajectory image cannot be obtained as in Fig. 4c because the moving targets moves very wide as shown in Fig.4b. Also, the channel size of $10 \times 20 \text{ } \mu\text{m}^2$ and laser spot size of $5 \text{ } \mu\text{m}^2$ were used in this work. I am confused about that.

8. How did the authors calculate detectivity? The authors should measure the noise density spectrum and calculate the noise for real detectivity.

9. The authors should make references to these prior works for comprehensiveness (Nature communications 2021, 12, 3559, ACS Nano 2021, 15, 9, 15362–15370).

Responses to reviewers' comments (NCOMMS-22-41687)

We are grateful to the reviewers' comments on our works. We also thank very much for the reviewers' time and efforts to provide us detailed and valuable comments. These suggestions are greatly helpful to us for improving the quality of this work. We have addressed all the concerns raised by the reviewer point-by-point as follows, and highlight the revised parts in red color.

Our responses to reviewer 1's comments are from **page R1** to **page R72**, in which reviewer 1's comments by the built-in comment function in pdf were cited as screen capture images. The responses to reviewer 2's comments are from **page R73** to **page R87**, and the responses to reviewer 3's comments are from **page R88** to **page R101**.

Reviewer #1

This manuscript describes an optical sensor chip equipped with the NbS₂/MoS₂ hybrid film phototransistors. The authors claim that this chip is a retinomorphic vision sensor that applies to static and dynamic object recognition. The basic optoelectrical properties of the transistor were examined experimentally, and the experimental results are presented reasonably. The slow kinetics and the flushable memory behavior of the photo-induced current could be utilized for unique applications. However, there are several flaws in the logic, explanations, and study design, at least in the current form of the manuscript. Besides, the originality of this study appears unclear and relatively weak when compared with the previous studies (reference #10-21). The detailed review comments are provided in the Adobe pdf file by the built-in comment function.

(1) ~~Retinomorphic~~ vision chip for high-accuracy ~~in-sensor~~ static (2)
and dynamic object recognition

1. None of the features (i.e., the sensor dimension, the network/physical structure, the computational elements/mechanisms, the information encoding, and the computational functions) implemented in the proposed chip appears not to be of neuromorphic/retinomorphic ones. Therefore, the terms, "neuromorphic" and "retinomorphic", are both not appropriate to describe the proposed chip, and thus should be deleted in the title as well as the text.

Reply: Thanks very much for the reviewer's comments. We admit the term of "retinomorphic" is not appropriate in manuscript and deleted it, because the coding and physical structure of NbS₂/MoS₂ sensor array are distant from that of biological retina. Instead, we think that it might be better if the term of "neuromorphic" can be kept. The word "neuromorphic vision sensors" have been extensively used to describe the vision sensors that can emulate the **functions** of human retina (e.g., adaptive vision, contrast enhancement, noise reduction, edge extraction, motion extraction) with both conventional CMOS approaches and emerging device approaches, but not to exactly emulate the physical structure and spike coding schemes in the retina or visual cortex (*Neuromorphic electronics based on copying and pasting the brain. Nat. Electron. 2021, 4(9): 635–644; 2022 roadmap on neuromorphic computing and engineering. Neuromorph. Comput. Eng. 2022, 2: 022501; Curved neuromorphic image sensor array using a MoS₂-organic heterostructure inspired by the human visual recognition*

system. Nat. Commun. 2020, 11: 5934; A Bioinspired Retinomorph Device for Spontaneous Chromatic Adaptation. Adv. Mater. 2022, 34(51): e2206816).

Retina in human vision system can not only detect optical signals but also conduct pre-processes of vision information. Benefiting from the reduced redundant information in retina, high-level visual cognitive processes can be more effectively performed in the visual cortex. In contrast to conventional photosensors exhibiting transient conductance changes to light inputs, the altered conductance states of proposed NbS₂/MoS₂ sensor can be well maintained after the light stimuli removed, which can also be additionally updated *via* following stimuli. By mapping the updated conductance values of the sensor array, an input image can be captured and converted into a refined image. Therefore, it can simultaneously perform the functions of both optical sensing and pre-processing, and realize the part of the functions of artificial retina (*Artificial retinas—fast, versatile image processors. Nature 1994, 372(6502): 197-198*). Similar characteristics have been studied in previous reported neuromorphic vision sensors (*See for examples: Optoelectronic resistive random access memory for neuromorphic vision sensors. Nat. Nanotechnol. 2019, 14(8): 776-782; Curved neuromorphic image sensor array using a MoS₂-organic heterostructure inspired by the human visual recognition system. Nat. Commun. 2020, 11: 5934; Co-assembled perylene/graphene oxide photosensitive heterobilayer for efficient neuromorphics. Nat. Commun. 2022, 13: 4996*).

It should be noted that our sensor array allows the extraction of useful information from massive unstructured data and demonstrates the capabilities of selective pre-

processing for spatial-temporal information, such as noisy filtering, contrast enhancement, and motion trace extraction. In addition, the NbS₂/MoS₂ sensor array can also be integrated with artificial neural network for pre-processing to improve the recognize accuracy of static images, which is similar to the role of the retina in human vision system. More importantly, another distinguishing advantage is that NbS₂/MoS₂ sensor array can perform trajectory detection operation with less data storage, redundancy and latency. Each pixel will spontaneously respond to the stimuli and parallel conduct real-time memory of spatial-temporal information, which is also the consistent with frameless paradigm of biological vision.

Therefore, in the revised manuscript, the term of “retinomorphic” has been deleted, but we wish to keep the terms of “neuromorphic” and “bioinspired”. The neuromorphic and bioinspired characteristics are discussed in details from two aspects: (1) At the pixel level, the light-tunable synaptic plasticity, which is emulated by analog conductance states, has been demonstrated more clearly. (2) Bioinspired implementation of low-level processing in sensory terminals, including static information and dynamic trace extraction.

(1) Light-tunable synaptic plasticity at the pixel level

For optical artificial synaptic devices with phototransistor configuration, it is routine to emphasize function realization in biological synapse, although it is hard to define a clear presynaptic membrane or postsynaptic membrane in device (*See for examples: An optoelectronic synapse based on α -In₂Se₃ with controllable temporal dynamics for multimode and multiscale reservoir computing. Nat. Electron. 2022, 5(11):*

761-773; *Bioinspired mechano-photonic artificial synapse based on graphene/MoS₂ heterostructure. Sci. Adv. 2021, 7(12): eabd9117; Photonic synapses based on inorganic perovskite quantum dots for neuromorphic computing. Adv. Mater. 2018, 30(38): 1802883*). Herein, light pulses and channel current can be analogous to stimulation and postsynaptic current (PSC), respectively (**Fig. R1**). The generation and collection of light triggered carriers in channel are similar to the neurotransmitter release and uptake at the synaptic cleft¹. Therefore, synaptic weight corresponds to the change in channel conductance or resistance, and synaptic plasticity refers to the change of synaptic weight.

Fig. R1 Analogies between biological synapse and artificial optical synaptic device. **a** Schematic of $\alpha\text{-In}_2\text{Se}_3$ optoelectronic synapse. (*Adapted from: An optoelectronic synapse based on $\alpha\text{-In}_2\text{Se}_3$ with controllable temporal dynamics for multimode and multiscale reservoir computing. Nat. Electron. 2022, 5(11): 761-773.*) **b** Schematic diagram of mechano-photonic artificial synapse based on graphene/MoS₂ heterostructure. (*Adapted from: Bioinspired mechano-photonic artificial synapse based on graphene/MoS₂ heterostructure. Sci. Adv. 2021, 7(12): eabd9117*)

We optimized the experiments and modified corresponding explanations for clear expression, including light-dependent plasticity, paired-pulse facilitation (PPF), short-term plasticity (STP) and long-term plasticity (LTP). In the original manuscript, the

overall excitatory postsynaptic current (EPSC) is proportional to light intensity, duration, frequency, and pulse number, respectively, indicating that synaptic plasticity can be enhanced by increasing learning intensity, times, and rehearse speed. PPF, which is manifested as the enhancement in amplitude of two rapidly evoked excitatory postsynaptic potentials, is revised in **Fig. R2**. During operation, the power density of pulse is fixed at $3.1 \mu\text{W } \mu\text{m}^{-2}$, and pulse width is fixed at 500 ms. Even though the power density of stimuli keeps the same, the ΔI of latter is obviously higher than that of former, which could be attributed to the superposition of subsequent excited carriers with residual carriers before the following pulse in short pulse interval. The kinetic of carriers is similar to that residual Ca^{2+} , Na^+ , and K^+ in the active regions of a synapse enhance subsequent action potential in biological PPF ². Moreover, the interval-dependent attenuation curve matches well with the double exponential function. The relation time of slow stage (8.24 s) is an order of magnitude larger than that of rapid stage (0.21 s), indicating a successful stimuli process. These results are highly consistent with previously reported PPF behaviors in optically stimulated device. (*An optoelectronic synapse based on $\alpha\text{-In}_2\text{Se}_3$ with controllable temporal dynamics for multimode and multiscale reservoir computing. Nat. Electron. 2022, 5(11): 761-773; In-sensor reservoir computing system for latent fingerprint recognition with deep ultraviolet photo-synapses and memristor array. Nat. Commun. 2022, 13, 6590; Large-Area Pixelized Optoelectronic Neuromorphic Devices with Multispectral Light-Modulated Bidirectional Synaptic Circuits. Adv. Mater. 2021, 33(45): 2105017;*

Fig. R2 PPF index as a function of interval time (Δt). Insert is time-resolved photocurrent of device at two successive light pulses with width of 500 ms

Generally, the difference between STP and LTP can be defined based on the lifetime and residual level of postsynaptic current (*See for examples: Co-assembled perylene/graphene oxide photosensitive heterobilayer for efficient neuromorphics. Nat. Commun.* 2022, 13: 4996; *Curved neuromorphic image sensor array using a MoS₂-organic heterostructure inspired by the human visual recognition system. Nat. Commun.* 2020, 11: 5934; *A bioinspired optoelectronically engineered artificial neurorobotics device with sensorimotor functionalities. Nat. Commun.* 2019, 10: 3873; *Synaptic plasticity: taming the beast. Nat. Neurosci.* 2000, 3(11): 1178-1183). The emulation of STP and LTP in our device can be realized by repeatedly applying light pulses (**Fig. R3**). The EPSC, triggered by 2 successive pulses, declines rapidly to the initial stage in a short time of 20 s, which is defined as STP. In contrast, by applying 60 successive pulses with identical duration and intensity, the higher gain value of EPSC is obtained, and it

could keep about 30% of peak current after a long decay time of 350 s, resulting in the formation of LTP.

Fig. R3 STP and LTP processes are revised under 2 and 60 illumination pulses.

The emulation of learning-experience behavior for our device is also provided to further demonstrate the optical synaptic characteristics of our devices. According to Ebbinghaus's theory³, relearning information in biological synapses takes less time than the initial learning. As shown in **Fig. R4**, current reaches the maximum value of 2.75 nA and decays exponentially by applying 50 optical pulses at original learning stage. In the relearning stage, the device only requires 31 light pulses to approach the same current level, which is similar with previous reported synaptic devices. (*Co-assembled perylene/graphene oxide photosensitive heterobilayer for efficient neuromorphics. Nat. Commun. 2022, 13: 4996; Mimicking associative learning using an ion-trapping non-volatile synaptic organic electrochemical transistor. Nat. Commun. 2021, 12: 2480*)

Fig. R4 The learning-experience behavior of the optical stimulated device based on NbS₂/MoS₂ hybrid film.

(2) Bioinspired implementation of low-level processing in sensory terminals

In traditional CMOS-based vision system, sensors output real-time electrical signals proportional to the intensity of the input stimuli, whereas back-end computation circuits extract important temporal and spatial information. Inspired by the biologic vision, NbS₂/MoS₂ sensors can execute photoelectric signal conversion and information processing in the same physical unit, which significantly reduces data transfer and simplifies the system structure.

For example, the photocurrent in device decays faster as stimulated by light pulse with lower light intensity. After 100 successive pulse training, the differences of remaining current as well as contrast are further enlarged (**Fig. R5 a**). Therefore, the noise filtering for static image can be demonstrated without processing units. The left panel of **Fig. R5 b** shows the light intensity mapping of input image, including a target letter “H” with 1 mW and a faint noisy letter “T” with 0.5 mW. The image information is repeatedly input into the sensor array for 100 times. After decaying for 150 s, the

remaining currents of pixels are collected and map-plotted (right panel of **Fig. R5 b**). Since current triggered by lower light intensity decay faster, whereas current triggered by higher light intensity decay slower. The corresponding average values of pixels in main letter and noisy letter are normalized as 1 and 0.22, respectively. Compared with light intensity contrast of input image, the decayed current contrast between pixels with different light illumination is enlarged from 1:0.5 to 1:0.22 without using any external circuits, just like when people focus on a specific object, the rest of the surrounding information will be mostly filtered.

Fig. R5 Revised noise filtering function of NbS₂/MoS₂ sensor array. **a** decay behavior of light-induced current and contrast. **b** Light intensity mapping of input image (left) and current mapping of pre-processed image (right).

To evaluate the image recognition accuracy by integrating pre-processing component, we further construct an entire artificial vision system including a NbS₂/MoS₂ sensor array for pre-processing and an artificial neural network for image recognition (post-processing) (**Fig. R6 a**). Each initial image consists of a bright informative letter, a faint noisy letter, and background white noise. The light intensity ratio of bright informative letter and noisy letter are fixed at 1:0.5. The light intensity ratio of bright informative letter and white noise is randomly generated in range from 1:0 to 1:0.6. Such images are pre-processed by our devices and then inputted into CNN

for high-level recognition. **Fig. R6 b** exhibits several revised results before and after noise filtering. It is evident that both intensity of noisy letter and white noise are reduced. For quantitative comparison, the recognition accuracies of vision systems are given in **Fig. R6 c**. The vision system without pre-processing integration shows unsatisfied accuracy of 77% because of noise interference. In contrast, after integrating pre-processing module, the vision system could reach a considerably enhanced recognition rate about 90%. The results indicate that the integration of pre-processing enables extraction main information and results in the improvement of processing speed and accuracy.

Fig. R6 Revised image recognition with integrating pre-processing module, in which the utilization of improper dataset is corrected. **a** Schematic diagram of constructed artificial vision system, in which both a noise letter and background noise have been revised as noisy information. **b** Comparison of image before and after filtering has been revised. **c** Recognition accuracies of the vision system with and without filtering pre-processing.

Another example is the trajectory extraction within sensor array. As shown in **Fig. R7**, in contrast to the realization of front-end motion detection in frame-based devices relying on additional circuits. The decay kinetics of conductance state in individual NbS₂/MoS₂ phototransistor allows the array to direct process and generate the spatial-temporal current information according to the stimuli sequences. The photocurrent of each pixel triggered earlier is lower than that of the pixel triggered later at the same measured time point. Thus, the motion trace (*i.e.*, trace shape and motion direction) can be derived by directly reading the current of device. Each pixel located at trajectory will spontaneously respond to the stimuli and parallelly conduct real-time memory of spatial-temporal information, consistent with the frameless paradigm of biological vision. A complete motion trace can be obtained by only executing one measurement after the moving trajectory finished, which allows us to effectively avoid the massive data storage, redundancy, and latency.

Fig. R7 Comparison between frame-based camera and NbS₂/MoS₂ sensor for on-chip trace extraction.

In summary, NbS₂/MoS₂ sensor array can not only exhibit light-tunable synaptic plasticity at the pixel level, but also pre-process temporal and spatial optical information for high-accuracy static image recognition and dynamic trace extraction. The implementation of statics main image and dynamic trajectory extraction in NbS₂/MoS₂ sensor array is very similar to the functions of the retina in human vision system, where low-level processing tasks are moved to the sensory terminals. We also noted that most of reported optical synaptic devices with similar characteristics are described as “neuromorphic” (*Optoelectronic resistive random access memory for neuromorphic vision sensors. Nat. Nanotechnol. 2019, 14(8): 776-782; Co-assembled perylene/graphene oxide photosensitive heterobilayer for efficient neuromorphics. Nat. Commun. 2022, 13: 4996; Photonic synapses based on inorganic perovskite quantum dots for neuromorphic computing. Adv. Mater. 2018, 30(38): 1802883*) or “bioinspired” (*Curved neuromorphic image sensor array using a MoS₂-organic heterostructure inspired by the human visual recognition system. Nat. Commun. 2020, 11: 5934; A bioinspired retinomorphic device for spontaneous chromatic adaptation. Adv. Mater. 2022, 34(51), 2206816*) devices. Therefore, we think that it would be better if the terms of “neuromorphic” and “bioinspired” can be kept.

(1) ~~Retinomorphic~~ vision chip for high-accuracy ~~in-sensor~~ static (2)
and dynamic object recognition

2. This should be deleted, since neither the static letter classification nor the trajectory tracing can be computed in the proposed sensor chip.

Reply: Thanks for the reviewer’s suggestions. As discussed above, our device can conduct low-level processing tasks in-sensor including noise filtering and motion extraction, which is similar to retina in biological vision system. The high-level sensory processing including static letter classification or computing for trajectory tracing indeed can only be performed in our device by combining with corresponding neural network. Considering of this we have revised the title of manuscript with “Neuromorphic vision chip for high-accuracy static image recognition and dynamic trace extraction”.

(3)

induced synaptic plasticity and ultralow power consumption of 4.2 pJ per spike during a synaptic event. A ~~retinomorphic~~ vision chip with 100 pixels can operate with the

3. Although the power consumption might be ultra-low, the laser power consumption, which is also necessary for generating the event signal, appears not to be low (e.g. 0.1 mW laser for 0.5 seconds).

Reply: Thanks for the reviewer’s suggestions. We calculated the laser power consumption with different pulse widths by equation ^{4,5}:

$$dE = S \times P \times dt \quad (1 - 1)$$

where S is the area of the device, P is the power density of the optical pulse with the duration of t . The minimum energy consumption is estimated to be 1 μ J with 0.1 mW laser power and 10 ms duration, which is comparable as previous reported emerging device ^{6,7}.

(4)
induced synaptic plasticity and ultralow power consumption of 4.2 pJ per spike during
~~a~~ synaptic event. A ~~retinomorphic~~ vision chip with 100 pixels can operate with the
4. The 4.2-pJ consumption is just one example of the energy consumption in the case
when the 0.1-mW laser light pulse is provided for the duration of 0.5 seconds. The
energy consumption in the device depends on the pulse duration determined by the
external laser device. Therefore, the energy consumption is not appropriate to be used
for describing features of the proposed device itself. The energy/power transfer
efficiency, instead of the consumption, could be used.

Reply: Thanks for the reviewer's suggestions. Since the biological brain can handle
massive information with extremely low energy consumption, it is necessary to
investigate the energy consumption of synaptic devices.

Up to now, there is no consensus on the calculation of energy consumption for
optical synaptic devices, and the following two main methods have been used.

In the first method, the energy consumption is the light pulse energy consumption
^{4,5}, which is calculated by formula:

$$dE = S \times P \times dt \quad (1 - 2)$$

where S is the area of the device, and P is the power density of the optical pulse with
the duration of t .

In the second method, the energy consumption is electrical response of synaptic
device ⁸⁻¹⁵, which is used by formula:

$$dE = V \times I \times dt \quad (1 - 3)$$

where V and I are the operation voltage and photocurrent of the device, respectively, and t is the duration of optical pulse.

It is reasonable to characterize the device consumption by combining above two methods. Considering of this, we have modified the experiments, and the consumption of electrical response and external laser pulse with different light duration is given in **Fig. R8**. Under light pulse (0.1 mW, 10 ms) and fixed V_{ds} (50 mV), the minimum energy consumption is 0.42 pJ and 1 μ J for energy consumption of electrical response and laser pulse, respectively.

Fig. R8 Electrical and external laser energy consumption under applying 0.1 mW optical pulse.

As reviewer suggested, the External Quantum Efficiency (EQE), which refers to the ratio of the number of charge carriers collected to the number of incident photons, is also calculated to characteristic the energy/power transfer efficiency. The EQE value of device at $V_{ds}=50$ mV is obtained by equation:

$$EQE(\%) = \frac{I_{Ph}/e}{P_{Inc}/\hbar\nu} \times 100 \quad (1 - 4)$$

where I_{Ph} , P_{Inc} , e and $\hbar\nu$ are photocurrent, incident power density of the optical pulse, unit charge and photon energy, respectively.

The calculated EQE with dependence on pulse duration is shown in **Fig. R9**. It is obvious that under the same light illumination, the higher EQE requires high-level of photocurrent, which is unfavorable to reducing electrical consumption. Therefore, devices with low energy consumption usually present a barely satisfactory EQE.

Fig. R9 The characteristic of EQE under applying 0.1 mW optical pulse with different duration.

We have corrected the related description in main text, and **Fig. R8** has been added in Supplementary Information as **Fig.S13**.

*“The energy consumptions of electrical response for MoS₂/NbS₂ phototransistor with different pulse durations were summarized and plotted in **Supplementary Figures S13-S14**, from which we can identify a minimum value of 0.42 pJ with a pulse duration of 0.01 s.”*

(5)
induced synaptic plasticity and ultralow power consumption of 4.2 pJ per spike during
a synaptic event. A ~~retinomorphic~~ vision chip with 100 pixels can operate with the

5. There is no spike generation in the device and the “event” is provided as the light pulse from an external laser device. Thus, this is not appropriate to be used, and should be deleted.

Reply: Thanks for the reviewer’s suggestions. The words “spike” and “event” are indeed inappropriate for the device. Corresponding words have been replaced with “laser pulse” in the manuscript.

(6)
integrated functions of perception, memory, and filtering capabilities, delivering a high accuracy of about 94% for static image recognition. More importantly, detection and

6. This is inappropriate wording. For example, "sensing" or "receiving", but not "perception", could be used.

Reply: Thanks for the reviewer’s suggestions. The word “perception” has been replaced with “sensing”.

accuracy. The retina in human eye can not only obtain high-quality visual images but also conduct low-level sensory processing and temporal memorizing, thus allowing the

7. The computations/functions of human retinas have not been revealed yet. Thus, the definition of the "low-level" is uncertain. The information processing found in the biological retinas should be specified.

Reply: Thanks for the reviewer’s suggestions. Low-level (pre-processing) processing includes noise suppression, image sharpening and motion extraction, *etc.* (*Near-sensor and in-sensor computing. Nat. Electron. 2020, 3(11): 664-671*). Biologically visual system is a high complicated hierarchical organization, and it is evident that the

different levels of visual system process different types of visual information. The researchers often classify visual information processes into low-level sensory processes and high-level cognitive processes to design rational architecture of vision machines^{16,17}, although the computations/functions of human retinas have not been revealed yet. Generally, low-level visual processing of biology is concerned with determining different types of contrast among images projected onto the retina, while high-level visual processing refers to the cognitive processes that integrate visual information from a variety of sources into human consciousness (*Low-Level Visual Processing: The Retina. Principles of Neural Science, Fifth Edition, 2014*). Kydon human-like vision system is a successful example designed by above classification¹⁷. In Kydon project, the lower layers perform low-level vision-processing tasks, such as filtering, segmentation, contour finding, curve fitting, and object graph generation. The upper layers can perform high-level vision tasks, such as pattern recognition, abstraction, and image understanding and interpretation. In 2003, Bourbakis *et al.* proposed a retina-like vision processor based on the lower part of Kydon system¹⁸. A certain set of low-level vision tasks, such as smoothing and light adaptation, edge detection. *etc.* are demonstrated. Recently, Chai *et al.* also defined low-level sensory processing, which can preliminarily and selectively extract useful data from a large volume of raw data by suppressing unwanted noise or distortion, or by enhancing the feature for further processing. Such low-level sensory processing including noise suppression, filtering, background extraction, feature enhancement, motion extraction, can effectively reduce the computational load and improve the efficiency for high-level processing tasks¹⁶.

The related description has been updated in main text:

“The retina in human eye can not only obtain high-quality visual images but also conduct low-level sensory processing such as noise suppression, image sharpening and motion extraction, etc.”

(8)

could mimic the colored and color-mixed recognition of human retina relying on the linear conductance change of channel under illumination ²⁷. However, previously

8. This is incorrect wording. The biological retinas do NOT recognize the visual information, but dynamically process and encode it into the biological (all-or-none) spike signals at every moment.

Reply: Thanks for the reviewer’s suggestions. We have revised the corresponding description in main text.

“Park et al. also reported an optic-neural synaptic device based on 2D h-BN/WSe₂ heterostructures. The intriguing chrominance-dependent synaptic dynamics allows the designing of artificial vision system for colored-mixed pattern recognition.”

hardware-implemented retinomorphie vision system, which monolithically integrates

(9)

the function of recognizing both static and moving targets ~~as the biological retina.~~

9. This should be deleted. The biological retinas do NOT "recognize" the information.

Reply: Thanks for the reviewer’s suggestions. We revised the word “recognizing” with “pre-processing”.

(10)

photoconductivity (PPC), featuring diversified synaptic functions such as paired pulse facilitation (PPF), the transition from short-term plasticity (STP) to longer-term plasticity (LTP) as well as long-term synaptic potentiation and inhibition (LTP/LTD).

10. The behaviors of the phototransistor presented in this study are distant from the biological synaptic functions, namely, PPF, STP, LTP, and LTD. The authors should reconsider the wordings/phrases here.

Reply: Thanks for the reviewer’s suggestions. We admit there are difference between the behavior of our phototransistor and biological synaptic functions. “Neuromorphic engineering” emerged as an interdisciplinary research field. It initially focuses on building electronic neural processing systems to directly emulate neuroscience principles where detailed synaptic chemical dynamics with high-fidelity are mandatory, but the definition of “neuromorphic” has recently been relaxed to emulate the qualitative features of biological neural systems for emerging neuromorphic device.

(Neuromorphic electronics based on copying and pasting the brain. Nat. Electron. 2021, 4(9): 635–644; An optoelectronic synapse based on α -In₂Se₃ with controllable temporal dynamics for multimode and multiscale reservoir computing. Nat. Electron. 2022, 5(11): 761-773; Co-assembled perylene/graphene oxide photosensitive heterobilayer for efficient neuromorphics. Nat. Commun. 2022, 13: 4996; A flexible ultrasensitive optoelectronic sensor array for neuromorphic vision systems. Nat. Commun. 2021, 12: 1798; Bioinspired mechano-photonic artificial synapse based on graphene/MoS₂ heterostructure. Sci. Adv. 2021, 7(12): eabd9117; Photonic synapses based on

inorganic perovskite quantum dots for neuromorphic computing. Adv. Mater. 2018, 30(38): 1802883).

For example, as shown in **Fig. R10**, PPF in biological synapses is manifested as the enhancement in amplitude of two rapidly evoked excitatory postsynaptic potentials¹⁹. The artificial synaptic device focuses on emulation of enhancement function for second stimuli-induced response by using analog channel current^{11,12,20,21}, although the mechanism and details of facilitation between device and biological might be different. As shown in **Fig. R11**, during operation, the power density of pulse is fixed at $3.1 \mu\text{W} \mu\text{m}^{-2}$, and pulse width is fixed at 500 ms. Even though the power density of stimuli keeps the same, the ΔI of latter is obviously higher than that of former, which could be attributed to the superposition of subsequent excited carriers with residual carriers before the following pulse in short pulse interval. Note that residual Ca^{2+} , Na^+ , and K^+ in the active regions of a synapse enhance subsequent action potential in biological PPF². Therefore, a simple analogy between PPF behavior of biologic synapses and that of artificial device has been proposed (**Fig. R12**) (*An optoelectronic synapse based on $\alpha\text{-In}_2\text{Se}_3$ with controllable temporal dynamics for multimode and multiscale reservoir computing. Nat. Electron. 2022, 5(11): 761-773*). Moreover, the interval-dependent attenuation curve matches well with the double exponential function, in which the maximum and minimum value of PPF index was calculated to be 129% and 113%, respectively. The relation time of slow stage (8.24s) is an order of magnitude larger than that of rapid stage (206 ms), indicating a successful stimuli process. These results are highly consistent with previously reported PPF behaviors in optically stimulated

synaptic devices (**Fig. R12**). (See for examples: *An optoelectronic synapse based on α - In_2Se_3 with controllable temporal dynamics for multimode and multiscale reservoir computing. Nat. Electron. 2022, 5(11): 761-773; In-sensor reservoir computing system for latent fingerprint recognition with deep ultraviolet photo-synapses and memristor array. Nat. Commun. 2022, 13: 6590; Large-Area Pixelized Optoelectronic Neuromorphic Devices with Multispectral Light-Modulated Bidirectional Synaptic Circuits. Adv. Mater. 2021, 33(45): 2105017; Realization of an Artificial Visual Nervous System using an Integrated Optoelectronic Device Array. Adv. Mater. 2021, 33(51): 2105485)*

Fig. R10 PPF occurs in biological synapses (*Adapted from: Short-term synaptic plasticity. Annu. Rev. Physiol. 2002, 64: 355-405*).

Fig. R11 PPF index of our device. The inset image plots the time-resolved photocurrent of device at two successive light pulses with width of 500 ms.

Fig. R12 PPF characteristic of previous reported device. **a** Schematic illustration and **b** PPF index of $\alpha\text{-In}_2\text{Se}_3$ optoelectronic synapse (*Adapted from: An optoelectronic synapse based on $\alpha\text{-In}_2\text{Se}_3$ with controllable temporal dynamics for multimode and multiscale reservoir computing. Nat. Electron. 2022, 5(11): 761-773*).

In neuroscience, long-term potentiation (LTP) is a persistent strengthening of synapses based on recent patterns of activity, while long-term depression (LTD) produces a long-lasting decrease in synaptic strength²². In extended neuromorphic engineering, it is routine, at least in present, to measure the lifetime and residual level of postsynaptic current to represent the synaptic strength for optically stimulated synaptic device (*Co-assembled perylene/graphene oxide photosensitive heterobilayer for efficient neuromorphics. Nat. Commun. 2022, 13: 4996; A bioinspired optoelectronically engineered artificial neurobotics device with sensorimotor functionalities. Nat. Commun. 2019, 10: 3873; Short-term plasticity and long-term potentiation mimicked in single inorganic synapses. Nat. Mater. 2011, 10: 591-595*).

For example, as shown in **Fig. R13a**, STP and LTP characteristics of optically stimulated device have been reported by Zhang *et al.* (*Co-assembled perylene/graphene*

oxide photosensitive heterobilayer for efficient neuromorphics. *Nat. Commun.* 2022, 13: 4996). STP is defined as when EPSC drops to 40 pA in a short interval of 30 s by applying 2 light pulses. For LTP, it takes 160 s to descend to the same current level after applying 13 light pulses. Furthermore, the light-controlled LTP and gate voltage controlled LTD behaviors are also reported in optical synaptic devices (**Fig. R13b** and **c**), in which the channel current of synaptic device is persistently enhanced and decreased by applying a train of optical and electrical stimuli, respectively. (*Photo-induced non-volatile VO₂ phase transition for neuromorphic ultraviolet sensors. Nat. Commun.* 2022, 13: 1729; *Photonic synapses based on inorganic perovskite quantum dots for neuromorphic computing. Adv. Mater.* 2018, 30(38): 1802883)

Fig. R13 STP, LTP and LTD of previous reported device. **a** Similar STP and LTP features of optical device reported by Zhang *et al.* (*Adapted from: Co-assembled perylene/graphene oxide photosensitive heterobilayer for efficient neuromorphics. Nat. Commun.* 2022, 13: 4996). **b-c** Similar LTP and LTD features of optical device reported by Li *et al.* (*Adapted from: Photo-induced non-volatile VO₂ phase transition for neuromorphic ultraviolet sensors. Nat. Commun.* 2022, 13: 1729).

Similarly, the demonstration of STP and LTP are revised in our device with the repeated light pulses. **Fig. R14** shows the value of EPSC for different numbers of light pulses. The EPSC, triggered by 2 successive pulses, declines rapidly to initial stage in a short time of 20 s, which is conceptually defined as STP. In contrast, by applying 60 successive pulses with same duration and intensity, the higher gain value of EPSC is obtained, and it could keep about 30% of peak current after a long decay time of 350 s, resulting in the formation of LTP.

Fig. R14 Revised demonstration of STP and LTP for our devices under 2 and 60 illumination pulses.

Figs. R11 and **R14** are updated as part of **Fig. 2**, and the related description is updated in main text:

“The τ_1 is one order of magnitude larger than τ_2 , indicating a successful stimuli process, consistent with previously reported optically stimulated synaptic devices.”

*“As shown in **Fig. 2f**, after applying two successive pulses with duration of 300 ms, photocurrent declines rapidly to the initial stage within 20 s, which conceptually refers to STP. LTP is achieved by applying 60 successive pulses, and 30% of peak photocurrent can be kept even after a long decay time of 350 s.”*

(11)

capabilities for static images, which delivers a high image recognition accuracy in artificial vision system. More importantly, the trajectory tracing of dynamic objects was same dataset and procedure. The recognition accuracies of two vision systems are given

(12)

in Fig. 3f. One can see that a remarkable improvement of recognition rate is achieved

11&12. This is not a fare evaluation. The accuracy of the letter classification by the artificial neural network largely depends on the degree of optimization of the network model with a given dataset. The relatively low accuracy can be realized by intentionally using the improper dataset, and this obviously leads the manipulative result. As described in the review comment for the abstract, the evaluation with CNN trained with the improper input image dataset is not fare. The low accuracy was intentionally generated by using the image with high contrast target letters overlapped with low contrast non-target letters. It is hardly possible to follow the authors' reasoning why the CNN evaluation is conducted with such manipulative data in this study.

Reply: Thanks for the reviewer's comments on our results of image recognition. We are sorry that the description was not clear in original manuscript. The purpose of experiment is to confirm whether the pre-processing function of our sensor array plays a role on the improvement of image recognition accuracy.

We compare the image recognition accuracy between an original dataset and a filtered dataset that has been pre-processed by NbS₂/MoS₂ sensor array. In the original dataset, the noise information containing both a noise letter and random background noise are overlapped to informative letter images. The filtered dataset was obtained by

pro-processing the original dataset with NbS₂/MoS₂ sensor array. Both datasets were then recognized using the same neural network with identical structure, and the training and evaluating procedures were also identical. Therefore, the only variable being manipulated in the experiment was the main information extraction function of our sensor array.

To further increase the reliability of our evaluation, we employed two different types of neural networks (artificial neural network (ANN) and CNN) to recognize the unfiltered and filtered datasets, as shown in **Fig. R15**. The results are presented in **Fig. R16**, which demonstrate the difference in recognition accuracy between the unfiltered and filtered datasets for both networks. These results provide robust evidences for the effectiveness of the array noise filtering on high-accuracy image recognition.

Fig. R15 Flowchart of the experimental procedure for verifying the noise filtering function of an NbS₂/MoS₂ hybrid film transistor array. Two datasets, filtered and unfiltered, were generated

and recognized by two types of neural networks: **a** artificial neural network (ANN) and **b** convolutional neural network (CNN).

Fig. R16 Recognition accuracies of **a** ANN and **b** CNN on filtered and unfiltered datasets.

Details on the stimulated filtering pre-processing of dataset is introduced as below:

As shown in **Fig. R17**, handwritten A-J letters images are taken from EMIST dataset as informative letter images, including 24000 training images and 4000 test images. The size of each image is 28×28 pixels. The unfiltered dataset is created by overlapping a dark noisy letter and additional random background noise to informative letter images. The light intensity ratio of bright informative letter and dark noisy letter are fixed at 1:0.5. The light intensity ratio of bright informative letter and background noise are randomly generated in the range of 1:0 to 1:0.6.

Fig. R17 Generation of filtered and unfiltered datasets

The filtered dataset can be obtained from unfiltered dataset through Python-simulated filtering pre-processing of NbS₂/MoS₂ sensor array. The simulation process can be divided into three steps:

(1) The input light intensity for each device in array is determined by the grayscale level of the corresponding pixel in unfiltered image. The adopted light stimulus is 100 successive light pulses with width of 300 ms. The output current of the simulated array is obtained by fitting the relationship between the input light intensity and the output current with actual experimental data. The fitting curve of output current with respect to input light intensity is demonstrated in **Fig. R18**.

(2) Device-to-device output current variance is considered. The effect of variance on the array output current is simulated by adding a set of Gaussian noise generated based on the calculated device-to-device variance. The device-to-device output variance of the array is determined by extracting the average variation of the dark current and photocurrents of the devices under different light intensity from experimental data.

(3) The final simulated current level of the array is obtained by adding the fitted current value and the simulated variance value. The normalized output current mapping of the array is treated as the pre-processed image in the filtered dataset.

Fig. R18 Fitting curve of output current with respect to normalized light intensity. The adopted light stimulus for experimental data is 100 successive light pulses with a width of 300 ms.

Figs. R15-R18 have been added in Supplementary Information as **Fig. S17-S20** to demonstrate the image recognition process, and related description have also been added in Supplementary Information.

“Construction of datasets and processing procedure for noise filtering

Details on the construction and of datasets and processing procedure for noise filtering are introduced as below:

*As shown in **Figure S17**, handwritten A-J letters images are taken from EMIST dataset as informative letter image, including 24000 training images and 4000 test images. The size of each image is 28×28 pixels.*

The unfiltered dataset is created by overlapping a dark noisy letter and additional random background noise to informative letter images for closing to real scenarios. The light intensity ratio of bright informative letter and dark noisy letter are fixed at

1:0.5. The light intensity ratio of bright informative letter and background noise are randomly generated from range 1:0 to 1:0.6.

Then, filtered dataset is obtained through Python-simulated filtering pre-processing of 28×28 NbS₂/MoS₂ sensor array. The whole simulation can be divided into three steps:

- (4) The input light intensity for each sensor in the 28×28 array is determined by the gray scale level of the corresponding pixel in the 28×28 sized unfiltered image. The adopted light stimulus is 100 successive light pulses with width of 300 ms. The output current of the simulated 28×28 array is obtained by fitting the relationship between the input light intensity and the output current with actual experimental data. The fitting curve of output current with respect to input light intensity is demonstrated in **Figure S18**.
- (5) Device-to-device variance of output current is also considered in simulating array output current. The effect of variance on the array output current is simulated by adding a set of Gaussian noise generated based on the calculated variance. The device-to-device output variance of the array is determined by extracting the average variation of the dark current and photocurrents of the devices under different light intensity from experimental data.
- (6) The final simulated current level of the array is obtained by adding the fitted output current and the simulated device-to-device output current variance. The normalized output current mapping of the array is treated as the pre-processed image in the filtered dataset.”

“Evaluating the effects of array noise filtering pre-processing for image recognition accuracy with both CNN and ANN

To verify the improvement in image recognition accuracy with integrating array noise filtering pre-processing, the unfiltered and filtered datasets were recognized using two different types of neural networks. As illustrated in **Figure S19**, both the unfiltered and filtered datasets were processed using the same artificial neural network (ANN) and the same convolutional neural network (CNN). The results are presented in **Figure**

S20. From results of both ANN and CNN, we can verify the effectiveness of the array noise filtering function on image recognition.”

(13)

experimentally demonstrated in ~~retinomorphic~~ hardware system ~~for the first time~~, showing a robust detection reliability in real scenarios. The development of

13. This should be deleted because this is not exactly true (see, for example, Okuno H, Hasegawa J, Sanada T, Yagi T. Real-time emulator for reproducing graded potentials in vertebrate retina. IEEE Trans Biomed Circuits Syst. 2015 Apr;9(2):284-95. doi: 10.1109/TBCAS.2014.2327103.).

Reply: Thanks for the reviewer’s suggestions. We revised the corresponding description as “**the trajectory detection and real-time memorization of dynamic objects was also experimentally demonstrated**”.

(14)

Fig. 1a depicts a schematic illustration of human visual system including eyeballs, optical nerve fibers, and brain visual cortex. The outer photoreceptor cells convert light

14. The panel 'a' should be deleted, because it is scientifically incorrect (for example, the LGN is missing; the optic nerve does not send the "pre-processed image" but the spike-encoded information), and may mis-lead naive readers.

Reply: Thanks for the reviewer’s suggestions. **Fig. 1a** schematically illustrates high complicated hierarchical organization of biologically visual system, in which the integration of sensing and pre-processing function in fore-end is emulated by NbS₂/MoS₂ sensor array. The revised figure is provided in **Fig. R19**. LGN has been marked, and the information transfer with spike-encoded process has been revised

Fig. R19 Schematic of human visual system.

We have corrected the related description in main text, and **Fig. R19** is updated as part of **Fig. 1** in main text.

“Fig. 1a depicts a schematic illustration of main parts in human visual system including retina, optic nerve, lateral geniculate nucleus (LGN) and visual cortex.”

(15)

stimulus into electrical potential change ²⁸, which are delivered to bipolar cells and ganglion cells for pre-processing to filter out the redundant visual data. The transformed

15. The horizontal cells and the amacrine cells are not mentioned, although they are essential elements for the information processing.

Reply: Thanks for the reviewer’s suggestions. We revised the corresponding description (page 5 in the revised manuscript):

“Light signals are first converted by photoreceptor cells and then pre-processed with the assistance of nerve cells including bipolar cells, horizontal cells, amacrine cells,

and ganglion cells, thus enabling the filtration of redundant and unstructured visual data”.

(16)

and pre-processed information is then transmitted to brain visual cortex through the optical nerve fibers for final processing and storage²⁹. Such unique information pre-

16. The LGN is missing.

Reply: Thanks for the reviewer’s suggestions. We revised the corresponding description:

“LGN subsequently receives and transmits information from retina to brain cortex for the cognition of visual stimuli.”

and pre-processed information is then transmitted to brain visual cortex through the optical nerve fibers for final processing and storage²⁹. Such unique information pre-

(17)

17. This term does not fit here. The visual cortex is not considered to store the information. The reference is a review paper about the retina but not cortex.

Reply: Thanks for the reviewer’s suggestions. We deleted the corresponding description and related reference.

(18)

optical nerve fibers for final processing and storage²⁹. Such unique information pre-processing schemes imbue brain visual cortex with high efficiency for the cognitive process of massive visual data.

18. The meaning is unclear.

Reply: Thanks for the reviewer’s suggestions. We revised the corresponding description: *“The functions in terms of noise filtering, contour finding, and object graph*

generation of retina imbue brain cortex with high efficiency for high-level processing of massive visual data processing.”

optical nerve fibers for final processing and storage²⁹. Such unique information pre-
(19)
processing schemes imbue brain visual cortex with high efficiency for the cognitive
process of massive visual data.

19. The visual cortex is not the only area cognitive process

Reply: Thanks for the reviewer’s suggestions. We deleted the corresponding words.

(20)
Fig. 2b and **Supplementary Figure S5** exhibit the transfer ($I_{ds}-V_{gs}$) and output characteristics ($I_{ds}-V_{ds}$) of NbS₂/MoS₂ phototransistor under 532 nm laser with different

20. All the properties shown in Fig. 2b and the related figures in supplementary materials should be those of the statistical data across multiple samples of the phototransister to demonstrate the practical usefulness of the in-sensor image processing.

Reply: Thanks for the reviewer’s suggestions. We measured the device-to-device variation and analyzed the data as previous study proposed²³. **Fig. R20a** exhibits the transfer curves of 100 NbS₂/MoS₂ phototransistors. The statistical distribution of ON/OFF ratio ($R_{ON/OFF}$), on-state current (I_{on}) and threshold voltage (V_{th}) are given in **Fig. R20b-d**, respectively. The mean values and corresponding standard deviations for $R_{ON/OFF}$, I_{on} , and V_{th} are calculated to be about $7.18 \pm 8.93 \times 10^2$, 6.04 ± 4.78 nA and 49.7 ± 3.4 V, respectively. We also investigated the pixel variation in the photoresponse. **Fig. R20e** and **f** show photoresponse curve of 100 pixels under 532 nm illumination with

intensity of 0.1 mW. The mean values and corresponding standard deviation for dark current (I_{dark}) and photocurrent (I_{ph}) are found to be 0.14 ± 0.09 nA and 1.87 ± 0.42 nA, respectively. These results indicate the excellent uniformity of array, which is critical for acquisition of high-quality filtered images.

Fig. R20 Device-to-device variations in the characteristics of NbS₂/MoS₂ phototransistors array. **a** transfer curves for 100 NbS₂/MoS₂ transistors of the device. **b** Statistical distribution of ON/OFF ratio is fitted with a logarithmic normal curve, in which the mean values and corresponding standard deviation are 7.18×10^2 and 8.93×10^2 , respectively. **c** The ON current (I_{on}) statistical distribution with mean values of 6.04 nA and corresponding standard deviation of 4.78 nA. **d** Threshold voltage (V_{th}) statistical distribution with mean values of 49.7 V and corresponding standard deviation of 3.4 V. **e-f** Dark current (I_{dark}) and photocurrent (I_{ph}) statistical distribution, in which the mean values and corresponding standard deviation for I_{dark} and I_{ph} are found to be 0.14 nA, 0.09 nA and 1.87 nA, 0.42 nA, respectively.

Fig. R20 has been added in Supplementary Information as **Fig. S15**, and related description has been updated in main text:

“The device-to-device variation was first statistically evaluated (Supplementary Figure S15). The 100% device yield and excellent uniformity of performance metrics are crucial for the acquisition of high-quality images.”

(21)

functions of biological synapse. The light spike can serve as stimulation on presynaptic neuron, while the amplitude of I_{ds} represents the conductance change in postsynaptic neuron. The paired-pulse facilitation (PPF) was first demonstrated by applying two
21. This does not make sense at all. If the light pulse is assumed to be the stimulus on presynaptic neuron, as described here, then the presynaptic neuron is supposed to be the NbS₂/MoS₂ phototransistor, and in turn, the current I_{ds} should be treated as the response of the presynaptic neuron. At least in a biological neural circuit, the (chemical) synapse has the function of the signal transmission from the presynaptic neuron to the postsynaptic neuron. In the proposed chip, there is no element representing a post-synaptic neuron nor its response. Hence, a synapse appears not to be definable in the chip.

Reply: Thanks for the reviewer’s suggestions. The description of the synaptic device is misleading in the original manuscript. As the replies for comments #1 and #10 mentioned, rigorous mimicry of the neuronal network implemented with emerging devices has proved difficult, and the neuromorphic engineering focuses on the emulation of qualitative features of biological neural systems (*Neuromorphic electronics based on copying and pasting the brain. Nat. Electron. 2021, 4(9): 635–644*). Especially for optical synaptic devices with phototransistor configuration, it is

routine to emphasize the function mimicking the signal transmission in biological synapses, although it is hard to define a clear presynaptic membrane or postsynaptic membrane due to the three-terminal structure of device (*An optoelectronic synapse based on α -In₂Se₃ with controllable temporal dynamics for multimode and multiscale reservoir computing. Nat. Electron. 2022, 5(11): 761-773; In-sensor reservoir computing system for latent fingerprint recognition with deep ultraviolet photo-synapses and memristor array. Nat. Commun. 2022, 13: 6590; Bioinspired mechano-photonic artificial synapse based on graphene/MoS₂ heterostructure. Sci. Adv. 2021, 7(12): eabd9117; A MoS₂/PTCDA hybrid heterojunction synapse with efficient photoelectric dual modulation and versatility. Adv. Mater. 2019, 31(3): 1806227; Photonic synapses based on inorganic perovskite quantum dots for neuromorphic computing. Adv. Mater. 2018, 30(38): 1802883).*

The dynamic response of postsynaptic current to stimulation is a critical feature of biological synapse, which refers to synaptic plasticity. Similarly, NbS₂/MoS₂ phototransistor exhibits nonlinear relationships between light stimuli and the output photocurrent magnitude, thus the light pulses and channel current can be analogous to stimulation and postsynaptic current, respectively. The generation and collection of light-triggered carriers in the channel are similar to the processes of neurotransmitter release and uptake at the synaptic cleft¹, in which the synaptic weight modulation corresponds to the change in channel conductance or resistance.

Similar devices have been widely reported in previous studies^{12,20,24,25}. For example, **Fig. R21a** schematically compares an optoelectronic synaptic device with

biological synapse (*An optoelectronic synapse based on α -In₂Se₃ with controllable temporal dynamics for multimode and multiscale reservoir computing. Nat. Electron. 2022, 5(11): 761-773*). By tuning the light intensity, the device exhibits STP behaviors with enhanced excitatory PSC (**Fig. R21b**). Another example is mechano-photonic artificial synapse (*Bioinspired mechano-photonic artificial synapse based on graphene/MoS₂ heterostructure. Sci. Adv. 2021, 7(12): eabd9117*), where effective modulation of light-dependent PSC can be realized by integrating a triboelectric nanogenerator for supplying gate voltage (**Fig. R21c-d**).

Fig. R21 Previously reported optoelectronic synaptic device. **a** Analogy between α -In₂Se₃ optoelectronic synaptic device and biological synapse. **b** Excitatory PSC with different light duration time and stimulation intensity (*Adapted from: An optoelectronic synapse based on α -In₂Se₃ with controllable temporal dynamics for multimode and multiscale reservoir computing. Nat. Electron. 2022, 5(11): 761-773*). **c** Schematic illustration of mechano-photonic artificial synaptic transistor. **d** PSC with different light stimulation intensity (*Adapted from: Bioinspired mechano-photonic artificial synapse based on graphene/MoS₂ heterostructure. Sci. Adv. 2021, 7(12): eabd9117*).

The related description in main text has been corrected:

“The tunable memory characteristics allow the emulation of synaptic plasticity based on NbS₂/MoS₂ phototransistor. Generation and collection of light-triggered carriers in the channel are analogy to the processes of neurotransmitter release and uptake at the synaptic cleft, respectively, and the change in channel conductance or resistance corresponds to the synaptic weight modulation.”

(22)

functions of biological synapse. The light spike can serve as stimulation on presynaptic neuron, while the amplitude of I_{ds} represents the conductance change in postsynaptic neuron. The paired-pulse facilitation (PPF) was first demonstrated by applying two

22. "pulse". The term "spike" may mislead the readers.

Reply: Thanks for the reviewer’s suggestions. The corresponding words have been revised with “pulse”.

(23)

neuron. The paired-pulse facilitation (PPF) was first demonstrated by applying two successive light pulses with duration of 500 ms and different interval durations at V_g

23. The behavior shown here appears not to be facilitation, but simply reflects the linear temporal convolution of the second response with the first response due to the slow response decay after the offset of light pulse. Obviously, the time course of the "PPF index" described by Equation (1) is almost identical to that of the response decay. Therefore, the term "paired-pulse facilitation" is not appropriate to express such a simple linear behavior.

Reply: Thanks for the reviewer’s suggestions. We think the behavior of device induced by two successive optical pulses can be described as the term “paired-pulse facilitation” in extended neuromorphic engineering.

In neuroscience, PPF is manifested as the enhancement in amplitude of two rapidly evoked excitatory postsynaptic potentials. As reply for comment #10 mentioned, the artificial synaptic device with emerging materials focuses on emulation of enhancement function for second stimuli-induced response by using analog channel current^{11,12,20,21}, although the mechanism and details of facilitation between device and biological might be different. As shown in **Fig. R22**, during operation, the power density of pulse is fixed at $3.1 \mu\text{W } \mu\text{m}^{-2}$, and pulse width is fixed at 500 ms. Even though the power density of stimuli keeps the same, the ΔI of latter is obviously higher than that of former, which could be attributed to the superposition of subsequent excited carriers with residual carriers before the following pulse in short pulse interval. Note that residual Ca^{2+} , Na^{+} , and K^{+} in the active regions of a synapse enhance subsequent action potential in biological PPF². Therefore, a simple analogy between PPF behavior of biologic synapses and that of artificial device has been proposed (**Fig. R23**) (*An optoelectronic synapse based on $\alpha\text{-In}_2\text{Se}_3$ with controllable temporal dynamics for multimode and multiscale reservoir computing. Nat. Electron. 2022, 5(11): 761-773*). Moreover, the interval-dependent attenuation curve matches well with the double exponential function, in which the maximum and minimum value of PPF index was calculated to be 129% and 113%, respectively. The relation time of slow stage (8.24s) is an order of magnitude larger than that of rapid stage (206 ms), indicating a successful stimuli process. These

results are highly consistent with previously reported PPF behaviors in optically stimulated synaptic devices (**Fig. R24**). (*An optoelectronic synapse based on α -In₂Se₃ with controllable temporal dynamics for multimode and multiscale reservoir computing. Nat. Electron. 2022, 5(11): 761-773; In-sensor reservoir computing system for latent fingerprint recognition with deep ultraviolet photo-synapses and memristor array. Nat. Commun. 2022, 13: 6590; Large-Area Pixelized Optoelectronic Neuromorphic Devices with Multispectral Light-Modulated Bidirectional Synaptic Circuits. Adv. Mater. 2021, 33(45): 2105017; Realization of an Artificial Visual Nervous System using an Integrated Optoelectronic Device Array. Adv. Mater. 2021, 33(51): 2105485*)

Fig. R22 PPF index of our device. The inset image plots the time-resolved photocurrent of device at two successive light pulses with the width of 500 ms.

Fig. R23 PPF feature of previously reported optoelectronic synaptic device. **a** Schematic illustration and **b** PPF index of α -In₂Se₃ optoelectronic synapse (*Adapted from: An optoelectronic synapse based on α -In₂Se₃ with controllable temporal dynamics for multimode and multiscale reservoir computing. Nat. Electron. 2022, 5(11): 761-773).*

Fig. R24 PPF characteristic of previous reported optoelectrical synaptic device. **a** PPF behavior of the a-GaO_x photo-synapse and **b** PPF indexes varying with pulse intervals from 45 ms to 1000 ms, fitted by a double-exponential function (*Adapted from: In-sensor reservoir computing system for latent fingerprint recognition with deep ultraviolet photo-synapses and memristor array. Nat. Commun. 2022, 13: 6590).*

We have corrected the related description, and **Fig. R22** is updated as part of **Fig. 2** in main text.

“The τ_1 is an order of magnitude larger than τ_2 , indicating a successful stimuli process, consistent with previously reported optically stimulated synaptic devices.”

(24)

The transition from ~~STP to LTP~~ can be demonstrated with the increasing of incident power. As shown in **Fig. 2f**, a gradual rise of EPSC from 0.03 to 0.12 nA is identified

24. “fast-rising response to slow-rising response” The terms “STP” and “LTP” are not appropriate to be used, because the response behaviors shown here are not similar with the biological STP and LTP behaviors in the synaptic gain. In other words, since there is no signal transmission gain definable in the proposed device, the analogy does not sound right, and thus, the use of STP and LTP here can mislead the readers.

Reply: Thanks for the reviewer’s suggestions. We think the terms “fast-rising response to slow-rising response” may not be that accurate to describe the device behavior under different optical stimuli intensities, since it only describes the light response process instead of the followed relaxation process after the light stimuli are removed. The implementation of pre-processing functions of our sensor array highly relies on the nonlinear photocurrent relaxing process. The light intensity-dependent relaxing rate of photocurrent enables the implementation of noise filtering, and trace extraction is also realized by time-dependent relaxing kinetic of photocurrent.

Therefore, we think that it would be better if the terms of “STP” and “LTP” can be kept. For artificial synaptic devices with emerging device, it is routine to define signal transmission gain as residual level of postsynaptic current, and distinguishing of STP and LTP is by measuring the level of residual postsynaptic current and its lifetime (*See for example: Co-assembled perylene/graphene oxide photosensitive heterobilayer for efficient neuromorphics. Nat. Commun. 2022, 13: 4996; A bioinspired*

optoelectronically engineered artificial neurobotics device with sensorimotor functionalities. Nat. Commun. 2019, 10: 3873; Short-term plasticity and long-term potentiation mimicked in single inorganic synapses. Nat. Mater. 2011, 10: 591–595; Large-area pixelized optoelectronic neuromorphic devices with multispectral light-Modulated bidirectional synaptic circuits. Adv. Mater. 2021, 33: 2105017).

As the reply for comment #10 mentioned, the demonstration of STP and LTP are revised in our device with the repeated light pulses. **Fig. R25** shows the value of EPSC for different numbers of light pulses. The EPSC, triggered by 2 successive pulses, declines rapidly to initial stage in a short time of 20 s, which is conceptually defined as STP. In contrast, by applying 60 successive pulses with same duration and intensity, the higher gain value of EPSC is obtained, and it could keep about 30% of peak current after a long decay time of 350 s, resulting in the formation of LTP.

Fig. R25 Revised demonstration of STP and LTP for our devices under 2 and 60 illumination pulses.

As shown in **Fig.R26a**, such STP and LTP characteristics of optical synaptic device have been reported by Zhang *et al.* (*Co-assembled perylene/graphene oxide*

photosensitive heterobilayer for efficient neuromorphics. Nat. Commun. 2022, 13: 4996). STP is defined as when EPSC drops to 40 pA in a short interval of 30 s by applying 2 light pulses. While, for LTP, it takes 160 s to descend to the same current level after applying 13 light pulses. Similar definition of STP and LTP is also reported by Kwon *et al.* (*Large-area pixelized optoelectronic neuromorphic devices with multispectral light-Modulated bidirectional synaptic circuits. Adv. Mater. 2021, 33: 2105017*) (**Fig.R26b and c**).

Fig. R26 STP and LTP of previous reported device. **a** Similar STP and LTP features of optical device reported by Zhang *et al.* (*Adapted from: Co-assembled perylene/graphene oxide photosensitive heterobilayer for efficient neuromorphics. Nat. Commun. 2022, 13: 4996*). **b-c** Similar STP and LTP features of optical device reported by Kwon *et al.* (*Adapted from: Large-area pixelized optoelectronic neuromorphic devices with multispectral light-Modulated bidirectional synaptic circuits. Adv. Mater. 2021, 33: 2105017*).

Fig. R25 is updated as part of **Fig. 2** in main text, and related description is shown as below:

“As shown in Fig. 2f, after applying two successive pulses with duration of 300 ms, photocurrent declines rapidly to the initial stage within 20 s, which conceptually refers to STP. LTP is achieved by applying 60 successive pulses, and 30% of peak photocurrent can be kept even after a long decay time of 350 s.”

(25)
frequency (Supplementary Figure S10). ~~The learning efficiency and recognition accuracy in the artificial neural network are mainly determined by the asymmetry ratio (AR) and linearity of synaptic devices.~~ As shown in Fig. 2g, the calculated AR
25. This sentence should be deleted here, since in general, this sentence is not necessarily true without specifying the model type of artificial neural networks employed, and without definitions of the learning efficiency and recognition accuracy. Besides, this is an abrupt sentence.

Reply: Thanks for the reviewer’s suggestions. Corresponding sentences have been deleted.

~~ratio (AR) and linearity of synaptic devices.~~ As shown in Fig. 2g, the calculated AR
(26)
value is estimated to be 0.63, which is outstanding in previously reported synaptic
26. The definition of the asymmetry ratio (AR) is unclear. In addition, the statistical value of the AR should be described.

Reply: Thanks for the reviewer’s suggestions. The asymmetric ratio (AR) is used to characterize the linear channel conductance change between the LTP and LTD for NbS₂/MoS₂ phototransistor²⁶. It can be calculated by equation^{27,28}:

$$AR = \frac{\max|G_p(n) - G_d(n)|}{G_p(15) - G_d(15)} \text{ for } n = 1 \text{ to } 15 \quad (1 - 5)$$

where $G_p(n)$ and $G_d(n)$ is channel conductance values after applying n th light pulse and n th electrical spike, respectively.

We investigated the statistic value of AR in sensor array, and the mean values and corresponding standard deviations are calculated to be about 0.65 and 0.03, respectively.

(Fig. R27).

Fig. R27 AR value distribution of NbS₂/MoS₂ sensor array. **a** Statistical distribution of AR value of device is fitted with a logarithmic normal curve, in which the mean values and corresponding standard deviation are 0.65 and 0.03, respectively. **b** AR value of partial devices in sensor array.

Fig. R27 has been added in Supplementary Information as **Fig. S12**, and related description has been updated in main text:

“The mean value and standard deviation of AR for MoS₂/NbS₂ sensor array are calculated to be 0.65 and 0.03, respectively, (Supplementary Figure S12).”

NbS₂/MoS₂ synaptic devices can significantly decrease to 50 mV with preserving (27) satisfied synaptic plasticity (Supplementary Figure S11). Thus, the minimum power consumption is determined to be 4.2 pJ per spike under activated by 500 ms duration

27. The definition of the satisfaction is unclear, and besides, this is a subjective but not objective expression. For example, the signal-to-noise ratio between the plateau current levels before and after the light pulse exposure could be used to determine a practically acceptable operating voltage.

Reply: Thanks for the reviewer’s suggestions. We have deleted the word “satisfied” in manuscript. The ratio of photo-induced persistent photocurrent and initial current is utilized to evaluate device performance. As shown in Fig. R28, the level of persistent photocurrent is twice the initial current by applying a light pulse (0.1 mW, 10 ms) at $V_{ds}=50$ mV, and it is one order of magnitude larger than that of dark current when the pulse during time increases to 1 s.

Fig. R28 Ratio of photo-induced persistent photocurrent and initial current with different pulse duration and $V_{ds}=50$ mV.

Fig. R28 has been added in Supplementary Information as **Fig. S14**.

(28)
satisfied synaptic plasticity (**Supplementary Figure S11**). Thus, the minimum power
consumption is determined to be 4.2 pJ per spike under activated by 500 ms duration

28. energy. The Joule is the unit of energy, but not the unit of power.

Reply: Thanks for the reviewer’s suggestions. The word “power” has been revised with “energy”.

art optical synaptic devices ⁴⁰⁻⁴⁵, indicating the great potentials of NbS₂/MoS₂
phototransistor for low-power optoelectronic applications (**Supplementary Table S1**).

29. As mentioned in the review comment for the abstract, the energy consumption is not appropriate to be used for describing/comparing features of the devices, since there are external energy/signal sources, namely, the laser equipments with various wavelengths of light, the output powers, and the pulse durations. The energy transfer efficiency, instead of the consumption, could be used.

Reply: Thanks for the reviewer’s suggestions. As reviewer suggested, the External Quantum Efficiency (EQE), which refers to the ratio of the number of charge carriers collected to the number of incident photons, is also calculated to characteristic the energy/power transfer efficiency. The EQE value of device at $V_{ds}=50$ mV is obtained by the equation:

$$EQE(\%) = \frac{I_{Ph}/e}{P_{Inc}/\hbar\nu} \times 100 \quad (1 - 6)$$

where I_{Ph} , P_{Inc} , e and $\hbar\nu$ are photocurrent, incident power density of the optical pulse, unit charge and photon energy, respectively.

The calculated EQE with dependence on pulse duration is shown in **Fig. R29**. It is obvious that under the same light illumination, the higher EQE requires high-level of photocurrent, which is unfavorable to reducing electrical consumption. Therefore, devices with low energy consumption usually reveals a barely satisfactory EQE.

Fig. R29 The characteristic of EQE under applying 0.1 mW optical pulse with different duration.

(30)

~~excellent synaptic performance of~~ individual NbS₂/MoS₂ phototransistor allows the implementation of image detection and memory simultaneously. An input letter “H”

30. This should be deleted. The logic is reversed here.

Reply: Thanks for the reviewer’s suggestions. The corresponding phrase has been deleted.

(31)

in **Fig. 3a**, the image “H” is well recognized after ~~training~~ by 60 successive light pluses

31. There is no explanation about the "training" even in the Methods section.

Reply: Thanks for the reviewer’s suggestions. Training refers to that image information is repeatedly input into the sensor array 60 times with a frequency of 2.5 Hz. The detailed process of input image information is realized by an “H” letter photomask with 60 successive light pulse illumination.

Corresponding sentence has been updated in main text, and detailed implementation process has been added in method section of main text:

“The image “H” is well recognized after repeatedly inputting optical stimuli for 60 times (532 nm, 3.1 $\mu\text{W } \mu\text{m}^{-2}$, 2.5 Hz).”

“Performance measurement of the sensor array for static targets was conducted with the assistance of “H” shaped photomask. After patterning the specific letters on glass substrate coated with AZ5214 photoresist, Ti/Au (20/100 nm) were deposited by electron-beam evaporation to block the undesired area for light illumination. The active size of photomask is consistent with that of sensor array, and is can be well aligned according to the markers.” (Method section)

(32)

Note that the conductance contrast is sufficient to identify the “H” pattern in the weight map even with a long decay duration of 300 s, indicating the realization of steady image

32. *There is no definition of "the weight map" even in the Methods section.*

Reply: Thanks for the reviewer’s suggestions. The current across the channel corresponds to synaptic weight of biological counterpart, thus the weight map refers to map of photocurrent.

The related description has been added in methods section of main text:

“The weight map is obtained by normalizing photocurrent of sensors in the array, which was collected with a multichannel semiconductor analyzer integrated into the home-built testing system.”

(33)

memory. Such characteristics are consistent with the feature of human visual system
in which memory retention gradually fades with the extension of storage time.

33. The sentence is scientifically incorrect. The long-lasting (>5 minutes) memory of an image presented for the duration of 0.5 sec is not considered to be formed (at least solely) by the visual system.

Reply: Thanks for the reviewer’s suggestions. Corresponding sentence has been deleted.

The nonlinear optical response of human retina allows it to accurately distinguish the primary information from the massive noisy inputs ⁽³⁴⁾ ¹⁰. Similarly, effective modulation

34. The reference paper must be wrong.

Reply: Thanks for the reviewer’s suggestions. Corresponding reference has been updated with references as below:

“8. Gollisch, T. & Meister, M. Eye smarter than scientists believed: neural computations in circuits of the retina. Neuron 65, 150-164 (2010).

71. Endeman, D. & Kamermans, M. Cones perform a non-linear transformation on natural stimuli. J. Physiol. 588, 435-446 (2010).

72. Enroth-Cugell, C. & Robson, J. G. The contrast sensitivity of retinal ganglion cells of the cat. J Physiol 187, 517-552 (1966).”

(35)

Fig. 3b, the 100% device yield and narrow distribution of current value indicates the

35. Not only the dark current distribution, but also the current distribution under exposure with the spatially uniform light patterns should be tested. This must be also critical for the high-quality image acquisition.

Reply: Thanks for the reviewer's suggestions. We investigated the photocurrent distribution of NbS₂/MoS₂ sensor array. **Fig. R30** shows photoresponse curve of 100 pixels under 532 nm illumination with intensity of 0.1 mW. The mean values for dark current (I_{dark}) and photocurrent (I_{ph}) are calculated to be 0.14 ± 0.09 nA and 1.87 ± 0.42 nA, respectively. These results indicate the excellent uniformity of array, which is critical for acquisition of high-quality filtered images.

Fig. R30 Device-to-device variation in the characteristics of NbS₂/MoS₂ phototransistors array.

a Light-induced current curve for 100 NbS₂/MoS₂ transistors of the device. **b-c** Dark current (I_{dark}) and photocurrent (I_{ph}) statistical distribution, in which the mean values and corresponding standard deviation for I_{dark} and I_{ph} are found to be 0.14 nA, 0.09 nA and 1.87 nA, 0.42 nA, respectively.

Fig. R30 has been added as part of **Fig.S15** in Supplementary Information, and related description has been added in main text:

“The device-to-device variation was first statistically evaluated (Supplementary Figure S15). The 100% device yield and excellent uniformity of performance metrics are crucial for the acquisition of high-quality images.”

(36)

Therefore, a pre-processed image is obtained by mapping the measured photocurrent of each pixel. One can see that the target letter “H” is clearly filtered with high image
36. The algorithm/method/operation of the preprocessing is not described. and hence, not be able to be understood by anyone. The authors must clearly explain the preprocessing by the chip. Besides, the "noisy letter" was intentionally added as the low-power laser light pattern. Therefore, the letter "I" in Fig. 3c and other overlapping letters in Fig. 3e-right were not noise patterns but the input patterns.

Reply: Thanks for the reviewer’s suggestions. We updated the experiment design to better explain the pre-processing functions of target image information extraction *via* reducing the noisy information interference based on our sensor array. The letter “I” in **Fig. 3c** and dark overlapping letters in right panel of **Fig. 3e**, which are input into sensor array together with informative letter, can be regarded as noisy information to interfere image recognition process.

The details are shown as below and updated in main text (page 11 and method section): The photocurrent in device decays faster as stimulated by light pulse with lower light intensity. After 100 successive pulse training, the differences of remaining current as well as contrast are further enlarged (**Fig. R31 a**). Therefore, noise filtering for static image is demonstrated in our device. We first used an optical mask to project

target letter on the sensor array with light intensity of 1 mW, and recorded the corresponding current of each pixel before erasing. Photocurrent of noisy letter with light intensity of 0.5 mW was also obtained using the same procedures. The collected data of two parts were finally gathered and map-plotted. The left panel of **Fig. R31 b** shows the light intensity mapping of input image, including a target letter “H” with 1 mW and a faint noisy letter “T” with 0.5 mW. The image information is repeatedly input into the sensor array for 100 times. After decaying for 150 s, remaining currents of pixels are collected and map-plotted (right panel of **Fig. R31 b**). Since current triggered by lower light intensity decay faster, whereas current triggered by higher light intensity decay slower. The corresponding average values of pixels in target letter and noisy letter are normalized as 1 and 0.22, respectively. Compared with light intensity contrast of input image, the decayed current contrast between pixels with different light illumination is enlarged from 1:0.5 to 1:0.22 without using any external circuits, just like when people focus on a specific object, the rest of the surrounding information will be mostly filtered.

Fig. R31 Revised filtering processing of device. **a** decay behavior of light-induced current and contrast. **b** Light intensity mapping of input image (left) and current mapping of pre-processed image (right).

Fig. R31 is updated as part of **Fig. 3** in main text, and related description is revised as below:

“Fig. 3b shows the decay characteristics of device as trained by 100 successive light pulses with light intensity of 3.1 and 1.6 $\mu\text{W } \mu\text{m}^{-2}$, respectively. One can see that the decay rate of the photocurrent at 3.1 $\mu\text{W } \mu\text{m}^{-2}$ is much slower than that at 1.6 $\mu\text{W } \mu\text{m}^{-2}$, and a considerably increased ratio of remaining photocurrent is estimated to be 4.5 after decaying for 150 s. The light-tunable plasticity above enables the contrast enhancement of input image, allowing the sensor array to extract target image from messy information for noise filtering. Correspondingly, Fig. 3c illustrates the grey scale of pixels for input images with a main letter “H” stimulated at 3.1 $\mu\text{W } \mu\text{m}^{-2}$ and a noisy letter “T” at 1.6 $\mu\text{W } \mu\text{m}^{-2}$. It is clear that the pre-processed image after repeated training demonstrates an enlarged difference between the grey scale of pixels over the input images, thus contributing to an output image with enhanced contrast. The results suggested the noise filtering function can be hardware implemented base on our NbS₂/MoS₂ sensor array without using any external circuits.”

“To demonstrate the function of noise filtering, we divided input image into two parts including target letter and noisy letter, which were projected on the sensor array separately. After trained by input optical stimuli for 100 times, the decaying current data of two parts are collected and map-plotted into the same image.” (Method section)

(37)

recognition. As shown in **Fig. 3e**, the filtering preprocessing evidently reduces the noises and improve the image quality, and contributes to the enhanced contrast between

37. It is not believed that each of these images is composed by the 10-by-10 pixels.

Reply: Thanks for the reviewer’s suggestions. The input and output images in **Fig. 3e** of main text are composed by 28-by-28 pixels. We choose handwritten letters image from EMNIST database (EMNIST: an extension of MNIST to handwritten letters. <https://arxiv.org/abs/1702.05373>) as static recognition object, in which the size of images is 28×28 pixels for preserving the details of handwritten letters.

The related description in main text has been corrected:

“Each image with 28×28 pixels consists of a bright informative letter, a dark interference letter, and background white noise.”

biological retina exhibits integrated sensing and spatial temporal differentiation functions^{47,48}, thus allowing the simplification of data processing step **(38)** **(Fig. 4a)**. As for

38. The illustration in Fig. 4a is scientifically incorrect, and thus, should be deleted not to mislead the readers.

Reply: Thanks for the reviewer’s suggestions. We are sorry for the scientifically incorrect illustration in **Fig. 4a**. We replaced **Fig. 4a** with comparison between our device and traditional CMOS platform for on-chip motion detection. As shown in **Fig. R32**, traditional sensor captures image frame-by-frame with a specific clock frequency, and massive images are arranged by time sequence to form the motion trace with the assistance of additional processing units and memory modules, which unavoidably increases the complexity of entire vision chip, latency and power consumption. In contrast, the decay kinetics of conductance state in individual NbS₂/MoS₂ transistor allows array to direct process and generates the spatial-temporal current information

according to the stimuli sequences. The photocurrent of each pixel triggered earlier is lower than that of the pixel triggered later at the same measured time point. Thus, the motion trace (*i.e.*, trace shape and motion direction) can be derived by directly reading the current of device. Moreover, each pixel located within the trajectory would spontaneously respond to the stimuli and parallelly real-time memorize the spatial-temporal dependent information. As a result, a complete motion trace could be obtained by only executing once measurement after the moving trajectory finished. The process allows us to potentially avoid the massive data storage, redundancy and latency caused by repeatedly reading.

Fig. R32 Comparison between frame-based camera and NbS₂/MoS₂ sensor for on-chip trace extraction.

Fig. R32 is updated as part of **Fig. 4** in main text, and related description is shown as below:

“As shown in Fig. 4a, front-end motion detection is generally realized based on CMOS vision chip, which captures visual information framed-by-frame at a predetermined rate. The separated memory and processing of frame data at sensor-level or pixel level would cause the unnecessarily inflating data.”

“The slow kinetics of time-dependent decay photocurrent allows to directly process and generate the spatiotemporal information within sensor, which can be parallelly real-time memorized in each pixel. It would remarkably advance the efficiency of data processing and contribute to the system simplification.”

(39)

Trajectory detection task for motion objects was demonstrated in our retinomorphie hardware platform. **Fig. 4b** presents the schematic diagram of sensor array, which **39.** *It is hardly acceptable that the step-by-step illumination of the laser light pulses is defined as an object motion with a trajectory; the laser light spot in the dark background is not similar to an object in the visual field in "real scenarios"; the illumination of the light pulse with a duration of >0.5 sec every 2 seconds cannot represent a motion in "real scenarios".*

Reply: Thanks for the reviewer’s suggestions. We apologize for the misleading description in original manuscript, and corresponding experiment and explanation have been updated. A continuous laser moving with uniform speed along the predefine trace is utilized to simulate the movement of target. As shown in **Fig. R33a**, the laser spot moves without any stop with the assistance of piezoelectric displacement actuator. The moving time between adjacent pixels is about 2.5 s, and the effective illuminating time of each pixel is about 500 ms. The detection process is schematically illustrated as **Fig. R33b-d**. Due to the kinetics of photocurrent and memory effect, the later triggered pixel exhibits a higher level of photocurrent, and such spatial-temporal dependent photocurrent of pixels is consecutive during a relatively long time. In the revised

manuscript, we select 10 time points for measurement to dynamically track the trajectory change. The acquisition of whole trace actually only requires one measurement after the moving process finished, which allows to potentially avoid the massive data storage, redundancy and latency caused by repeated reading.

The simple structure and autonomous operation are distinguishing features of our device. Each pixel of array only contains a NbS₂/MoS₂ phototransistor, and it will spontaneously respond to the stimuli and parallelly real-time memorize spatial-temporal information, which allows NbS₂/MoS₂ sensor array to realize the in-sensor pre-processing functionality. The information consecutively varies with time even after stimuli finished, and the amplitudes of photocurrent relies on the stimulation order.

Fig. R33 Schematic illustration of trace detection. **a** Schematic illustration of laser spot moving. **b** The pre-defined trace. **c** Time-depend decay current of each pixel at trace. **d** Current mapping of array after moving.

We agree with the reviewer that the experiment condition is not in well accordance with real scenarios because of the ideal background, thus “real scenarios” has been

deleted in manuscript. Indeed, motion detection for smart home systems has been reported by using visible light communication (VLC) technology²⁹⁻³³. As shown in **Fig. R34**, a uniform highlight LED array is utilized as a transmitter, and a camera will capture the motions and data of finger by detecting light reflected on finger. The background of this case is very similar to our demonstration, because ambient illuminance is significantly weaker as opposed to the illuminance of the transmitter³¹. Compare with the motion detection approach in VLC, which relies on motion detection algorithm to extract and recognize the trace, our NbS₂/MoS₂ sensor array could perform trajectory extraction in sensor terminals. It could significantly reduce the burden on center processor and complexity of algorithm.

Fig. R34 Visible light communication system with motion detection (*Adapted from: High-accuracy scheme based on a look-up table for motion detection in an optical camera communication system. Opt. Express 2022, 28: 10270-10279*).

The related description has been added in main text

“A continuous laser scanning along the predefined trace is utilized to simulate the moving trajectory. The laser spot moves without any stop with the assistance of a piezoelectric displacement actuator. The moving time between adjacent pixels is about

2.5 s, and the effective illuminating time of each pixel is about 500 ms.” (Method section)

(40)

the corresponding input trajectories from dataset. ~~As illustrated in Fig. 4e, the detected accuracies of all 10 group are all above 92% after algorithmic correction, indicating~~

40. Fig. 4e and the related sentences should be deleted, since the result is too obvious even without employing an external detection method. The number "92 %" does not provide any information without a quantitative comparison with others.

Reply: Thank you for your review of our manuscript. The purpose of **Fig. 4e** and the related sentences is to quantitatively demonstrate the robustness and generalizability of the frameless paradigm with the consideration of device variance. By including device variance in our simulation, we aim to show that our vision chip can accurately detect moving targets even with some level of variance. The pre-processing characteristic of sensor array enables the trace extraction conducted in sensor terminals, and development of corresponding image recognition algorithm and datasets is required to quantitatively evaluate the detection accuracy for moving trajectory.

The stimulated trajectory detection process is shown as follows: to enhance the robustness and generalizability of the method being tested, 10 datasets, each containing 40 unique trajectories, were first generated using a random trajectory generating program in the Python programming language. The starting point, ending point, and moving direction of the trajectory are all random, ensuring that the dataset is representative of a large population of possible trajectories. These datasets were then

employed to simulate the output current of the array over the entire duration of each trajectory with consideration of device variance. We next restored each trajectory from the simulated array output current (**Fig. R35**) and compared it to the actual one in the dataset. To determine the order in which the trajectory pixels were passed through in the restored trajectories, we compared the average output photocurrent of each device in the final 5 seconds. The accuracy of the trajectory detection on each dataset can be finally calculated as the ratio of correctly restored trajectories to the total number of trajectories in the dataset. By evaluating the method's performance on 10 datasets, we can infer that the method produces consistent results and it is not specific to a single dataset.

Fig. R35 Output result of simulation.

To further demonstrate the high recognition accuracy of our system, we generated a large dataset containing 400 randomly generated trajectories, in which the shape of trajectories is different from that of the existing 10 datasets (**Fig. R36**). A high detection accuracy of 91% was achieved based on this new large dataset, which strongly confirms the robustness and reliability of our NbS₂/MoS₂ sensor array in detecting arbitrary trajectories.

Fig. R36 Example trajectories in 400-sized trajectory datasets that are correctly detected.

We appreciate reviewer's suggestion about including a comparison with other datasets. However, to the best of our knowledge, this standard dataset for this application scenario has not been generated and used previously, making a rigorous comparison difficult. We acknowledge that there have been some works with similar scenarios, but the adopted databases have different parameters, such as the background size and the randomness of the paths, that affect the validity of the comparison. As such, we believe that presenting the accuracy of our frameless paradigm in detecting

spontaneous responses in an array still provides valuable information for the readers on the robustness and generalizability of the frameless paradigm despite lack of rigorous comparison.

However, considering the potential applications of our device, motion detection in optical camera communications (OCC) might be selected for rough comparison due to similar possible operation scene in smart device control ²⁹⁻³³. In typical OCC scheme, optical camera is employed to record the image data of gestures in each frame through detecting light reflected on finger ³⁰. An efficient motion detection algorithm is highly required for processing and difference quantization to extract and recognize the finger trace. Although above 90% accuracy is achieved in existing scheme, a reliable and accurate detection of motion depends largely on the achievable frame rate and optimization of algorithm ^{29,30}. The trajectory detecting simulation in manuscript has proved that our NbS₂/MoS₂ sensor array could reach comparable accuracy in similar operation scene. Moreover, the pre-processing characteristics of NbS₂/MoS₂ sensor array enable trajectory extraction in sensor terminal, which could significantly reduce the burden on center processor and complexity of algorithm.

Therefore, it is valuable to quantify this accuracy and present it in the paper as it provides a quantitative evaluation of the performance of our NbS₂/MoS₂ array in trajectory detection. **Fig. 4e** in main text and the related discussion allow readers to better understand the method used to determine the trajectory detection accuracy and the specific accuracy for each adopted dataset.

(41)

the corresponding input trajectories from dataset. ~~As illustrated in Fig. 4e, the detected accuracies of all 10 group are all above 92% after algorithmic correction, indicating~~

41. There is no explanation about the method of the "detection".

Reply: Thanks for the reviewer's comment. We apologize that the description of this procedure was not clear in original manuscript. The detail description regarding the complete trajectory detection process is supplied as follow:

The complete workflow of trajectory detection experiment is illustrated in Fig. R37. To generate the trajectory datasets, we developed a random trajectory-generating program in the Python programming language that generates trajectories with random starting points, random ending points, and a fixed length of 10 on a 10×10 background. The total 10 datasets, each containing 40 unique trajectories, are generated with this random trajectory-generating program.

Fig. R37 Workflow of trajectory detection.

To simulate the output current of the array for the entire duration of each trajectory in the dataset, we first fit the device photocurrent under 1mW illumination with experimental data in the Python environment (**Fig. R38**). Then, we used this fitted behavior to simulate the output of each device in the array over the entire duration of each trajectory in the dataset. The output current of the devices in non-trajectory pixels remained at the dark current level, while the output current of the devices in trajectory pixels has the expected decay behavior due to light stimulation. The starting time of the photocurrent decay for each trajectory pixel device was determined through the order in which the devices were passed through. Devices that were passed through earlier have longer photocurrent decay time. The photocurrent variance of the devices was also accounted for in the simulation. We extracted the variance of the peak photocurrents and dark currents from the decay behavior of NbS₂/MoS₂ array devices and used the average of these variances as the variance of the photocurrent during the decay process. We also added Gaussian white noise generated from this calculated variance to the simulated array output current at every timepoint to emulate the effect of variance on the array output current.

Fig. R38 Fitted curve of NbS₂/MoS₂ device current under 1mW illumination.

To determine the NbS₂/MoS₂ array trajectory detection accuracy, we restore each trajectory from simulated array output current and compare the restored trajectory to the actual one in trajectory dataset. To identify the order in which the trajectory pixels were passed through in the restored trajectories, we compared the average output photocurrent of each device in the final 5 seconds. The device with larger averaged output photocurrent was passed earlier. We used the averaged current over the final 5 seconds rather than the current at a single timepoint to reduce the effect of variance on the light current value and improve the accuracy of trajectory recognition by the array. The array's trajectory detection accuracy on each dataset was calculated as follows:

$$Accuracy = \frac{\text{Correctly restored trajectory number}}{\text{trajectory number in dataset}} \quad (1 - 7)$$

Figs. R37-R38 have been added in Supplementary Information as **Figs. S21-S22**, and related description of detection process have also been added in Supplementary Information:

“Simulation for trajectory detection

*The complete workflow of trajectory detection is illustrated in **Figure S21**. To generate the trajectory datasets, we developed a random trajectory generating program in the Python programming language that generates trajectories with random starting point, random ending point, and fixed length of 10 on a 10 ×10 background. 10 datasets, each containing 40 unique trajectories, are generated with this random trajectory generating program.*

*To simulate the output current of the array for the entire duration of each trajectory in the dataset, we first fit the device photocurrent under 3.1 μW μm⁻² illumination with experimental data in the Python environment (**Figure S22**). Then, we used this fitted behavior to simulate the output of each device in the array over the entire duration of each trajectory in the dataset. The output current of the devices in*

non-trajectory pixels remained at the dark current level, while the output current of the devices in trajectory pixels has the expected decay behavior due to light stimulation. The starting time of the photocurrent decay for each trajectory pixel device was determined through the order in which the devices were passed through. Devices that were passed through earlier have a longer photocurrent decay time.

The photocurrent variance of the devices was also accounted in the simulation. We first extracted the variance of the peak photocurrent and dark current from the decay characteristics of NbS₂/MoS₂ array devices. Then, the measured variance values are utilized as a part of simulation variance for decay process. Gaussian white noise generated from this calculated variance was added into the simulated array output current at every timepoint as another part of simulation variance.

To determine the NbS₂/MoS₂ array trajectory detection accuracy, we restored each trajectory from simulated array output current and compared the restored trajectory to the actual one in trajectory dataset. To determine the order in which the trajectory pixels were passed through in the restored trajectories, we compared the average output photocurrent of each device in the final 5 seconds. The device with larger average output photocurrent was passed earlier. We used the average current over the final 5 seconds rather than the current at a single timepoint for the purpose of reducing the effect of variance on the light current value and improve the accuracy of trajectory recognition by the array. The array's trajectory detection accuracy on each dataset was calculated as follows:

$$Accuracy = \frac{\text{Correctly restored trajectory number}}{\text{trajectory number in dataset}} \quad (1)$$

”

(42)

~~NbS₂/MoS₂ hybrid films. The device can closely emulate the photo-induced synaptic plasticity of retina neurons including STP, LTP, and transition between them. The~~

42. This sentence should be deleted. There is little scientific evidence that demonstrates synaptic plasticities including STP, LTP and transition between them in the biological retinas in the literature. Please indicate appropriate reference papers if ever exist.

Reply: Thanks for the reviewer's suggestions. This sentence has been deleted.

Reviewer #2 (Remarks to the Author):

The paper reports experimental results of a fabricated 10×10 array of phototransistor devices based on NbS₂/MoS₂ hybrid films. These devices exhibit retention properties of the illumination and the retention can be erased/modulated by applying voltage pulses on the device gate. Authors claim that these devices emulate biological retina and have very singular properties for filtering and motion detection.

The main shortcomings of the paper are:

1. Authors compare their device with standard CMOS vision sensors claiming that (page 3) “state of the art digital vision systems are developed based on CMOS platform consists of separated image sensor for receiving visual inputs, memories for storing information and processors for performing neuromorphic computation”. However, although this is a frequent approach, there are many examples of smart vision sensors published in the literature where photosensor devices are monolithically integrated with front-end retinomorphic filtering and processing both in the analog and digital domain. See just a recent example: Wu, Jiangchao, et al. "A Multi-Mode CMOS Vision Sensor With On-Chip Motion Direction Detection and Simultaneous Energy Harvesting Capabilities." IEEE Sensors Journal (2022). Or Posch, Christoph, et al. "Retinomorphic event-based vision sensors: bioinspired cameras with spiking output." Proceedings of the IEEE 102.10 (2014): 1470-1484. In page 4, the authors claim “the trajectory tracing of dynamic objects was experimentally demonstrated in retinomorphic hardware systems for the first time”. This is a very strong claim which is not true, as many integrated CMOS vision arrays with motion detection have been

published. See just one example. A. Simoni, G. Torelli, F. Maloberti, A. Sartori, S. E. Plevridis and A. N. Birbas, "A single-chip optical sensor with analog memory for motion detection," in IEEE Journal of Solid-State Circuits, vol. 30, no. 7, pp. 800-806, July 1995, doi: 10.1109/4.391119. So the claim should be rephrased and properly put into context.

Reply: Thanks very much for the reviewer's comments on values of our works and providing latest articles concerning motion detection of conventional frame-based sensor and dynamic vision sensor (DVS). We rephrased the corresponding claims and gave detailed explanation. These bioinspired applications based on Si CMOS technology are realized at sensor terminal, where separated memory and processing units still exist, resulting in limited integration. Our NbS₂/MoS₂ sensor array integrates the function of sensing, memory and processing into same device. In the traditional frame-based vision system, visual information is captured framed-by-frame at a predetermined rate, which is independent of dynamic target in the scene. This acquisition method leads to unnecessarily inflating data rate and volume. The realization of front-end motion detection in frame-based device still relies on separated memory and processing of frame data at sensor-level or pixel level ^{34,35}. Moreover, additional on-chip module integration unavoidably increases the complexity of entire vision chip, resulting in potentially latency and power consumption increase.

DVS is a very excellent vision sensor based on biological principles, which emulates a simplified structure and encoding format of retina at pixel level. Asynchronous output in the form of an event stream with address-event representation

(AER) enable DVS exhibits advantages in redundancy suppression and data compression. At pixel level, external differencing circuits convert analogue sensory signals into asynchronous spikes events *via* spike coding processing³⁶⁻³⁸. Such architecture is typical near-sensor computing paradigm, where tasks including data generation, collection and computation are performed close to the sensory component¹⁶.

It should be mentioned that all these CMOS vision sensors for motion detection suffer from the shortcomings of circuit complexity, distortion of analog-to-digital conversion and poor compatibility with back-end processing platform^{39,40}. In contrast, the distinguishing feature of our sensor array is simple and compact, which only contains one individual NbS₂/MoS₂ phototransistor in each pixel. The slow decay kinetics of conductance state in individual NbS₂/MoS₂ phototransistor allows the array to direct process and generates the spatial-temporal current information. The information consecutively varies with time, and the amplitudes is dependent on the stimulation order. Each pixel located within trajectory would spontaneously respond to the stimuli and parallelly real-time memorizes and output spatial-temporal dependent photocurrent during a relatively long time. In the revised manuscript, we select 10 time points for measurement to dynamically track the trajectory change. The acquirement of whole trace only requires one measurement after the moving process finished, which could effectively avoid the redundancy and latency.

The discussion of references mentioned have been added in introduction and results part of main text. We also cited the related references reviewer mentioned. Related description is shown as below:

“Although retina-inspired vision chips based on silicon CMOS technology have been proposed, the applications still suffer from the shortcomings of circuit complexity, distortion of analog-to-digital conversion and poor compatibility with back-end processing platform. Therefore, the hardware implementation of neuromorphic image sensors, which are highly integrated with visual information sensing, memory, and pre-processing functions, are urgent to advance the artificial vision system.”

*“As shown in **Fig. 4a**, front-end motion detection is generally realized based on CMOS vision chip, which captures visual information framed-by-frame at a predetermined rate. The separated memory and processing of frame data at sensor-level or pixel level would cause the unnecessarily inflating data.”*

“Dynamic vision sensor (DVS) was also developed based on CMOS platform, which can asynchronously output in the form of an event stream with address-event representation (AER). Although DVS can be applied for real-time and high temporal resolved motion detecting, obtained raw data or event stream still need to be successively transmitted to external memory unit for post-processing, thus posing challenges for the hardware simplification of vision system.”

“In contrast, according to the difference of electrical output in spatiotemporal dimension, the NbS₂/MoS₂ sensor array can derive the pre-processed information

regarding to the relationship between the object motion position and timeline. The slow kinetics of time-dependent decay photocurrent allows to directly process and generate the spatiotemporal information within sensor; which can be parallelly real-time memorized in each pixel. It would remarkably advance the efficiency of data processing and contribute to the system simplification.”

“17. Posch, C., Serrano-Gotarredona, T., Linares-Barranco, B. & Delbruck, T. Retinomorph Event-Based Vision Sensors: Bioinspired Cameras With Spiking Output. Proc. IEEE Inst. Electr. Electron. Eng. 102, 1470-1484 (2014).

74. Wu, J. et al. A Multimode CMOS Vision Sensor With On-Chip Motion Direction Detection and Simultaneous Energy Harvesting Capabilities. IEEE Sens. J. 22, 12808-12819 (2022).

75. Simoni, A. et al. A single-chip optical sensor with analog memory for motion detection. IEEE J. Solid-State Circuits 30, 800-806 (1995).”

We also rephrased the claim in main text as below:

“the trajectory detection and real-time memorization of dynamic objects was also experimentally demonstrated.”

2. In page 12, the authors state “In conventional vision systems, the motion of objects is captured frame-by-frame based on CMOS image sensors platform. Although this is true, there are other neuromorphic CMOS vision sensors computing asynchronously

temporal contrast at the pixel level. They are the so-called dynamic vision sensors which are really the state of the art in CMOS for object motion detection based on vision. This technology should be referenced and used as baseline comparison.

Reply: Thanks for the reviewer's suggestions. The references about DVS have been added in manuscript (references 11-14), and the comparison of NbS₂/MoS₂ sensor array with DVS is given in **Table R1**. DVS is a very excellent vision sensor based on biological principles, which emulates a simplified structure and encoding format of retina at pixel level. Asynchronous output in the form of an event stream with address-event representation (AER) enable DVS exhibits advantages in redundancy suppression and data compression. At pixel level, external differencing circuits convert analogue sensory signals into asynchronous spikes events *via* spike coding processing³⁶⁻³⁸. The shortcomings of DVS includes complex circuits and large pixel area, and the data still need to be transmitted to external memory unit for post-processing due to lacking of memory function.

In contrast, the distinguishing feature of our sensor array is simple and compact, which only contains one individual NbS₂/MoS₂ phototransistor in each pixel. The slow decay kinetics of conductance state in device allows to direct process and generate the spatial-temporal current information, which can be parallelly real-time memorized in each pixel. Moreover, the simplified structure of sensor potentially decreases chip area and power consumption. The power consumption of our sensor array is five order of magnitude smaller than that of DVS.

Table R1 Comparison of NbS₂/MoS₂ sensor array with DVS.

	This work	Delbruck et al. ³⁶	Boahen et al. ⁴¹	Barranco et al. ⁴²
Operation form	Analog spatial-temporal current	Event-induced spike signals		
Pixel complexity	1 phototransistor	26 transistors, 3 caps, 1 photodiode	38 transistors, 1 photodiode	> 15 transistors, 1 photodiode
Chip area	1×0.6 mm²	6×6 mm ²	3.5×3.3 mm ²	5.5×5.6 mm ²
Power consumption	4.2 nW/Chip 42 pW/Pixel	24 mW/Chip 1.5 μW/Pixel	62.7 mW/Chip 10.8 μW/Pixel	132 mW/Chip 8 μW/Pixel
Application	Static and dynamic scenes	Dynamic scenes		

The references about DVS are added in manuscript (reference 11-14), **Table R1** has been added in Supplementary Information as **Table S2**, and related description has been updated in main text:

“Dynamic vision sensor (DVS) was also developed based on CMOS platform, which can asynchronously output in the form of an event stream with address-event representation (AER). Although DVS can be applied for real-time and high temporal resolved motion detecting, obtained raw data or event stream still need to be successively transmitted to external memory unit for processing, thus posing challenges for the hardware simplification of vision system.”

“In contrast, according to the difference of electrical output in spatiotemporal dimension, the NbS₂/MoS₂ sensor array can derive the pre-processed information regarding to the relationship between the object motion position and timeline. The slow kinetics of time-dependent decay photocurrent allows to directly process and generate the spatiotemporal information within sensor, which can be parallelly real-time memorized in each pixel. It would remarkably advance the efficiency of data processing and contribute to the system simplification.”

“11. Culurciello, E., Etienne-Cummings, R. & Boahen, K. A. A biomorphic digital image sensor. IEEE J. Solid-State Circuits 38, 281-294 (2003).

12. Lichtsteiner, P., Posch, C. & Delbruck, T. A 128X128 120 dB 15 ms Latency Asynchronous Temporal Contrast Vision Sensor. IEEE J. Solid-State Circuits 43, 566-576 (2008).

13. Lenero-Bardallo, J. A., Serrano-Gotarredona, T. & Linares-Barranco, B. A 3.6 μs Latency Asynchronous Frame-Free Event-Driven Dynamic-Vision-Sensor. IEEE J. Solid-State Circuits 46, 1443-1455 (2011).

14. Zaghoul, K. A. & Boahen, K. Optic nerve signals in a neuromorphic chip II: Testing and results. IEEE Trans. Biomed. Eng. 51, 667-675 (2004).”

3. In order to test the motion detection capabilities of the sensor a custom database of rocket trajectories is used and classified into ten groups obtaining a 92% accuracy in the classification. However, the accuracy is completely dependent on the complexity of the task. In order to evaluate the performance, it should be better explained how the

dataset is acquired and how the 10 classes are determined. Furthermore, a comparison with an already existing dataset would be required for benchmarking.

Reply: Thank you for your review of our manuscript. The purpose of **Fig. 4e-d** and related sentence is to quantitatively demonstrate the robustness and generalizability of frameless paradigm with spontaneous response in array. The pre-processing characteristic of sensor array enables the trace extraction conducted in sensor terminals, and development of corresponding image recognition algorithm and datasets is required to quantitatively evaluate the detection accuracy for moving trajectory.

The complete workflow of trajectory detection experiment is illustrated in **Fig. R39**. To enhance the robustness and generalizability of the method being tested, 10 datasets, each containing 40 unique trajectories, were first generated using a random trajectory generating program in the Python programming language. The starting point, ending point, and moving direction of the trajectory are all random, ensuring that the dataset is representative of a large population of possible trajectories.

Fig. R39 Workflow of trajectory detection.

These datasets were then employed to simulate the output current of the array over the entire duration of each trajectory with consideration of device variance. We first fit the device photocurrent under 1mW illumination with experimental data in the Python environment (**Fig. R40**). Then, we used this fitted behavior to simulate the output of each device in the array over the entire duration of each trajectory in the dataset. The output current of the devices in non-trajectory pixels remained at the dark current level, while the output current of the devices in trajectory pixels has the expected decay behavior due to light stimulation. The starting time of the photocurrent decay for each trajectory pixel device was determined through the order in which the devices were passed through. Devices that were passed through earlier have longer photocurrent decay time. The photocurrent variance of the devices was also accounted for in the

simulation. We extracted the variance of the peak photocurrents and dark currents from the decay behavior of NbS₂/MoS₂ array devices and used the average of these variances as the variance of the photocurrent during the decay process. We also added Gaussian white noise generated from this calculated variance to the simulated array output current at every timepoint to emulate the effect of variance on the array output current.

Fig. R40 Fitted curve of NbS₂/MoS₂ device current under 1mW illumination.

We next restored each trajectory from the simulated array output current (**Fig. R41**) and compared it to the actual one in the dataset. To determine the order in which the trajectory pixels were passed through in the restored trajectories, we compared the average output photocurrent of each device in the final 5 seconds. We used the averaged current over the final 5 seconds rather than the current at a single timepoint to reduce the effect of variance on the light current value and improve the accuracy of trajectory recognition by the array. The accuracy of the trajectory detection on each dataset can be finally calculated as the ratio of correctly restored trajectories to the total number of trajectories in the dataset. By evaluating the array's performance on 10 individual datasets, we can infer that the method produces consistent results and it is not specific

to a single dataset. To determine the accuracy of each dataset, we compare the restored trajectory to the actual one in the trajectory dataset and calculate the accuracy as follows:

$$Accuracy = \frac{\text{Correctly restored trajectory number}}{\text{trajectory number in dataset}} \quad (2 - 1)$$

Fig. R41 Output result of simulation.

As for the complexity of the task, the random trajectory-generating program used to generate the datasets can provide representative trajectories that the sensor may encounter in real-world scenarios. We also generated another large dataset with 400 non-repeating trajectories to further evaluate the sensor's performance (**Fig. R42**). The array was able to achieve an accuracy of 91% based on this new dataset, further demonstrating the robustness and generalizability of the sensor's motion detection

capabilities. Therefore, the high accuracy achieved based on these 10 individual datasets can be considered as a good indicator of the sensor's performance.

Fig. R42 Example trajectories in 400-sized trajectory datasets that are correctly detected.

We appreciate reviewer's suggestion about including a comparison with other datasets. However, to the best of our knowledge, this standard dataset for this application scenario has not been generated and used previously, making a rigorous comparison difficult. We acknowledge that there have been some works with similar scenarios, but the adopted databases have different parameters, such as the background size and the randomness of the paths, that affect the validity of the comparison. As such, we believe that presenting the accuracy of our frameless paradigm in detecting spontaneous responses in an array still provides valuable information for the readers on the robustness and generalizability of the frameless paradigm despite lack of rigorous comparison.

Considering the potential applications of our device, motion detection in optical camera communications (OCC) might be selected for rough comparison due to similar

possible operation scene in smart device control ²⁹⁻³³. In typical OCC scheme, optical camera is employed to record the image data of gestures in each frame through detecting light reflected on finger ³⁰. An efficient motion detection algorithm is highly required for processing and difference quantization to extract and recognize the finger trace. Although 90% accuracy is achieved in existing scheme, a reliable and accurate detection of motion depends largely on the achievable frame rate and optimization of algorithm ^{29,30}. The trajectory detecting simulation in manuscript has proved that our NbS₂/MoS₂ sensor array could reach comparable accuracy in similar operation scene. Moreover, the pre-processing characteristics of NbS₂/MoS₂ sensor array enable trajectory extraction in sensor terminal without information loss, which could significantly reduce the burden on center processor and complexity of algorithm.

Figs. R39-R40 have been added in Supplementary Information as **Figs. S21-S22**, and related description for detailed simulation procedure have also been added in Supplementary Information:

“Simulation for trajectory detection

*The complete workflow of trajectory detection is illustrated in **Figure S21**. To generate the trajectory datasets, we developed a random trajectory generating program in the Python programming language that generates trajectories with random starting point, random ending point, and fixed length of 10 on a 10 ×10 background. 10 datasets, each containing 40 unique trajectories, are generated with this random trajectory generating program.*

*To simulate the output current of the array for the entire duration of each trajectory in the dataset, we first fit the device photocurrent under 3.1 μW μm⁻² illumination with experimental data in the Python environment (**Figure S22**). Then, we*

used this fitted behavior to simulate the output of each device in the array over the entire duration of each trajectory in the dataset. The output current of the devices in non-trajectory pixels remained at the dark current level, while the output current of the devices in trajectory pixels has the expected decay behavior due to light stimulation. The starting time of the photocurrent decay for each trajectory pixel device was determined through the order in which the devices were passed through. Devices that were passed through earlier have a longer photocurrent decay time.

The photocurrent variance of the devices was also accounted in the simulation. We first extracted the variance of the peak photocurrent and dark current from the decay characteristics of NbS₂/MoS₂ array devices. Then, the measured variance values are utilized as a part of simulation variance for decay process. Gaussian white noise generated from this calculated variance was added into the simulated array output current at every timepoint as another part of simulation variance.

To determine the NbS₂/MoS₂ array trajectory detection accuracy, we restored each trajectory from simulated array output current and compared the restored trajectory to the actual one in trajectory dataset. To determine the order in which the trajectory pixels were passed through in the restored trajectories, we compared the average output photocurrent of each device in the final 5 seconds. The device with larger average output photocurrent was passed earlier. We used the average current over the final 5 seconds rather than the current at a single timepoint for the purpose of reducing the effect of variance on the light current value and improve the accuracy of trajectory recognition by the array. The array's trajectory detection accuracy on each dataset was calculated as follows:

$$\text{Accuracy} = \frac{\text{Correctly restored trajectory number}}{\text{trajectory number in dataset}} \quad (1)$$

”

Reviewer #3 (Remarks to the Author):

The manuscript of “Retinomorphic vision chip for high-accuracy in-sensor static and dynamic object recognition” written by Pei-Yu Huang et al, implemented retinomorphic sensor based on NbS₂/MoS₂, featuring remarkable photo-induced synaptic plasticity. The detection and recognition of moving object trajectory is impressive. The manuscript needs however major revision. After addressing the questions below, the manuscript may be considered for publication in this journal.

1. The authors need to show and describe “home-built testing system”. Figure 1d is not “home-built testing system”. (Figure 1d is photograph of the array and schematic of the phototransistor)

Reply: Thanks very much for the reviewer’s comments on the values of our works. The photograph of the home-built testing system is provided in **Fig. R43**. The NbS₂/MoS₂ phototransistors chip is directly bonded and integrated into PCB board, and an automatic testing system with multiple channels collects the current *via* contact pins around the board. Then, the collected data is processed by customized software and shown on screen in form of a graph. As reviewer suggested, we supplied the photograph of the home-built testing system in Supplementary Information as **Fig. S1**.

Fig. R43 Home-built testing system. **a** Digital photograph of home-built testing system. **b** “H”-shaped photomask and corresponding testing result. **c** PCB board with 100 contact pins.

Fig. R43 has been added in Supplementary Information as **Fig. S1**.

2. There are some grammar mistakes. For example: Page 6, Line 13.

Reply: Thanks for the reviewer’s suggestions. The “HRTME” has been revised with “HRTEM”.

3. In Fig. 2b, for the measurements under illumination, the power per unit area is quite large (16 mW). Does illumination have any irreversible on device operate? Also, provide incident optical power density.

Reply: Thanks for the reviewer’s suggestions. As the reviewer suggested, we investigated the stability of the device. Under 16 mW illumination, a persistent shift of transfer curve can be identified after turning off the laser, which indicates an irreversible degradation of device. To avoid the damage of device, we modified the experiment and evaluated the device performance with the maximum light intensity of 6 mW. As shown in **Fig. R44**, the transfer curve of device collected after 1 hour of illumination matches well the that at the initial stage. In addition, the corresponding incident optical power density is insert in **Fig. R45** as legends.

Fig. R44 Transfer curves (I_{ds} - V_{gs}) of device at $V_{ds}=1$ V with 6 mW incident light powers and dark.

Fig. R45 Transfer curves (I_{ds} - V_{gs}) of device at $V_{ds}=1$ V with different incident light powers.

Fig. R45 is updated as part of **Fig. 2** in main text.

4. Transfer and output curves of the device under illumination, and Fig. 2f were compared at different incident powers. It should be fairer to compare with equal powers.

Reply: Thanks for the reviewer's suggestions. We revised corresponding figures for comparison at same incident powers (**Fig. R46-R48**).

Fig. R46 Transfer curves (I_{ds} - V_{gs}) of device at $V_{ds}=1$ V with different incident light powers.

Fig. R47 Output curves of NbS₂/MoS₂ hybrid films device with various light intensity at fixed V_{gs} . **a** $V_{gs}=-50$ V, **b** $V_{gs}=0$ V and **c** $V_{gs}=50$ V.

Fig. R48 Optoelectronic characteristics of bare MoS₂ films device. Output curves of device with various light intensity at **a** $V_{gs}=-50$ V, **b** $V_{gs}=0$ V and **c** $V_{gs}=50$ V.

Fig. R46 is updated as part of **Fig. 2** in maintext, and **Fig. R47** and **R48** is updated as part of Supplementary Information (**Fig. S6** and **S7**).

5. Fig 2c, S8, and S9 do not clearly show PPC. After turning off the light, the current quickly approaches its initial state, which is ambiguous to call it pronounced PPC. Especially, in Figure S8, the current reaches about 87.5% of the initial current (i.e., dark current). Fig S10 better represents the PPC effect.

Reply: Thanks for the reviewer's suggestions. **Fig. 2c** and **Fig. S8** have been revised to demonstrate PPC clearly (**Fig. R49** and **Fig. R50**). As shown in **Fig. R49**, the persistent photocurrent of device is increased to about 45 % of peak photocurrent after removal of light stimulus. **Fig. R50** shows the decaying behavior of device during 1000 s. Compared with the data in original manuscript, the updated figure reveals more pronounced PPC effect with slower relaxing kinetic.

Fig. R51b (**Fig. S9b** in original manuscript) reveals the response speed of device. The magnitude of PPC and dark current are normalized to 1 and 0, respectively. By applying a positive voltage pulse, the PPC states can rapidly convert to initial stage (i.e. dark current) within 200 ms.

Fig. R49 Photoresponse of device to four light ON/OFF cycles at $V_{gs} = -50$ V.

Fig. R50 Retention characteristics of PPC in NbS₂/MoS₂ hybrid films device.

Fig. R51 Response speed of NbS₂/MoS₂ hybrid films device. **a** Photo-triggered rising time. **b** Electrical reset time, indicating PPC states can rapidly convert to initial stage within 200 ms.

Fig. R49 is updated as part of **Fig. 2** in main text, and **Fig. R50** is updated in Supplementary Information (**Fig. S9**).

6. The cross-talk between adjacent pixels should be characterized. A detailed study of photocurrent mapping by scanning photo-current microscopy may be helpful.

Reply: Thanks for the reviewer's suggestions. We first investigated the cross-talk characteristics in sensor array by using Kim *et al.* proposed methods⁴³. **Fig. R52a** show the optical microscopy image of an illuminated pixel and 8 adjacent pixels. The distances between the center of illuminated pixel and that of adjacent pixels are 97 μm (right, left), 61 μm (top, bottom), and 113 μm (diagonal), respectively. The corresponding photocurrents of illuminated pixel and adjacent pixels are given in **Fig. R52b**, indicating a negligible cross-talk between adjacent pixels for static image sensing.

In addition, we used same method to investigate the cross-talk during the movement of the targets. As shown in **Fig. R52c** and **d**, when target moves from the

bottom right pixel to top left pixel, there is no obvious photocurrent detected at pixels outside the trace. The results suggest that our sensor array is also capable of detecting moving target without cross-talk.

Fig. R52 Cross-talk characteristics of $\text{NbS}_2/\text{MoS}_2$ phototransistors array. **a** Optical microscopy image of an illuminated pixel and 8 adjacent pixels. The distances between center of illuminated pixel and that of adjacent pixels are 97 μm (right, left), 61 μm (top, bottom), and 113 μm (diagonal), respectively. **b** Corresponding photocurrent mapping of illuminated pixels and adjacent pixels. **c** Optical microscopy image of moving trace and adjacent pixels. The target moves from the bottom right pixel to top left pixel. **d** Corresponding photocurrent mapping of pixels at the trace and outside the trace.

Fig. R52 has been added in Supplementary Information (**Fig. S16**), and related description has been updated in main text:

“Supplementary Figure S16 demonstrates a negligible cross-talk between adjacent pixels in sensor array.”

*7. Contrary to the cross-talk issue in question #6, the acquired trajectory image cannot be obtained as in Fig. 4c because the moving targets moves very wide as shown in Fig.4b. Also, the channel size of 10*20 μm^2 and laser spot size of 5 μm^2 were used in this work. I am confused about that*

Reply: Thanks for the reviewer’s suggestions. We modified the experiment of trace extraction and related explanation in manuscript. The size of laser spot is increased to 10 μm^2 , which ensures the covering of active area of individual pixel. The spot moves with a uniform speed along the predefine trace in order to simulate the movement of target. As shown in **Fig. R53a**, the laser spot moves without any stop with the assistance of a piezoelectric displacement actuator. The moving time between adjacent pixels is about 2.5 s, and the effective illuminating time of each pixel is about 500 ms. The detection process is schematically illustrated as **Fig. R53b-d**. Due to the slow kinetics of photocurrent and memory effect, the later triggered pixel exhibits a higher level of photocurrent, and such spatial-temporal dependent photocurrent of pixels is consecutive during a relatively long time. In manuscript, we select 10 time points for measurement to demonstrate the dynamic change of motion trace and corresponding current information in array. The acquirement of whole trace actually only requires one measurement after the moving process finished, which allows to avoid the redundancy and latency.

Fig. R53 Schematic illustration of trace detection. **a** Schematic illustration of laser spot moving. **b** The pre-defined trace. **c** Time-depend decay current of each pixel at trace. **d** Current mapping of array after moving.

Fig. R54 shows the difference between traditional frame-based CMOS sensor and our device array for on-chip sensing motion. The conventional sensor captures image frame-by-frame with a specific clock frequency, and massive images are arranged by time sequence to form the motion trace with the assistance of additional processing units and memory modules, which unavoidably increases the complexity of entire vision chip, latency and power consumption. In contrast, the slow decay kinetics of conductance state in individual NbS₂/MoS₂ phototransistor allows array to direct process and generates the spatial-temporal current information according to the stimuli sequences. Each pixel located within the trajectory would spontaneously respond to the stimuli and parallelly real-time memorize and output the spatial-temporal dependent information. It would remarkably advance the efficiency of data processing and contribute to the system simplification.

In summary, the distinguishing feature of NbS₂/MoS₂ sensor array is that it can sense the optical stimuli and real-time generate and memorize the related spatial-temporal information, thus effectively decreasing the complexity of array, massive data storage, redundancy and latency. The spatial resolution of the detected trace can be further improved by increasing the density of the pixels.

Fig. R54 Comparison between frame-based camera and NbS₂/MoS₂ sensor for on-chip trace detection.

Fig. R54 has been added as part of **Fig. 4** in main text, and related description has been updated

*“As shown in **Fig. 4a**, front-end motion detection is generally realized based on CMOS vision chip, which captures visual information framed-by-frame at a predetermined rate. The separated memory and processing of frame data at sensor-level or pixel level would cause the unnecessarily inflating data.”*

“In contrast, according to the difference of electrical output in spatiotemporal dimension, the NbS₂/MoS₂ sensor array can derive the pre-processed information regarding to the relationship between the object motion position and timeline. The slow

kinetics of time-dependent decay photocurrent allows to directly process and generate the spatiotemporal information within sensor, which can be parallelly real-time memorized in each pixel. It would remarkably advance the efficiency of data processing and contribute to the system simplification.”

“A continuous laser scanning along the predefined trace is utilized to simulate the moving trajectory. The laser spot moves without any stop with the assistance of a piezoelectric displacement actuator. The moving time between adjacent pixels is about 2.5 s, and the effective illuminating time of each pixel is about 500 ms.” (Method section)

8. How did the authors calculate detectivity? The authors should measure the noise density spectrum and calculate the noise for real detectivity.

Reply: Thanks for the reviewer’s suggestions. The measured the noised density spectrum with respect to frequency (f) and calculated real detectivity are provided in **Fig. R55**.

In original manuscript, the specific detectivity is calculated by the equation:

$$D^* = \frac{RS^{\frac{1}{2}}}{(2eI_{dark})^{\frac{1}{2}}} \quad (3 - 1)$$

where R , S , e , and I_{dark} are responsivity, effective area, elementary charge and dark current, respectively.

To better evaluate the real detectivity, noise density spectrum of NbS₂/MoS₂ phototransistor was measured in dark condition (**Fig. R55a**). The real detectivity can be derived as follow:

$$D_{real}^* = \frac{RA^{\frac{1}{2}}}{S_n^{\frac{1}{2}}} \quad (3 - 2)$$

where the parameters R , A , and S_n are responsivity, effective area, and spectral noise power.

The maximum D_{real}^* are calculated to be 1.1×10^7 , 2.5×10^7 , 1.03×10^8 and 2.83×10^8 with $\Delta f = 1/10/100/1000$ Hz. **Fig. R55b** compare the detectivity of NbS₂/MoS₂ phototransistor obtained using above two methods. The value calculated from the noise density spectrum is lower than that obtained from equation (3-1). Similar results are also observed in bare MoS₂ device (**Fig. R55c and d**).

Fig. R55 Noise density spectrum with respect to frequency (f) and real specific detectivity. **a-b** Noise density spectrum of NbS₂/MoS₂ device and comparison of specific detectivity by using two methods. **c-d** Noise density spectrum of bare MoS₂ device and comparison of specific detectivity by using two methods.

Fig. R56 has been added as part of Supplementary Information (**Fig. S8**).

9. The authors should make references to these prior works for comprehensiveness (Nature communications 2021, 12, 3559, ACS Nano 2021, 15, 9, 15362–15370)

Reply: Thanks for the reviewer’s suggestions. The corresponding articles have been cited into sentences of “Metal oxide films, two-dimensional (2D) van der Waals (vdW) materials and related heterostructures have attracted extensive attention due to their applications in optoelectronic synaptic devices” and “Particularly, atomic-thin 2D vdW heterostructures reveal strong light-matter interaction, wide spectral range of optical absorption, and gate-tunable band structure”, respectively, in revised manuscript.

Related papers have been added as references 20 and 36 in manuscript.

“20. Hong, S. et al. Neuromorphic Active Pixel Image Sensor Array for Visual Memory. ACS Nano 15, 15362-15370 (2021).

36. Hong, S. et al. Highly sensitive active pixel image sensor array driven by large-area bilayer MoS₂ transistor circuitry. Nat. Commun. 12, 3559 (2021).”

References

1. John, R. A. et al. Synergistic Gating of Electro-Iono-Photoactive 2D Chalcogenide Neuristors: Coexistence of Hebbian and Homeostatic Synaptic Metaplasticity. *Adv. Mater.* **30**, e1800220 (2018).
2. Yang, Y. M. et al. Enhancing the fidelity of neurotransmission by activity-dependent facilitation of presynaptic potassium currents. *Nat. Commun.* **5**, 4564 (2014).
3. Underwood, B. J. Interference and forgetting. *Psychol. Rev.* **64**, 49-60 (1957).
4. Zhu, C. et al. Optical synaptic devices with ultra-low power consumption for neuromorphic computing. *Light Sci. Appl.* **11**, 337 (2022).
5. Pradhan, B. et al. Ultrasensitive and ultrathin phototransistors and photonic synapses using perovskite quantum dots grown from graphene lattice. *Sci. Adv.* **6**, eaay5225 (2020).
6. Qin, S. et al. A light-stimulated synaptic device based on graphene hybrid phototransistor. *2d Mater.* **4**, 035022 (2017).
7. Dai, S. et al. Light-Stimulated Synaptic Devices Utilizing Interfacial Effect of Organic Field-Effect Transistors. *ACS Appl. Mater. Interfaces* **10**, 21472-21480 (2018).
8. Wang, Y. et al. Optoelectronic Synaptic Devices for Neuromorphic Computing. *Adv. Intell. Syst.* **3**, 2000099 (2020).
9. Shen, Z. et al. Emerging Optical In-Memory Computing Sensor Synapses Based on Low-Dimensional Nanomaterials for Neuromorphic Networks. *Adv. Intell. Sys.* **4**, 2100236 (2022).
10. Sun, Y. et al. Programmable van - der - Waals heterostructure - enabled optoelectronic synaptic floating-gate transistors with ultra-low energy consumption. *InfoMat.* **4**, e12317 (2022).
11. Karbalaee Akbari, M. & Zhuiykov, S. A bioinspired optoelectronically engineered artificial neurorobotics device with sensorimotor functionalities. *Nat. Commun.* **10**, 3873 (2019).
12. Zhang, Z. et al. In-sensor reservoir computing system for latent fingerprint recognition with deep ultraviolet photo-synapses and memristor array. *Nat. Commun.* **13**, 6590 (2022).
13. Hu, Y. et al. Ultralow Power Optical Synapses Based on MoS₂ Layers by Indium-Induced Surface Charge Doping for Biomimetic Eyes. *Adv. Mater.* **33**, e2104960 (2021).
14. Yin, L. et al. Optically Stimulated Synaptic Devices Based on the Hybrid Structure of Silicon Nanomembrane and Perovskite. *Nano Lett.* **20**, 3378-3387 (2020).
15. Li, Q. et al. Ultralow Power Wearable Organic Ferroelectric Device for Optoelectronic Neuromorphic Computing. *Nano Lett.* **22**, 6435-6443 (2022).
16. Zhou, F. & Chai, Y. Near-sensor and in-sensor computing. *Nat. Electron.* **3**, 664-671 (2020).
17. Bourbakis, N. G. & Mertoguno, J. S. Kydon: An autonomous, multi-layer image-understanding system: Lower layers. *Eng. Appl. Artif. Intell.* **9**, 43-52 (1996).
18. Mertoguno, S. & Bourbakis, N. G. A digital retina-like low-level vision processor. *IEEE Trans. Syst., Man, Cybern. B, Cybern.* **33**, 782-788 (2003).
19. Zucker, R. S. & Regehr, W. G. Short-term synaptic plasticity. *Annu. Rev. Physiol.* **64**, 355-405 (2002).
20. Liu, K. et al. An optoelectronic synapse based on α -In₂Se₃ with controllable temporal dynamics for multimode and multiscale reservoir computing. *Nat. Electron.* **5**, 761-773 (2022).

21. Zhu, Q. B. et al. A flexible ultrasensitive optoelectronic sensor array for neuromorphic vision systems. *Nat. Commun.* **12**, 1798 (2021).
22. Cooke, S. F. & Bliss, T. V. Plasticity in the human central nervous system. *Brain* **129**, 1659-1673 (2006).
23. Dodda, A. et al. Active pixel sensor matrix based on monolayer MoS₂ phototransistor array. *Nat. Mater.* **21**, 1379-1387 (2022).
24. Zhang, H. S. et al. Co-assembled perylene/graphene oxide photosensitive heterobilayer for efficient neuromorphics. *Nat. Commun.* **13**, 4996 (2022).
25. Yu, J. et al. Bioinspired mechano-photonic artificial synapse based on graphene/MoS₂ heterostructure. *Sci. Adv.* **7**, eabd9117 (2021).
26. Abraham, W. C. & Goddard, G. V. Asymmetric relationships between homosynaptic long-term potentiation and heterosynaptic long-term depression. *Nature* **305**, 717-719 (1983).
27. Yang, C. S. et al. All-Solid-State Synaptic Transistor with Ultralow Conductance for Neuromorphic Computing. *Adv. Funct. Mater.* **28**, 1804170 (2018).
28. Wang, Y. et al. Dual-Modal Optoelectronic Synaptic Devices with Versatile Synaptic Plasticity. *Adv. Funct. Mater.* **32**, 2107973 (2021).
29. Teli, S., Cahyadi, W. A. & Chung, Y. H. Optical Camera Communication: Motion over Camera. *IEEE Commun. Mag.* **55**, 156-162 (2017).
30. He, J., Huang, Z. & Yu, K. High-accuracy scheme based on a look-up table for motion detection in an optical camera communication system. *Opt. Express* **28**, 10270-10279 (2020).
31. Chung, Y. H., Cahyadi, W. A. & Teli, S. Trained neurons-based motion detection in optical camera communications. *Opt. Eng.* **57**, 040501-040501. (2018).
32. Sewaiwar, A., Tiwari, S. V. & Chung, Y. H. Visible light communication based motion detection. *Opt. Express* **23**, 18769-18776 (2015).
33. Teli, S. R., Zvanovec, S. & Ghassemlooy, Z. Experimental Investigation of Neuron Based Motion Detection in Internet of Things using Optical Camera Communications. *2019 26th International Conference on Telecommunications (ICT)*, 202-205 (2019).
34. Wu, J. et al. A Multimode CMOS Vision Sensor With On-Chip Motion Direction Detection and Simultaneous Energy Harvesting Capabilities. *IEEE Sens. J.* **22**, 12808-12819 (2022).
35. Simoni, A. et al. A single-chip optical sensor with analog memory for motion detection. *IEEE J. Solid-State Circuits* **30**, 800-806 (1995).
36. Lichtsteiner, P., Posch, C. & Delbruck, T. A 128X128 120 dB 15 μ s Latency Asynchronous Temporal Contrast Vision Sensor. *IEEE J. Solid-State Circuits* **43**, 566-576 (2008).
37. Culurciello, E., Etienne-Cummings, R. & Boahen, K. A. A biomorphic digital image sensor. *IEEE J. Solid-State Circuits* **38**, 281-294 (2003).
38. Posch, C., Serrano-Gotarredona, T., Linares-Barranco, B. & Delbruck, T. Retinomorphic Event-Based Vision Sensors: Bioinspired Cameras With Spiking Output. *Proc. IEEE Inst. Electr. Electron. Eng.* **102**, 1470-1484 (2014).
39. Liao, F., Zhou, F. & Chai, Y. Neuromorphic vision sensors: Principle, progress and perspectives. *J. Semicond.* **42** (2021).
40. Gallego, G. et al. Event-Based Vision: A Survey. *IEEE Trans. Pattern. Anal. Mach. Intell.* **44**, 154-180 (2022).
41. Zaghloul, K. A. & Boahen, K. Optic nerve signals in a neuromorphic chip II: Testing and results. *IEEE Trans. Biomed. Eng.* **51**, 667-675 (2004).

42. Lenero-Bardallo, J. A., Serrano-Gotarredona, T. & Linares-Barranco, B. A 3.6 μ s Latency Asynchronous Frame-Free Event-Driven Dynamic-Vision-Sensor. *IEEE J. Solid-State Circuits* **46**, 1443-1455 (2011).
43. Hong, S. et al. Highly sensitive active pixel image sensor array driven by large-area bilayer MoS₂ transistor circuitry. *Nat. Commun.* **12**, 3559 (2021).

REVIEWER COMMENTS

Reviewer #1 (Remarks to the Author):

The manuscript has been updated by including many supplementary data/results. However, despite the detailed comments provided in the initial review, the authors seem not to understand those correctly. More importantly, the quality of the study has not been improved at all and the rebuttal letter has clarified the author's misconceptions about the biological nervous system in the first place, as well as the visual science and neuromorphic engineering.

It is unfortunate that this reviewer cannot find a potential of this manuscript to be accepted for publication, at least in this journal. Nevertheless, this reviewer provided further comments in the pdf file of the manuscript, hoping the authors may improve their study and manuscript for the future.

Reviewer #2 (Remarks to the Author):

The paper demonstrates the implementation of a 10x10 vision sensor fabricated with NbS₂/MoS₂ hybrid films which exhibit a persistent response to illumination. The paper proposes to use this characteristic for integrating sensing, memory, and filtering capabilities, performing trajectory detection.

The paper demonstrates experimental results and compares the performance with benchmarked datasets.

Just a minor remaining comment.

'However, previously reported neuromorphic image sensors are largely limited to static information detecting and processing.'

This sentence should be rephrased as new dynamic vision sensors are based on the detection of temporal changes in illumination. Nowadays, most neuromorphic vision sensors are based on this concept. Many functional, high-resolution dynamic vision sensor architectures have been published in the last years since the first DVS was published in 2008 as already referenced in the experimental part of the paper.

Reviewer #3 (Remarks to the Author):

The authors have improved the manuscript according to the comments and the manuscript now is acceptable for publication.

Responses to reviewers' comments (NCOMMS-22-41687A)

We are grateful for the reviewers' comments on our work. We also thank very much for the reviewers' time and efforts to provide us with detailed and valuable comments. These suggestions are greatly helpful to us for improving the quality of this work. We have tried our best to address all the concerns raised by the reviewer point-by-point as follows, and highlighted the revised parts in red color.

Our responses to reviewer 1's comments are from **page R1** to **page R23**, in which reviewer 1's comments by the built-in comment function in pdf were cited as screen capture images. The responses to reviewer 2's comments are from **page R24**.

Reviewer #1

The manuscript has been updated by including many supplementary data/results. However, despite the detailed comments provided in the initial review, the authors seem not to understand those correctly. More importantly, the quality of the study has not been improved at all and the rebuttal letter has clarified the author's misconceptions about the biological nervous system in the first place, as well as the visual science and neuromorphic engineering.

It is unfortunate that this reviewer cannot find a potential of this manuscript to be accepted for publication, at least in this journal. Nevertheless, this reviewer provided further comments in the pdf file of the manuscript, hoping the authors may improve their study and manuscript for the future.

Reply: Thanks very much for the reviewer's comments. We are sorry for the misconceptions about the biological nervous system in the initial responses. After carefully reading the referee's two-round comments, we noted that the definition of synaptic behaviors in electronic and optoelectronic devices obviously deviates from the basic concepts of neuroscience, even though some functions may seem similar to those of biological synapses. According to the reviewer's constructive suggestions, we have

revised all the descriptions about the optical modulated current behaviors of NbS₂/MoS₂ phototransistors to avoid possible misunderstanding of the biological nervous system and neuromorphic engineering for readers. In addition, we have replaced the term “chip” with “**sensor array**”, which more accurately describes the devices in this work and avoids misunderstandings of our work’s purpose and importance. This work aims to report an emerging sensor array with pre-processing functions instead of a conventional processing chip. The proposed prototype vision sensors can not only detect optical signals as commonly reported photodetectors, but also enable the implementation of more advanced processing functions **in-situ** the sensor array.

To be more specific, this work aims to construct a front-end sensor array monolithically integrated with both image sensing and several **pre-processing capabilities** (e.g., contrast enhancement and trajectory restoration), as illustrated in **Fig. R1**. The pre-processing capabilities of the sensor array originate from the unique and non-linear time-dependent current relaxation dynamics in each sensor cell (corresponding to each pixel), which is distinguishable from the conventional photodiode cell of a CMOS image sensor. For more clear clarification, we achieved multifunctional integration at the **device level** instead of conventional system-on-chip (SoC) integration or heterogeneous unit integration at the chip level. The functional integration at the device level can largely improve the integration density and simplify the circuit design complexity for future high-density and low-power applications. Especially, for power-limited and size-limited applications such as drones, our array can exhibit great advantages in terms of density, low power, and low latency since data transfer can be largely reduced between the sensor and processing units.

Fig.R1 Schematic diagram of perceiving both static and moving targets, in which neuron-inspired sensor array executes both image sensing and several front-end pre-processing tasks at device level for reducing the redundant data and the design complexity.

For example, the contrast enhancement between the target letter and background can be achieved by our sensor array itself in addition to the image sensing, since the current relaxation rates induced by the highlight target letter pixels with higher intensities are slower than that induced by background pixels with lower intensities, which contributes to an enlarged contrast. Additionally, due to the optically-induced current response and time-dependent relaxation characteristics, the sensor array can register the sequence of stimuli independently according to the different conductance states in trajectory pixels without any external circuit integration. The above features allow our device to execute certain processing functions in addition to optical sensing compared to the CMOS image sensors, which only perform image sensing function. Although our sensor array cannot implement the functions as much as the image processing systems integrated with CMOS image sensors and processing chip, the processing function integrated into the sensor array (without increasing the cell size) can help to reduce the redundant data and the design complexity of the post-processors.

As mentioned in the reviewer's comments, to perform high-level recognition in a manner similar to biological vision system, the prototype sensors inevitably require the integration of external circuit units for **post-processing (Fig. R1)**. However, it should be noted that the integrated image sensing and pre-processing functions in our array can effectively reduce the design complexity and unnecessary data transmission within the vision system, as well as alleviate the computing burden in the post-processors.

Therefore, compared to the traditional CMOS-based sensors, our sensor array demonstrates superiorities in system simplification and multiple functions integration.

To guarantee accurate description about the role of our sensor array in artificial vision system, **Fig. 1b** in main text has been revised as below:

Fig 1b Implementation of a neuro-inspired vision system for precepting both static and moving targets.”

(1)

Neuromorphic vision chip for high-accuracy static image recognition and dynamic trace extraction

1. “Neuro-inspired”, but not “Neuro-morphic” is suited for the proposed chip.

Reply: Thanks very much for the reviewer’s comments. As the term “neuromorphic” may not be suitable here, the term “Neuromorphic” has been replaced with “Neuro-inspired” according to the reviewer’s suggestion.

(2)

Bioinspired ~~neuromorphic~~ vision system holds great promises to address the growing

2. This is redundant, and does not fit to the manuscript.

Reply: Thanks for the reviewer’s suggestions. The word “Neuromorphic” has been deleted.

(3)

~~synaptic~~ plasticity and low electrical energy consumption. Neuromorphic vision chip

3. This should be deleted.

Reply: Thanks for the reviewer's suggestions. The word "synaptic" has been deleted.

(4)

sensing, memory, and filtering capabilities for static image, ~~delivering a high image~~

~~recognition accuracy of about 90%.~~ More importantly, detection and in-sensor

4. This is incorrect. ANN/CNN, but not the chip, delivers the recognition with the high accuracy.

Reply: Thanks for the reviewer's suggestions. We are sorry that the description was not clear in the original manuscript. The recognition procedure is indeed performed by ANN/CNN in post-processing. We have revised the corresponding description in the main text:

"A neuro-inspired vision sensor array with 10×10 NbS₂/MoS₂ phototransistors enables the highly-integrated functions including sensing, memory, and contrast enhancement capabilities for static image, which benefits the convolutional neural network (CNN) with a high image recognition accuracy of about 90%."

(5)

extraction of moving object trajectory were experimentally implemented, ~~with a high~~

~~restoration accuracy of over 92%.~~ The demonstration of the neuromorphic vision chip

5. The restoration accuracy is attributed to the external processing, but not the chip.

Reply: Thanks for the reviewer's suggestions. We are sorry that the description was not clear in the original manuscript. The trajectories restoration is definitely processed by our NbS₂/MoS₂ vision sensor array. As shown in **Fig. R2**, the optically-induced current relaxation allows the sensor to clearly register the sequence of stimuli without any external processing circuit. Additionally, attributing to the retention of current, all of trace information can be well registered in NbS₂/MoS₂ vision sensor itself. In other words, the sequence of trace has been recorded in form of different conductance states within the sensor array during a certain period.

It should be noted that it is still necessary to evaluate the practical restored accuracy with consideration of device (pixel) variance. Inevitable pixel-to-pixel variation may make the sensor array register the wrong sequence of stimuli, thus decreasing the restoration accuracy in experimental implementation. Considering this, we performed simulation to restore 400 random trajectories in Python based on the extracted relaxation characteristics of NbS₂/MoS₂ sensor array. The restored sequence of each trajectory, which is in form of graded conductance states, is then compared with the corresponding input trajectory to determine the accuracy. The processing mentioned by the reviewer is exactly to evaluate the practical accuracy with consideration of pixel variance through comparing restored trajectory data from the sensor with original trajectories, which has been provided detailed demonstration in Supplementary Information Figures S21-S22 and the corresponding description.

Fig. R2 Schematic illustration of trajectory restoration. **a** Schematic illustration of laser spot moving trajectory. **b** Photocurrent evolution of different pixels. **c** Image of restored trajectory by a single measurement.

(6)

implementation of high-performance artificial ~~neuromorphic~~ vision system.

6. *This is redundant, and does not fit to the manuscript.*

Reply: Thanks for the reviewer's suggestions. The word "neuromorphic" has been deleted.

(7)

processing such as ~~noise suppression, image sharpening and motion extraction, etc.~~,

7. *References must be cited.*

Reply: Thanks for the reviewer’s suggestions. The following references have been cited.

“7. van Rossum, M. C. & Smith, R. G. Noise removal at the rod synapse of mammalian retina. *Vis. Neurosci.* **15**, 809-821 (1998).

8. Field, G. D. & Rieke, F. Nonlinear signal transfer from mouse rods to bipolar cells and implications for visual sensitivity. *Neuron* **34**, 773-785 (2002).

9. Sampath, A. P. & Rieke, F. Selective transmission of single photon responses by saturation at the rod-to-rod bipolar synapse. *Neuron* **41**, 431-443 (2004).

10. Kalloniatis, M. & Luu, C. Light and dark adaptation. In *Webvision: The Organization of the Retina and Visual System* (eds Kolb, H. et al.) (Univ. Utah Health Sciences Center, 1995).

11. Meister, M. & Tessier-Lavigne, M. Low-Level Visual Processing: The Retina. in *Principles of Neural Science, Fifth Edition* (eds E.R. Kandel et al.) Ch. 26, (McGraw-Hill Education, 2014).

12. Olveczky, B. P., Baccus, S. A. & Meister, M. Segregation of object and background motion in the retina. *Nature* **423**, 401-408 (2003).

13. Olveczky, B. P., Baccus, S. A. & Meister, M. Retinal adaptation to object motion. *Neuron* **56**, 689-700 (2007).”

(8)

development of fully hardware-implemented **neuromorphic** vision system, which

8. This is redundant. Besides, since there is little scientific evidence that demonstrates the trajectory extraction carried by any biological retina, the absence of it hardly stymies the development of neuromorphic vision system. Thus, "neuromorphic" should be deleted.

Reply: Thanks for the reviewer’s comment. The term of “neuromorphic” has been deleted according to reviewer’s suggestion.

(9)

A **neuromorphic** vision chip incorporated with a 10 × 10 sensor array enables the

9. "Neuro-inspired", but not "Neuro-morphic" is suited for the proposed chip. Besides, 10 x 10 sensor pixels are too less in number as for a vision chip.

Reply: Thanks for the reviewer’s suggestions. The word “neuromorphic” has been replaced with “neuro-inspired”. As the review mentioned, we admit that the 10×10 sensor pixels are less compared to silicon-based vision chips due to the limitation of the current fabrication technology and uniformity of large-scale materials. Nonetheless, the number of pixels in our prototype vision sensor array is still comparable to the current emerging bio-inspired vision sensor array based on 2D materials equipped with processing capabilities, such as 10×10 pixels MoS₂-based chip for visual adaptation (*Bioinspired in-sensor visual adaptation for accurate perception. Nature Electronics, 2022, 5(2): 84-91*) and 3×3 pixels WS₂-based chip for ultrafast machine vision (*Ultrafast machine vision with 2D material neural network image sensors. Nature, 2020, 579(7797): 62-66*).

As a proof-of-concept, our work demonstrates the integration of processing functions for both static and motion objects in a highly simplified optical sensor array without external processing circuit for the first time, which has not been reported in the current in-sensor processing system based on the emerging devices yet. Our design provides a solution to reduce redundant data and construct a cost-effective machine vision system. Following the referee’s suggestion, we fabricated a larger scale sensor array with 32×32 NbS₂/MoS₂ phototransistor matrix to demonstrate the good scalability, as shown in **Fig. R3**.

Fig. R3 NbS₂/MoS₂ sensor array with 32×32 device matrix. **a** Photograph of the 32×32 sensor array. **b** Magnified image of the core area of sensor array. Scale bar: 5 mm.

(10)

system. More importantly, the trajectory ~~detection and real-time memorization~~ of

10. "registration", rather than "detection" or "memorization" is better suited for the function of the chip.

Reply: Thanks for the reviewer's suggestions. The words "detection" and "memorization" has been replaced with "registration".

(11)

the cognition of visual stimuli ⁴¹⁻⁴⁵. The functions in terms of noise filtering, contour ~~finding~~, and ~~object-graph-generation~~ of retina imbue brain cortex with high efficiency

11. The retina does not "find" contours, but may enhance contrast edges.

Reply: Thanks for the reviewer's suggestions. The corresponding phrase has been replaced with "edge enhancement".

(12)

~~finding~~, and ~~object-graph-generation~~ of retina imbue brain cortex with high efficiency

12. The retina does not generate any graph.

Reply: Thanks for the reviewer's suggestions. We are sorry for the mistakes and the corresponding phrase has been deleted.

(13)

amplitude of PSC (A₂) triggered by the second light pulse is significantly higher than that (A₁) triggered by the first pulse. It can be attributed to the superposition of

(14)

pulse interval⁵³. The PPF index is defined as the ratio of A_2/A_1 , and the relationship between the PPF index and pulse interval time (Δt) is plotted in Fig. 2e. With the increase of the Δt from 0.02 to 30 s, the PPF index gradually declines from 129% to 112%. It can be fitted well by a double-exponential function:

$$PPF \text{ index} = \alpha_1 e^{-\Delta t/\tau_1} + \alpha_2 e^{-\Delta t/\tau_2} + C \quad (1)$$

where α_1 , α_2 and C are the facilitation constants, and Δt , τ_1 , and τ_2 represent the interval time in two pulses, rapid relaxation time and slow relaxation time, respectively. The two characteristic timescales, τ_1 and τ_2 , are estimated to be 8.24 s and 0.21 s, respectively. The τ_1 is an order of magnitude larger than τ_2 , indicating a successful stimuli process, which is consistent with previously reported optically stimulated synaptic devices⁵⁴⁻⁵⁸. The dynamic tunable processes of synaptic plasticity were also demonstrated by changing light intensity, duration, and frequency (Supplementary Figure S11).

13&14. At least in the field of neuroscience, the amplitude of the second response should be measured from the level just before the onset of the second stimulation pulse (the trough), not from the original baseline before the first stimulation pulse. Hence, it is doubtful that the proposed sensor exhibits a PPF-like property. As described above, the authors' understanding of biological paired-pulse facilitation is completely incorrect, so as in several published papers cited here. Thus, if the authors wish to demonstrate PPF-like properties, then the authors must re-analyze the data as a whole.

Reply: Thanks for the reviewer's suggestions. We are sorry for the misunderstanding regarding the basic concepts in the field of neuroscience, including paired-pulse facilitation (PPF), short-term plasticity (STP) and long-term plasticity (LTP), etc. In the initial responses, we were too focused on the results of our synaptic function emulation to properly address the fundamental definition of a biological synapse.

As a result, the term “PPF” is not appropriate to express the characteristics of the NbS₂/MoS₂ phototransistor. Note that the channel conductance of NbS₂/MoS₂ phototransistor can be effectively modulated by pulse interval time. Hence, we wish to use the term “pulse-interval dependent conductance plasticity” to describe the device behavior.

For PPF measurement, in the field of neuroscience, the amplitude of the second response should be measured from the level just before the onset of the second stimulation pulse (the trough). In the field of 2D electronic and optoelectronic devices, the amplitude of the second response is usually measured from the original baseline. The routine measurement method between the two fields is rather distinct. We apologize for the misconception of PPF definition again.

We have revised the corresponding description in main text:

*“The tunable memory characteristics of NbS₂/MoS₂ phototransistor enable it exhibit diversified photo-modulation conductance plasticity. The pulse-interval dependent plasticity was demonstrated by applying two successive light pulses with duration of 500 ms and different interval durations at $V_g = -50$ V. We can see that the amplitude of photocurrent (A_2) triggered by the second light pulse is significantly higher than that (A_1) triggered by the first pulse. It can be attributed to the superposition of subsequently excited carriers with residual carriers before the following pulse in a short pulse interval. The modulation ratio is defined as A_2/A_1 , and the relationship between the ratio and pulse interval time (Δt) is plotted in **Fig. 2e**. With the increase of the Δt from 0.02 to 30 s, the modulation ratio gradually declines from 129% to 112%. It can be fitted well by a double-exponential function:*

$$\text{Modulation ratio} = \alpha_1 e^{-\Delta t/\tau_1} + \alpha_2 e^{-\Delta t/\tau_2} + C \quad (1)$$

where α_1 , α_2 and C are the facilitation constants, and Δt , τ_1 , and τ_2 represent the interval time in two pulses, rapid relaxation time and slow relaxation time, respectively. The two

characteristic timescales, τ_1 and τ_2 , are estimated to be 8.24 s and 0.21 s, respectively. The τ_1 is an order of magnitude larger than τ_2 , indicating a successful stimuli process, consistent with previously reported neuro-inspired optically stimulated devices. The dynamic tunable processes of conductance plasticity were also demonstrated by changing light intensity, duration and frequency (**Supplementary Figure S11**).”

(15)

amplitude of PSC (A_2) triggered by the second light pulse is significantly higher than that (A_1) triggered by the first pulse. It can be attributed to the superposition of

15. What is the "PSC"? It must be spelled out on first occurrence.

Reply: Thanks for the reviewer’s suggestions. We are sorry for the mistake. The “PSC” refers to postsynaptic current. As reviewer’s suggestion, synaptic functions are not appropriate to be used for describing behavior of our device. Therefore, the word “PSC” has been replaced by “photocurrent”.

(16)

20 s, which conceptually refers to STP. LTP is achieved by applying 60 successive

(17)

20 s, which conceptually refers to STP. LTP is achieved by applying 60 successive

pulses, and 30% of peak photocurrent can be kept even after a long decay time of 350

16&17. Again, in the field of neuroscience, such a property is not called STP. This simply reflects slow dynamics of the response decay. Any type of potentiation was not demonstrated in the experiment here. If the authors wish to demonstrate it, then a test stimulus pulse should be applied during this slow decay phase. And if the peak response amplitude to the test pulse is larger than that induced by the same test pulse applied before the two successive pulses, then there might be a form of potentiation. Again, in the field of neuroscience, such a property is not called LTP. The authors must re-do experiments and analyses.

Reply: Thanks for the reviewer’s suggestions. We are sorry for the misunderstanding about the concepts of “STP” and “LTP”. To guarantee the accuracy, we replaced the

terms “short-term plasticity (STP)” and “long-term plasticity (LTP)” with “short-term relaxation (STR)” and “long-term relaxation (LTR)”.

The implementation of pre-processing functions of NbS₂/MoS₂ highly relies on the non-linear photocurrent relaxing process. The light intensity-dependent relaxation rate of photocurrent enables sensor enlarge the contrast between target letter and background, and trace extraction is also realized by time-dependent relaxation kinetic of photocurrent. Considering of this point, we have revised the corresponding description in main text:

“The evolution of short-term relaxation (STR) and long-term relaxation (LTR) are revealed by applying different optical pulses number to phototransistor. As shown in Fig. 2f, after applying two successive pulses with duration of 300 ms, photocurrent declines rapidly to the initial stage within 20 s, which conceptually refers to STR. LTR is achieved by applying 60 successive pulses, and 30% of peak photocurrent can be kept even after a long decay time of 350 s.”

(18)

s. Fig. 2g shows the characteristics of optical potentiation and electrical inhibition. The

18. It is believed that this is not potentiation but accumulation due to the slow response decay.

Reply: Thanks for the reviewer’s suggestions. The phrase “optical potentiation” has been revised with “optical accumulation”.

(19)

primary information from the massive noisy inputs 8.71.72. Similarly, effective

19. These reference papers do not support the text written here. It is seriously doubtful that the authors understand these papers and other neuroscience references.

Reply: Thanks for the reviewer’s suggestions. We are sorry that some references might not properly cited. These references reported the mechanism of separating the small single-photon signal in one or a few rods from the continuous electrical noise that is present in all photoreceptors. This mechanism is different from the function of our proposed sensor array, which enlarges light signals contrast from backgrounds and objects selectively through the non-linear relationship between photocurrent relaxation

rate and light intensity. Therefore, the corresponding references have been deleted. The corresponding description in main text has been revised:

“The effective modulation of conductance relaxation speed is also revealed in NbS₂/MoS₂ optical sensor through tuning the intensity of light stimulation, thus enabling the experimentally implementation of image pre-processing for contrast enhancement between target image and background.”

(20)

². It is clear that the pre-processed image after repeated training demonstrates an enlarged difference between the grey scale of pixels over the input images, thus contributing to an output image with enhanced contrast. The results suggested the noise filtering function can be hardware implemented base on our NbS₂/MoS₂ sensor array without using any external circuits.

(21)

background noise signals of input visual information can be effectively smoothed for high-accuracy static image recognition.

20&21. It is obvious that the pre-processing, i.e. noise reduction, is not executed by the proposed chip, but externally by the manipulation of laser power and pulse repetition, both of which are none of the chip function. Namely, it is not demonstrated that the chip has such a pre-processing capability. The chip passively responded to the manipulated stimulus inputs. Again, the noise reduction is done by the intentional manipulation of laser stimuli. The chip simply responded to the manipulated stimulus inputs.

Reply: Thanks for the reviewer’s suggestions. We are sorry that the description was not clear in the original manuscript. The work aims to construct an optical sensor array with several certain pre-processing functions instead of a processing chip. Note that the sensor array features a highly integrated structure that only contains 100 individual NbS₂/MoS₂ phototransistors as 100 pixels, without integration of external processing circuits to perform pre-processing tasks.

The “noise reduction” refers to the capability that can process signals from backgrounds and objects selectively. Benefiting from the non-linear response between photocurrent relaxation rate and light intensity, our sensor array can conduct the function of “noise reduction” independently. Particularly, low-intensity background light signals are suppressed, while high-intensity feature signals are enhanced, thus increasing the contrast between target information and background for highlighting the key feature of image. To better describe this function, we also replaced the term “noise reduction” with “contrast enhancement” for accuracy.

Indeed, a biological eye also continuously receives a series of light stimulus with different powers and frequencies, which can be regarded as a series of frames inputs. Due to the non-linear relaxation behavior of photocurrent in our sensor, the effect of contrast enhancement can be accumulated by applying a continuous light stimulus, just like when people observe certain objects, the information will be more impressive with increased observation duration. Here are some recent papers published in Nature series journals that may prove the promising potentials of neuro-inspired devices in image enhancement through the non-linear current behaviors (*Optoelectronic resistive random access memory for neuromorphic vision sensors. Nature Nanotechnology, 2019, 14(8): 776-782, Curved neuromorphic image sensor array using a MoS₂-organic heterostructure inspired by the human visual recognition system. Nature Communications, 2020, 11: 5934; Photo-induced non-volatile VO₂ phase transition for neuromorphic ultraviolet sensors. Nature Communications, 2022, 13(1): 1729).*

In contrast, the similar function of “contrast enhancement” cannot be realized by traditional photodetectors independently. Here, a commercial silicon photodetector (purchased from Metatest corporation in Nanjing, China) was utilized for demonstration under the identical testing conditions (**Fig. R4a**). As shown in **Figs. R4b-e**, it is obvious that the silicon photodetector cannot store the light information with persistent conductance update, as well as enlarge the difference on external light stimuli. To compensate above limitation of CMOS-based photodetectors, additional hardware integration and external algorithms must be utilized¹⁻⁴, which poses great

challenges to achieve a compromise between efficient hardware utilization and processing performance for optimal cost efficiency.

In summary, the hardware implementation of image enhancement based on emerging NbS₂/MoS₂ vision sensor enables the vision system to maintain high accuracy when making a compromise for hardware simplification.

Fig. R4 Image enhancement performance of silicon photodetector in same testing condition with main text. **a** Photograph of purchased silicon photodetector. **b** Behavior of photocurrent triggered by different light intensity. **c** Normalized light intensity of input image. **d** Normalized peak photocurrent of silicon photodetector array. **e** Normalized current of silicon photodetector array with same scale bar of intensity after decaying 150 s.

(22)

To evaluate the recognition accuracy of the images before and after noise filtering,

22. Because the noise reduction was not done by the chip as described above, the whole CNN part in the manuscript is believed not to make a sense in this study. If the authors wish to have these materials, then those must be described as the supplementary information, but not in the main body of the manuscript.

Reply: Thanks for the reviewer’s suggestions. As responses to comment #20 and #21 mentioned, the contrast enhancement function is indeed utilized by our sensor array independently. Because of the enhanced image contrast between the target letter and background, a remarkable improvement in recognition rate in the post-ANN/CNN can be demonstrated after the MoS₂/NbS₂ sensor array employed for the front-end image pre-processing, which has the potential to reduce the burden on post-processing and

hardware requirements of post-processor. Note that the same accuracy can also be achieved in a vision system with conventional silicon image sensors by integrating an external contrast enhancement circuit or increasing the scale of CNN, but it would inevitably increase the cost and complexity of the whole vision system.

In summary, the demonstration of the CNN part can facilitate the readers to understand the advanced performance of NbS₂/MoS₂ sensor in constructing a cost-effective and compact vision system.

(23)

in Fig. 3f. One can see that a remarkable improvement of recognition rate is achieved

from 78% to 90% after the integration of pre-processing module. It means that with the

23. If ones are familiar with machine learning, then they know that a well-trained CNN model can classify the input images shown on the left (unfiltered) and right (filtered) of Fig. 3e with approximately the same accuracy. It is doubtful that the CNN training procedure here was inadequate. Or, the accuracy was somehow controlled to be lower for the unfiltered images than for the filtered images.

Reply: Thanks very much for the reviewer's comment. We are sorry that the description was not clear in the original manuscript. As shown in **Fig. R5**, an enhanced image dataset based on our sensor array and an unenhanced image dataset based on traditional silicon photodetectors are generated from the identical original dataset, and both generated datasets are input into the identical CNN model for training and testing.

As you mentioned, a CNN model can classify the raw images and pre-processed images with approximately the same accuracy if the CNN can be well-trained that equipped with larger number of layers. However, it needs high level of power consumption, area consumption and computing resources. It should be noted that small CNN models are preferred in some scenarios with limited power, size or computation resources, such as in edge computing applications and mobile devices. To improve the accuracy of small-sized CNN models on noisy images, pre-processing techniques such as image denoising, sharpening, or enhancement is highly required. Therefore, these image pre-processing capabilities of front-end vision sensor would be highly demanded

to increase the recognition accuracy of small-sized CNN model, making it suitable for edge computing scenarios where computing resource constraint exists.

We acknowledge that the development of neuro-inspired vision sensors based on emerging materials is still in its infancy. Although real-scenario application of these sensor array requires substantial effort, the integration of multiple processing functions into a single phototransistor without need for external processing circuit is bringing revolutionary strategy in data transfer reduction and system structure simplification.

Fig. R5 Schematic procedures for recognition process between enhanced image and unenhanced image.

(24)

~~neuromorphic-vision chip~~ is also ~~capable~~ of detecting and recognizing the trajectory of

24. The chip itself can not detect or recognize it. The chip might be able to helpful.

Reply: Thanks for the reviewer’s suggestions. The corresponding sentence has been revised in main text:

“In addition to the static image sensing and pre-processing, the designed NbS₂/MoS₂ neuro-inspired vision sensor is also helpful for detecting and recognizing the trajectory of moving objects.”

(25)

from pixel 0 to pixel 9. To dynamically track the trajectory change, the remaining photocurrent of each pixel along the trajectory was collected in sequence for 10 times and map-plotted in panel (i) of Fig. 4c. The collected photocurrent of each pixel

25. This is post-processing, which still needs a computer or a processing device other than the chip and laser. This must pose "challenges for the hardware simplification of vision system", as similar to DVS and any other optical sensor units.

Reply: Thanks for the reviewer's suggestions. We are sorry that the description was not clear in original manuscript. The trajectories restoration can be processed in NbS₂/MoS₂ vision sensors independently, while the "post-processing" that the reviewer mentioned is the visualization of restored trajectories data through simply reading the different conductance states of sensors.

DVS is a very excellent vision sensor based on biological principles, which emulates a simplified structure and encoding scheme of the retina at the pixel level. Asynchronous output in the form of an event stream with address-event representation (AER) enables DVS to exhibit advantages in redundancy suppression and data compression. At the pixel level, external differencing circuits convert analogue sensory signals into asynchronous spikes events via spike coding processing⁵⁻⁷. The main limitation of DVS is exactly the complexity of hardware circuits in each pixel. Moreover, although DVS can directly execute the spiking coding of raw image data, restoring the trajectory by DVS also requires additional circuits.

In contrast to DVS that consists of multiple transistors, capacities and photodiodes in each pixel^{6,8,9}, the pixel of NbS₂/MoS₂ sensor only contains one individual NbS₂/MoS₂ phototransistor. Such a simplified sensor structure has the potential to decrease hardware cost and power consumption. Moreover, the complete trajectory information can be registered in NbS₂/MoS₂ sensor, which effectively reduces the data transmission frequency within vision system.

In the manuscript, we intentionally select 10 time points to measure the conductance of the sensor array for demonstrating of trajectory evolution. Note that the complete trajectory information can be registered in NbS₂/MoS₂ vision sensors independently due to the time-dependent current relaxation dynamics. Therefore, visualization of complete restored trajectories merely needs a single measurement of conductance of sensor array, which minimizes the unnecessary data transmission within vision system. Moreover, the implementation of trajectories restoration by directly reading

conductance of pixels avoids the utilization of complex algorithms and additional circuits integration required in traditional CMOS vision chip for trajectory restoration, which would help to reduce the computation resources consumption of post-processing.

(26)

The robustness and generalizability of frameless paradigm with spontaneous response in array enable the trace extraction conducted in sensor terminals. To verify the detection reliability of the vision chip for sensing numerous object trajectories, we developed algorithm for motion recognition and set up corresponding datasets. 10 datasets, each containing 40 unique trajectories, were first generated using a random trajectory generating program. The starting point, ending point, and moving direction of the trajectory are all random, ensuring that the dataset is representative of a large population of possible trajectories. These datasets were then employed to simulate the

26. In a real-world visual scenes, a moving object is not necessarily a bright spot, but can be darker than the background. Moreover, there may be multiple moving objects, and the shape, speed, and/or contrast between those different objects may differ. It is obvious that the present chip is not helpful at all for separately detecting two trajectories of even two light spots even with utilizing an external computer; Thus, although the new material optical sensor proposed in this study would be of interest in the material science field, yet the proposed 10-by-10-pixel sensor chip, which requires a laser unit for stimulation and a computer for post-processing, is not applicable to the real-world vision, at least in the current form.

Reply: Thanks for the reviewer's suggestions and concerns regarding the applicability of the sensor to real-world visual scenes. We admit that the current emerging sensor design may have limitations in detecting moving objects that are not bright spots and that differ in shape, speed, and/or contrast, and it is true that our sensor array cannot rival commercial CMOS photodetectors.

Nevertheless, trajectory restoration fully realized in a highly simplified optical sensor array, which only consists of phototransistor pixels, has never been reported. Due to the intrinsic time-dependent current relaxation dynamics of NbS₂/MoS₂ phototransistor, the data generation of restored trajectory is fully processed in our NbS₂/MoS₂ vision sensor array without the need for an external processing circuit. The computer no longer needs to process massive raw data from sensors in chronological order and compute the reconstructed trajectory image. Instead, a restored trajectory image is easily obtained by reading the conductance state of the sensors, which can effectively reduce the computational burden on post-processing and data movement within vision system.

The development of individual neuro-inspired phototransistor with integrated several processing functions is still in its infancy. There are still many challenges for emerging materials in integration density, reliability, and compatibility with back-end processing units and peripheral circuits. In fact, to add processing functions in the sensor cell is a breakthrough in the recent three years, which requires breakthrough in both the device structure design and mechanism design compared with the conventional photodiode or photodetectors. Prior research works were all restricted to small size array and static image pre-processing in a limited application scenario. For example, previously reported the MoS₂/organic phototransistor sensor array shows the capability of acquisition of a pre-processed image from massive noisy optical inputs. However, the application condition is highly limited to a set of specific noisy optical inputs, where the frequency of noisy inputs is lower than that of informative inputs (*Curved neuromorphic image sensor array using a MoS₂-organic heterostructure inspired by the human visual recognition system. Nature Communications, 2020, 11: 5934*). The VO₂ phototransistor array with capability of information extraction from multi-wavelength inputs is also demonstrate in a limited application scenario, where the wavelength of target information and interference information are completely separated (*Photo-induced non-volatile VO₂ phase transition for neuromorphic ultraviolet sensors. Nature Communications, 2022, 13: 1729*).

Although real-scenario application of these sensor array still a distant prospect, as a proof-of-concept, our works makes more people understand the advantage of the neuro-inspired vision sensor based on emerging 2D materials, which can accelerate the development of novel vision chip with highly simplified configuration.

(27)

of device-to-device variance (Supplementary Figures S21-S22). Fig. 4d shows the selected 12 different rocket paths and corresponding recovered trace images. These exported results are well identical to the corresponding input trajectories from dataset. As illustrated in Fig. 4e, the detected accuracies of all 10 databases are above 92%, highlighting the stability of our trajectory detection scheme. The development of

27. This can be a supplementary material, but not a part of the main body.

Reply: Thanks for the reviewer’s suggestions. The corresponding contents have been moved to supplementary materials part.

The Figs. 4d and 4e in main text have been added in supplementary materials as Fig. S23.

Figure S23. The simulation result of trajectory restoration. **a** 12 typical recovered and corresponding input traces of rocket. The starting point, ending point, and moving direction of each trajectory are all random, ensuring that a large population of possible trajectories can be detected by NbS₂/MoS₂ sensor array. **b** Detection accuracies of 10 trajectory groups.”

(28)

NbS₂/MoS₂ **neuromorphic vision chip** for motion recognition effectively compensate
the shortcoming in sensing moving targets for emerging neuromorphic systems ⁷⁷⁻⁷⁹.

28. This is seriously doubtful, based on the present experimental results.

Reply: Thanks for the reviewer's suggestions. To guarantee accuracy, the corresponding description has been revised as below:

“The development of NbS₂/MoS₂ neuro-inspired vision sensor array with front-end trajectory extraction provides a promising strategy for sensing moving targets in emerging compact vision systems.”

Reviewer #2

The paper demonstrates the implementation of a 10x10 vision sensor fabricated with NbS₂/MoS₂ hybrid films which exhibit a persistent response to illumination. The paper proposes to use this characteristic for integrating sensing, memory, and filtering capabilities, performing trajectory detection. The paper demonstrates experimental results and compares the performance with benchmarked datasets. Just a minor remaining comment: “However, previously reported neuromorphic image sensors are largely limited to static information detecting and processing.” This sentence should be rephrased as new dynamic vision sensors are based on the detection of temporal changes in illumination. Nowadays, most neuromorphic vision sensors are based on this concept. Many functional, high-resolution dynamic vision sensor architectures have been published in the last years since the first DVS was published in 2008 as already referenced in the experimental part of the paper.

Reply: Thanks very much for the reviewer’s comments on the values of our works. We acknowledge that dynamic vision sensors architectures for motion detection have been developed for many years, and most of which are based on traditional silicon platform. To guarantee accuracy, the corresponding sentence has been revised as below:

“However, previously reported neuro-inspired image sensor based on emerging 2D materials are largely limited to static information detecting and processing.”

References

1. Narendra, P. M. & Fitch, R. C. Real-time adaptive contrast enhancement. *IEEE Trans. Pattern Anal. Mach. Intell.* **3**, 655-661 (1981).
2. Abdullah-Al-Wadud, M., Kabir, M., Akber Dewan, M. & Chae, O. A Dynamic Histogram Equalization for Image Contrast Enhancement. *IEEE Trans. Consum. Electron.* **53**, 593-600 (2007).
3. Arici, T., Dikbas, S. & Altunbasak, Y. A histogram modification framework and its application for image contrast enhancement. *IEEE Trans. Image Process.* **18**, 1921-1935 (2009).
4. Huang, S. C. & Chen, W. C. A new hardware-efficient algorithm and reconfigurable architecture for image contrast enhancement. *IEEE Trans. Image Process.* **23**, 4426-4437 (2014).
5. Culurciello, E., Etienne-Cummings, R. & Boahen, K. A. A biomorphic digital image sensor. *IEEE J. Solid-State Circuits* **38**, 281-294 (2003).
6. Lichtsteiner, P., Posch, C. & Delbruck, T. A 128X128 120 dB 15 μ s Latency Asynchronous Temporal Contrast Vision Sensor. *IEEE J. Solid-State Circuits* **43**, 566-576 (2008).
7. Posch, C., Serrano-Gotarredona, T., Linares-Barranco, B. & Delbruck, T. Retinomorphing Event-Based Vision Sensors: Bioinspired Cameras With Spiking Output. *Proc. IEEE Inst. Electr. Electron. Eng.* **102**, 1470-1484 (2014).
8. Zaghoul, K. A. & Boahen, K. Optic nerve signals in a neuromorphic chip II: Testing and results. *IEEE Trans. Biomed. Eng.* **51**, 667-675 (2004).
9. Lenero-Bardallo, J. A., Serrano-Gotarredona, T. & Linares-Barranco, B. A 3.6 μ s Latency Asynchronous Frame-Free Event-Driven Dynamic-Vision-Sensor. *IEEE J. Solid-State Circuits* **46**, 1443-1455 (2011).

REVIEWER COMMENTS

Reviewer #1 (Remarks to the Author):

The manuscript has been updated by clarifying the authors' intentions, and by correcting the authors' misconceptions about the biological nervous system. Accordingly, the text in the revised manuscript is thought to be improved. Unfortunately, however, the main results and figures have been basically unchanged. Thus, the quality of the study has been almost the same as that before the first revision. It is really difficult for this reviewer to expect that the authors will be able to improve the quality of the study anymore, and thus to believe the potential of this manuscript to be accepted for publication, at least in this journal, Nature Communications. Nevertheless, again, this reviewer provided further comments in the pdf file of the manuscript, hoping the authors may improve the quality of their study and manuscript for the future.

Reviewer #2 (Remarks to the Author):

The paper has improved. However some notions introduced in the introduction have to be corrected. Also, the paper needs a native speaker revision. It is plagued of grammatical errors.

Reviewer #3 (Remarks to the Author):

The authors have improved the manuscript according to the comments and the manuscript now is acceptable for publication.

Responses to reviewers' comments (NCOMMS-22-41687B)

We are grateful for the reviewers' comments on our work. We also thank very much for the reviewers' time and efforts to provide us with detailed and valuable comments. These suggestions are greatly helpful to us for improving the quality of this work. We have tried our best to address all the concerns raised by the reviewer point-by-point as follows, and highlighted the revised parts in red color.

Our responses to reviewer 1's comments are from **page R1 to page R24**, and the responses to reviewer 2's comments are from **page R25 to page 26**. All reviewers' comments by the built-in comment function in pdf were cited as screen capture images.

Reviewer #1

The manuscript has been updated by clarifying the authors' intentions, and by correcting the authors' misconceptions about the biological nervous system. Accordingly, the text in the revised manuscript is thought to be improved. Unfortunately, however, the main results and figures have been basically unchanged. Thus, the quality of the study has been almost the same as that before the first revision. It is really difficult for this reviewer to expect that the authors will be able to improve the quality of the study anymore, and thus to believe the potential of this manuscript to be accepted for publication, at least in this journal, Nature Communications. Nevertheless, again, this reviewer provided further comments in the pdf file of the manuscript, hoping the authors may improve the quality of their study and manuscript for the future.

Rely: Thanks very much for the reviewer's comments. We are sorry that the Reviewer has obtained this impression. We answered all questions in full and in good faith during the previous two rounds of the review. To reiterate the main criticism and our response:

1. The Reviewer requested statistics regarding the device-to-device variation of the sensor array. We responded by providing new experimental results in the first

rounds of review, and further experimental results were given in the latest round of review (comment # 18).

2. The Reviewer regarded that our work has misconceptions about the biological nervous system, as well as the visual science and neuromorphic engineering. Following the reviewer's suggestions, all the descriptions about the optical modulated current behaviors of NbS₂/MoS₂ phototransistors have been revised to avoid possible misunderstanding of the biological nervous system and neuromorphic engineering for readers in the second rounds of review. In the latest round of review, we further updated the **Fig. 1a** of main text for better analogizing between sensor array and biological retinal photoreceptor cells in response to comment #16.
3. The Reviewer raised concern for the pre-processing functions of contrast enhancement (e.g. noise filtering). We provided detailed explanations about the operation principle in previous two rounds of review. Since the current relaxation rates induced by the highlight target letter pixels with higher intensities are slower than that induced by background pixels with lower intensities, the contrast enhancement between the target letter and background can be achieved by our sensor array itself. In the latest round of review, the Reviewer requested more results of contrast enhancement performance for different light intensity combinations and relaxation time, and we hope our answer to comments #23-#25 addresses this concern.
4. The Reviewer questioned the validity of trajectory registration in sensor array. We explained the operation principle about in-sensor trajectory registration in previous two rounds of review. Benefiting from the optically-induced current response and time-dependent relaxation characteristics, the sensor array can register the sequence of stimuli independently according to the different conductance states in trajectory pixels without any external circuit integration. Although real-scenario application still a distant prospect, as a proof-of-concept, our works provides a high-efficient method to process moving object, which is expected to great facilitate the development of neuro-inspired optical sensors in field of detecting dynamic scene.

In the latest round of review, the Reviewer requested more results of trajectory registration for moving light spot with different combinations of velocity and light intensity. We think our response below (to comments #27) addresses this point.

5. The reviewer was doubtful about the simplification of vision system and the practicality in real visual environments with integration of our sensor array. We responded that we achieved multifunctional integration at the device level instead of conventional system-on-chip (SoC) integration or heterogeneous unit integration at the chip level. The functional integration at the device level can largely improve the integration density and simplify the circuit design complexity of sensor array for future high-density and low-power applications. For practicality of neuro-inspired optical sensor in real visual environments, we admit that there is still much work to be done, but the integration of multiple processing functions into the sensor cell is a breakthrough in the recent three years, which is bringing revolutionary strategy in data transfer reduction and system structure simplification. In the latest round of review, we reaffirmed main claims about the simplification of vision system and the practicality of sensors in answer to comments #10 and #24.

The new criticisms raised by the Reviewer are point-by-point responded below:

(1)

Neuro-inspired ~~vision~~ sensor array for high-accuracy static

1. Use "optical" instead of "vision" to be suited for this study, and be consistent with the authors' response to the review comments as in the rebuttal letter, and be consistent with others in the main text.

Reply: Thanks very much for the reviewer's comments. The term "vision" has been replaced with "optical" according to the reviewer's suggestion.

(2)

Neuro-inspired ~~vision~~ sensor array for high-accuracy static

2. Insert "usable" here to be more precise.

Reply: Thanks very much for the reviewer’s comments. **Professional language-editing editor of Springer Nature** suggested that the title of manuscript should be revised with “**Neuro-inspired optical sensor array with high-accuracy static image recognition and dynamic trace extraction**”.

Neuro-inspired vision system holds great promises to address the growing demand
(3)
for mass data processing in the Internet of Things (IoT) era. In addition to the capability

3. Is this statement really relevant to, or an important concept in, this study?

Reply: Thanks very much for the reviewer’s comments. The neuro-inspired optical sensor array can perform multiple functions including sensing and processing, which is highly required for the edge computing. For more accuracy, we revised the corresponding description in main text:

“Neuro-inspired vision systems hold great promise to address the growing demands of mass data processing for edge computing.”

(4)

convolutional neural network (CNN) with a high image recognition accuracy of about 90%. More importantly, ~~detection and~~ in-sensor ~~extraction~~ of moving ~~object~~ trajectory

4. This should be removed, The value "90%" has no meaning as this value depends on the CNN architecture and its training and optimization methods, none of which have been thoroughly investigated in this study, and been employed for demonstration purpose only. Thus, it is not appropriate to mention it especially in the abstract.

Reply: Thanks very much for the reviewer’s comments. The value “90%” has been deleted according to the reviewer’s suggestion.

The related description has been revised in main text:

“A neuro-inspired optical sensor array with 10×10 NbS₂/MoS₂ phototransistors enabled the highly integrated functions of sensing, memory, and contrast enhancement

capabilities for static images, which benefits convolutional neural network (CNN) with a high image recognition accuracy.”

(5)

90%. More importantly, ~~detection and~~ in-sensor ~~extraction~~ of moving ~~object~~ trajectory

5. As already described in my second review comments, it is not appropriate to use the word "detection" for the in-sensor function proposed by this study.

Reply: Thanks very much for the reviewer’s comments. We replaced the term “**detection and in-sensor extraction**” with “**in-sensor registration**” according to the reviewer’s suggestion.

(6)

90%. More importantly, ~~detection and~~ in-sensor ~~extraction~~ of moving ~~object~~ trajectory

6. Use "registration" instead of "extraction" to be suited for the in-sensor function proposed in this study, and be consistent with the authors' response to the review comments as in the rebuttal letter, and be consistent with others in the text.

Reply: Thanks very much for the reviewer’s comments. The term “**extraction**” has been replaced with “**registration**” according to the reviewer’s suggestion.

(7)

90%. More importantly, ~~detection and~~ in-sensor ~~extraction~~ of moving ~~object~~ trajectory

7. Using 'object' is not appropriate as the input was not a visual image/video, but a single bright point of light with a single motion trajectory. Thus, remove "object" throughout the text, and use "light point", for example, to describe the fact accurately.

Reply: Thanks very much for the reviewer’s comments. The term “**object**” has been replaced with “**light spot**” for accuracy.

(8)

were experimentally implemented, with a high restoration accuracy of over 92%. The

8. delete "with", and insert, for example "so that the post-processing can yield".

Reply: Thanks very much for the reviewer's comments. By combination of the suggestion from reviewer and professional language-editing editor of Springer Nature, the related sentence has been revised as below:

"More importantly, in-sensor trajectory registration of moving light spots was experimentally implemented such that the post-processing could yield a high restoration accuracy."

(9)

were experimentally implemented, with a high restoration accuracy of over 92%. The

9. This should be removed. The value "92" has no meaning as this value depends on the evaluation method, which has not been thoroughly investigated in this study. Besides, the readers cannot decide/know whether this value is high enough or low. Thus, it is not appropriate to mention it especially in the abstract.

Reply: Thanks very much for the reviewer's comments. The value "92%" has been deleted according to the reviewer's suggestion.

(10)

demonstration of the neuro-inspired vision sensor array enables simplification of hardware complexity, providing an attractive platform for the implementation of high-

10. The authors did not demonstrate it at all in this study. Without a quantitative comparison with previously proposed hardware set-ups of so-called "artificial vision system", it is impossible for the readers to decide/know whether the set-ups of authors' vision system is simple enough or complex. Besides, the set-up shown in Fig. S1 appears

to be huge in size and complex with multiple external equipments including the laser unit, at least when compared with the CMOS-based artificial vision system, for example DVS, as in the authors references.

Reply: Thanks very much for the reviewer's comments. Our work aims to construct a front-end **sensor array** with both image sensing and several pre-processing capabilities, instead of a complete artificial vision system. As the reviewer mentioned, the laser unit shown in **Fig. S1** of manuscript is not part of our sensor array, and is included to illustrate the set-up of array measurement, which is a routine way to simulate the projection of external images onto a sensor array through laser point illumination (*Bioinspired in-sensor visual adaptation for accurate perception. Nature Electronics, 2022, 5:84-91; Photo-induced non-volatile VO₂ phase transition for neuromorphic ultraviolet sensors. Nature Communications, 2022, 13: 1729*).

Notably, the pre-processing capabilities of the sensor array originate from the unique and non-linear time-dependent current relaxation dynamics in each sensor cell (corresponding to each pixel), which is distinguishable from the conventional photodiode cell of a CMOS image sensor and can largely improve the integration density and simplify the circuit design complexity of sensor array. As shown in **Table R1** (**Table S2** in Supplementary Information of second revised version), we compared our sensor array with the pixel array of commercial DVS in terms of operation form, pixel complexity, sensor area and applications. The pixel of DVS consists of many transistors, capacitors and photodiodes, while the pixel of our sensor array only contains single NbS₂/MoS₂ phototransistor. It is clear that our optical sensor demonstrates superiority in circuit complexity compared with DVS, and the integration of multiple processing functions into a simplified architecture provides a promising path for designing compact vision system in the future.

Table R1. Comparison of NbS₂/MoS₂ sensor array with DVS.

	This work	Delbruck et al. ¹	Boahen et al. ²	Barranco et al. ³
Operation form	Analog spatial-temporal current	Event-induced spike signals		
Pixel complexity	1 phototransistor	26 transistors, 3 caps, 1 photodiode	38 transistors, 1 photodiode	> 15 transistors, 1 photodiode
Sensor area	1×0.6 mm²	6×6 mm ²	3.5×3.3 mm ²	5.5×5.6 mm ²
Application	Static and dynamic scenes	Dynamic scenes		

In order to be more accurate, we also revised related description according to reviewer's suggestion:

“Our neuro-inspired optical sensor array could provide an attractive platform for the implementation of high-performance artificial vision systems.”

Table R1 was also added in Supplementary Information as Table S3.

(11)

signals sensing and image processing, which demonstrates highly simplified hardware system compared to the conventional retinomorphic chip based on CMOS platform.

11. Rephrase this. For example, "which is expected to offer a highly..."

Reply: Thanks very much for the reviewer's comments. According to reviewer's suggestion, and description has been revised as below:

“They are considered as ideal material candidates to develop neuro-inspired vision systems with the capabilities of visual signal sensing and image processing, which is expected to offer a highly simplified hardware system compared to the conventional retinomorphic chip based on the CMOS platform.”

combining with a plano-convex lens, the sensor can acquire the pre-processed image

(12)

from a set of noisy optical input without redundant and communication. Park *et al.*

12. English? This part can be removed.

Reply: Thanks very much for the reviewer's comments. The related part has been deleted according to reviewer's suggestion.

(13)

static information detecting and processing. The absence of ~~trajectory extraction~~ for moving targets stymies the development of fully hardware-implemented **machine**

13. Consider to use "computation" instead of "trajectory extraction".

Reply: Thanks very much for the reviewer's comments. The term “**trajectory extraction**” has been replaced with “**computation**” according to reviewer's suggestion.

(14)

~~vision~~ **sensor array with 10×10 NbS₂/MoS₂ devices** enables the ~~highly-integrated~~ functions including sensing, memory, and **contrast enhancement** for static images,

14. The concept, "integration", is not appropriate here. Consider to use "multiple" instead of "highly-integrated".

Reply: Thanks very much for the reviewer's comments. The term “**highly-integrated**” has been replaced with “**multiple**” according to reviewer's suggestion.

(15)

dynamic ~~objects~~ was also experimentally demonstrated, ~~showing~~ a high restoration

15. Consider to use "offering" not "showing".

Reply: Thanks very much for the reviewer's comments. By combination of the suggestion from reviewer and professional language-editing editor of Springer Nature, the term “**showing**” has been replaced with “**providing**” according to reviewer's suggestion.

(16)

Fig. 1a depicts a schematic illustration of main parts in human visual system

16. Fig. 1a should be removed, or at least moved to the supplementary materials, since the biological vision system is clearly distant from the optical sensor array developed in this study. Thus, the figure does not help explain the contribution of the present study.

The functionality of the sensor array developed in this study can be explained by the slow response kinetics of the sensor itself. And this may be analogous to the slow kinetics of light-induced responses of the rod-type photoreceptor cells in biological retinas. Indeed, it is conceivable that, although the time scale of processing may be different from that of the proposed sensor array, some functions demonstrated in this manuscript (e.g. trajectory registration for the point of light under the dark background) are observable at the level of photoreceptor-cell layer in biological retinas. And, even at the level of second-order neurons in biological retinas (i.e. bipolar-cell or horizontal-cell layer), much more sophisticated image/visual processing than those demonstrated in this manuscript are performed. This reviewer suspects that the authors are unaware of such basic facts about biological retinas.

In any case, the inspiration must have come from the biological retinal photoreceptor cells, but not from the whole retina or whole visual system in humans. Thus, Fig. 1a is not suitable as a main figure.

Reply: Thanks very much for the reviewer's comments. As the reviewer discussed, the biological vision system illustrated in **Fig. 1a** of original manuscript is distant from the optical sensor array developed in our study. Therefore, we revised **Fig. 1a** with **Fig. R1** according to the suggestion, in which optically-induced current response and time-dependent relaxation characteristics of optical sensor array are analogous to the kinetics of light-induced responses of the photoreceptor cells in biological retinas.

Fig. R1. Implementation of a neuro-inspired optical sensor array for the pre-processing of both static and moving targets, in which optically-induced current response and time-dependent relaxation characteristics of optical sensor array is analogous to the kinetics of the light-induced responses of the photoreceptor cells in biological retinas.

Fig. R1 is updated **Fig. 1a** in main text.

(17)

trajectory ~~tracing~~ of dynamic objects. **Fig. 1c** shows a digital photographic image of the

17. Use "registration" not "tracing".

Reply: Thanks very much for the reviewer's comments. The term "tracing" has been replaced with "registration" according to reviewer's suggestion.

(18)

two characteristic timescales, τ_1 and τ_2 , are estimated to be 8.24 s and 0.21 s, respectively. The τ_1 is an order of magnitude larger than τ_2 , indicating a successful

18. Need the statistics across multiple measurements in a given sample, and the statistics across different samples.

Reply: Thanks for the reviewer's suggestions. Statistics of modulation ratio (A_2/A_1) are provided across 10 measurements. As shown in **Fig. R2**, the results still can be fitted well by a double-exponential function. The statistic values of two characteristic timescales (τ_1 and τ_2) for different device samples are also given in **Fig. R3**, the mean

values and corresponding standard deviations for τ_1 and τ_2 are calculated to be about 3.78 ± 2.55 and 0.18 ± 0.08 s.

Fig. R2 Modulation ratio of conductance as a function of interval time (Δt). The inset figure plots the time-resolved photocurrent of device at two successive light pulses with the pulse width of 500 ms. Each A_2/A_1 ratio corresponding to a certain pulse interval contains 10 measurements.

Fig. R3 Statistic values of the characteristic constants (τ_1 and τ_2) for different pixels. **a** Relaxation time constant (τ_1) statistical distribution with mean values of 3.78 s and corresponding standard deviation of 2.55 s. **b** Relaxation time constant (τ_2) statistical distribution with mean values of 0.18 s and corresponding standard deviation of 0.08 s.

Fig. R2 is updated as **Fig. 2e** in main text, and **Fig. R3** has been added in Supplementary

Information as **Fig. S11**. Related description has been updated in main text:

“The two characteristic timescales, τ_1 and τ_2 , are estimated to be 3.78 s and 0.18 s, respectively (Supplementary Figure S11).”

stimuli process, consistent with previously reported neuro-inspired optically stimulated devices ⁶⁰⁻⁶⁴. The dynamic tunable processes of **conductance** plasticity were also

19. If this is consistent with the previous devices, then the question should be "what is the novelty or qualitative difference of the proposed device compared with previous ones?"

Are these time constants significantly larger than those in previous devices? If these time constants are comparable to those in previous devices, then basically (or qualitatively) the memory-like behavior, contrast enhancement, and trajectory registration could be realized also with the previous devices, and the difference between the previous and present studies may be whether those functions are experimentally demonstrated or not. If this is the case more or less, the novelty of the present study is only in the difference in materials/methods used for fabricating the devices, but not of fundamental developments in the field of neuro-inspired optical sensors.

Reply: Thanks for the reviewer's suggestions. As shown in **Table R2**, the time constants of our device show superiority to those of previously reported neuro-inspired optoelectronic devices. In addition, we have to clarify that the "pulse-interval dependent plasticity" is just a basic characteristic of neuro-inspired optical devices, which can be evaluated by time constant. **The novelty of this study lies in the integration of processing functions for both static and motion objects in a highly simplified optical sensor array without external processing circuits.** In addition, in contrast to previous works that only reported certain pre-processing functions for static images, we innovatively proposed an algorithm at device level based on kinetic of photocurrent in NbS₂/MoS₂ optical device and deployed it for motion scenarios successfully.

In summary, the conception of in-sensor trajectory registration provides a high-efficient method to process moving object, which is expected to great facilitate the development of neuro-inspired optical sensors in field of sensing and pre-processing dynamic scene.

Table R2. Comparison of time constants with previous optical devices.

Materials	Time constants (τ_1/τ_2)	Reference
GaO _x	0.28/0.029 s	4
PTHF percentage	0.57/0.072 s	5
P3HT/CsPbBr ₂ I	0.011/0.0003 s	6
MAPbI ₃ /IZO	3.43/0.36 s	7
IGZO/CdS	3.5/0.1 s	8
NbS ₂ /MoS ₂	3.78/0.18 s	Our work

Table R2 has been added in Supplementary Information as **Table S1**.

The evolution of short-term relaxation (STR) and long-term relaxation (LTR) is **(20)** revealed by applying different optical pulses number to phototransistor. As shown in

20. English?

Reply: Thanks very much for the reviewer's comments. The related sentence has been revised as below:

"The evolution of short-term relaxation (STR) and long-term relaxation (LTR) is accomplished by applying different optical pulses to phototransistor."

which we can identify a minimum value of 0.42 pJ with a pulse duration of 0.01 s. **(21)** value is comparable to those of previously reported state-of-art optical neuro-inspired devices ⁷¹⁻⁷⁶, indicating the great potentials of NbS₂/MoS₂ phototransistor for low-

21. If this is comparable to previous devices, then the question should be "what is the novelty of the present device compared with previous ones?"

Reply: Thanks very much for the reviewer's comments. As mentioned in response to comment #19, the hardware implementation of neuron-inspired vision sensor and algorithm deployment for dynamic objects are challenging, and previous devices with low energy consumption only demonstrate simple functions, such as signals detection and static image processing. Our work demonstrated an effectively in-sensor trajectory

registration strategy without external computing circuits, which is a breakthrough for application of the emerging device. Moreover, attributing to the van der Waals integration of metallic NbS₂ component into MoS₂ counterpart, our sensor array can operate at lower voltage bias than previous device. Therefore, our device could integrate multiple functions with comparable energy consumption, including detecting, memory, image enhancement and in-sensor trajectory registration, which can effectively extend the functionality of emerging neuro-inspired optical device.

(22)

light-tunable **conductance relaxation speed** ~~above~~ enables the contrast enhancement of

22. Describe the quantitative definition of so-called "contrast".

Reply: Thanks very much for the reviewer's comments. The term "**contrast**" refers to the ratio of collected photocurrent between illuminated target pixels and background pixels.

The related description has been updated in main text:

"Here, the contrast is defined as the ratio of the collected photocurrent between illuminated target pixels and background pixels."

(23)

target letter "H" stimulated at 3.1 $\mu\text{W } \mu\text{m}^{-2}$ and a **background** letter "T" at 1.6 $\mu\text{W } \mu\text{m}^{-2}$

23. It is believed that these values were intentionally chosen so that the contrast between the so-called "target" and "background" become high enough after a few hundred seconds. Thus, there seems no reasoning for those two values. This is a kind of trick, but definitely not a science nor engineering.

Reply: See below responses to comment #24.

(24)

results suggested the contrast enhancement between target image and background can be hardware implemented base on our NbS₂/MoS₂ sensor array without using any external circuits.

24. This is not true. In order to make the so-called "contrast enhancement" to work properly, one must select and set a good combination of the laser powers for the target and the background, and select and set the timing to readout the image (dependently on the pre-set laser power combination) so that the contrast between the target and background is high enough for making the post-processing to work. These parameter selections and settings for inputing images to device, and for reading out the pre-processed images from the device requires intentional manual operations (or external computerized system). Thus, the contrast enhancement cannot be implemented without using any external circuits, and much more importantly, does not work for real visual environments.

Reply: Thanks very much for the reviewer's comments of #23 and #24. In the original manuscript, only one set of input image (contrast of 1/0.5 and relaxation time of 150 s) was investigated, and the experimental implementation of contrast enhancement is not enough. As the reviewer suggested, the comparison between the different sets of light intensity combinations was supplied. **Fig. R4** shows the variation of the output current ratio with relaxation time for three typical input light intensity ratios. The input light intensity ratios of 1/0.8 (1.25), 1/0.6 (1.67) and 1/0.5 (2) are enlarged to 1/0.5 (2), 1/0.4 (2.5) and 1/0.33 (3) after light stimuli finished, and monotonically increase to 1/0.42 (2.4), 1/0.27 (3.6) and 1/0.22 (4.5) after relaxation time of 150 s. **The results demonstrate that our sensor array can enhance the image contrast in a wide range of light intensity combinations and relaxation times.**

Fig. R4 Current ratio varied with relaxation time for input light intensity ratio of 1/0.8, 1/0.6 and 1/0.5. The dash lines indicate the original light intensity ratio.

Since the development of neuro-inspired vision sensors based on emerging materials and emerging in-sensor computing is still at its early stages, the vision array with in-sensor computing capabilities usually lacks standard ancillary components comparable to those found in commercial CMOS vision chips, including lenses for focusing panoramic scenes onto sensor array and peripheral circuits for reading conductance state of sensor array. Using laser point illumination with a certain range of intensity to simulate the projection of images onto sensor array is a widely alternative method for studying the optical sensor (*Bioinspired in-sensor visual adaptation for accurate perception. Nature Electronics, 2022, 5:84-91; Photo-induced non-volatile VO₂ phase transition for neuromorphic ultraviolet sensors. Nature Communications, 2022, 13: 1729*). **We have to emphasize that the photocurrent of each phototransistor in sensor array is faithfully read out by sourcemeter units, and the acquired data do not need to be post-processed by any external computerized system.**

We admit that there is still much work to be done before neuro-inspired vision sensors based on emerging devices can work for real visual environments, but the integration of multiple processing functions into the sensor cell is a breakthrough in the recent three years, which is bringing revolutionary strategy in data transfer reduction and system structure simplification.

Figs. R4 has been added in Supplementary Information as **Fig. S18**. Related description has been updated in main text:

“Similar increases in the output photocurrent ratio was also observed for input light intensity ratios of 1/0.8 and 1/0.6 (Supplementary Figure S18).”

“These results indicate that the contrast enhancement between the target image and background can be hardware implemented based on our NbS₂/MoS₂ sensor array without using any external computing circuits.”

(25)

S20). As shown in **Fig. 3e**, the pre-processing of sensor array evidently reduces the noises and improves the image quality, and contributes to the enhanced contrast between the target letter and background with sharpened details. For comparison, an artificial vision system without the pre-processing function was also trained and tested with the same dataset and procedure. The recognition accuracies of two vision systems are given in **Fig. 3f**. One can see that a remarkable improvement of recognition rate is achieved from 78% to 90% after the integration of pre-processing module. It means that with the integration of NbS₂/MoS₂ neuro-inspired sensor array for image pre-processing, the background noise signals of input visual information can be effectively smoothed for high-accuracy static image recognition.

25. If authors wish to keep a result of the CNN experiments, then the comparison between the two different sets of input images (i.e. raw datasets and enhanced datasets) is not enough at all. The authors need to show quantitative relationships between the contrast value and the recognition accuracy by using different combinations of laser light intensities for the target and background (and/or by using the image sets obtained at different time points of reading the preprocessed images out of the device).

Reply: Thanks very much for the reviewer’s comments. We further investigated the relationships between the contrast value and the recognition accuracy by using different combinations of laser light intensities for the target and background. As shown in **Fig. R5**, with the relaxation time of the sensor array fixed at 150 s, we compared the two cases with the normalized light intensity ratio of target and background of 1/0.55 and 1/0.63, respectively. For the input images with different contrast, the output image from NbS₂/MoS₂ sensor array exhibits reduced noise and sharpened details between target letter and background, resulting in much higher recognition accuracy for convolutional neural network (CNN).

Fig. R5 Contrast enhancement performances of sensor array and recognition accuracy of CNN for input images with different contrast of 1/0.55 and 1/0.63. **a-b** Comparison of image before and after pre-processing. **c-d** Recognition accuracies of CNN with and without pre-processing.

To study the effect of relaxation time on the image enhancement performance, we selected two typical reading time points while keeping the input image contrast fixed at 1/0.5 (**Fig. R6**). In the first case, the current of sensor array is read immediately after light illumination finished. In the second case, the current of sensor array is read with relaxation time of 75 s, half of the relaxation time adopted in main text. The NbS₂/MoS₂ sensor array evidently shows robustness and reliability in reducing the noises within relaxation range from 0 s to 150 s, which contributes to the remarkable improvement of recognition accuracy for noisy image.

Fig. R6 Contrast enhancement performances of sensor array and recognition accuracy of CNN for relaxation time of 0 s and 75 s. **a-b** Comparison of image before and after pre-processing. **c-d** Recognition accuracies of CNN with and without pre-processing.

Figs. R5 and R6 have been added in Supplementary Information as **Figs. S23 and S24**.

Related description has been updated in main text:

“Fig. 3e and Supplementary Figures S2-S24 show the performance of the artificial vision system for different combinations of input light intensity and relaxation time. The results of the output image dataset indicated that the pre-processing of the sensor array evidently reduced the noise, improved the image quality and contributed to the enhanced contrast between the target letter and background with sharpened details.”

(26)

trajectory of moving objects. As shown in Fig. 4a, front-end motion detection is generally realized based on CMOS vision chip, which captures visual information framed-by-frame at a predetermined rate. The separated memory and processing of frame data at sensor-level or pixel level would cause the unnecessarily inflating data^{78,79}. Dynamic vision sensor (DVS) was also developed based on CMOS platform, which can asynchronously output in the form of an event stream with address-event representation (AER)^{18-21,24,80}. Although DVS can be applied for real-time and high temporal resolved motion detecting, obtained raw data or event stream still need to be successively transmitted to external memory unit for post-processing, thus posing challenges for the hardware simplification of vision system (Supplementary Table S2).

26. This part of description and the corresponding figures [i.e. two subpanels on the left-most (rocket cartoon) and the upper (CMOS-based system) in Fig. 4a] must be irreverent to be used as a main text and figure. If the authors wish to keep these, please consider to move those to the Discussion section or Supplementary materials.

Reply: Thanks very much for the reviewer's comments. Fig. 4a of original manuscript has been revised with Fig. R7 according to reviewer's suggestion. The two subpanels on the left-most (rocket cartoon) and the upper (CMOS-based system) in original Fig. 4a have been updated as Fig. R8, which is added in Supplementary materials as Fig. S25.

The related description of conventional CMOS vision chip and DVS had been supplied in main text according to reviewer#2's suggestion. Therefore, we think that is would be better if the related description can be kept.

Fig. R7 Schematic illustration of in-sensor trajectories registration.

Fig. R8 Schematic illustration of front-end motion detection realized by frame-based conventional CMOS vision chip.

Fig. R7 is updated as Fig. 4a in main text, and Fig. R8 has been added in Supplementary Information as Fig. S25.

(27)

end trajectory extraction provides a promising strategy for sensing moving targets in emerging compact vision systems⁸¹⁻⁸³.

27. Since the in-sensor trajectory registration are owing to the slow kinetics (memory-like behavior) of the optical sensor itself, the velocity of motion for registering its trajectory must be limited within a certain range. The authors need to investigate the operating range of the combination of motion velocity and light intensity for the trajectory registration to work properly. Otherwise no strategy can be made.

Reply: Thanks very much for the reviewer's comments. Only one combination of motion velocity (40 $\mu\text{m/s}$) and light intensity (3.1 $\mu\text{W}/\text{um}^2$) was investigated in the second revised manuscript. According to reviewer's suggestion, we provide the

trajectory registration performance of the sensor array with different combinations of motion velocity and light intensity. As shown in **Figs. R9a** and **b**, our sensor array can stably execute in-sensor trajectory registration, when velocity of light spot is increased from 20 $\mu\text{m/s}$ to 80 $\mu\text{m/s}$. The similar result was also observed with the light power density ranging from 1.6 $\mu\text{W}/\mu\text{m}^2$ to 4.1 $\mu\text{W}/\mu\text{m}^2$ (**Figs. R9c** and **d**).

Fig. R9 Performance of in-sensor trajectory registration for different velocity and light intensity of moving light spot. **a-b** NbS₂/MoS₂ sensor array for slower and faster trajectory registration. **c-d** NbS₂/MoS₂ sensor array for lower light intensity and higher light intensity trajectory registration (left panel shows real-time photocurrent of pixels located at trajectory, right panel shows current mapping of trajectory registration).

Fig. R9 has been added in Supplementary Information as **Fig. S26**. Related description has been updated in main text:

“Additionally, the sensor array also showed the stable trajectory registration performance for moving light spots with different combinations of velocity and light intensity (Supplementary Figure S26).”

Reviewer #2

The paper has improved. However, some notions introduced in the introduction have to be corrected. Also, the paper needs a native speaker revision. It is plagued of grammatical errors.

(1)

Advanced artificial vision system is essential for the development of applications

1. systems are essential for

Reply: Thanks very much for the reviewer's comments. **We have revised our manuscript through the professional language-editing service from Springer Nature.** The related sentence has been revised as below:

“Advanced artificial vision systems are essential for the development of applications such as intelligent homes, autopilot vehicles, and video content analysis.”

(2)

memories for storing information, and processors for performing ~~neuromorphic~~

(3)

computation. The ~~coarsely parallel~~ structure of the von Neumann computer faces great

2. common digital vision systems doing processing on the output of conventional image sensors are not considered neuromorphic at all.

Reply: Thanks very much for the reviewer's comments. The term “**neuromorphic**” has been deleted.

(2)

memories for storing information, and processors for performing ~~neuromorphic~~

(3)

computation. The ~~coarsely parallel~~ structure of the von Neumann computer faces great

3. The von Neumann architecture is not parallel at all. It is based on sequential processing.

Reply: Thanks very much for the reviewer's comments. The related sentence has been revised as below:

“The sequential processing paradigm of the von Neumann computer faces great challenges of energy consumption, redundancy, and latency for processing explosive growth of visual data.”

References:

1. Lichtsteiner, P., Posch, C. & Delbruck, T. A 128X128 120 dB 15 μ s Latency Asynchronous Temporal Contrast Vision Sensor. *IEEE J. Solid-State Circuits* **43**, 566-576 (2008).
2. Zaghoul, K. A. & Boahen, K. Optic nerve signals in a neuromorphic chip II: Testing and results. *IEEE Trans. Biomed. Eng.* **51**, 667-675 (2004).
3. Lenero-Bardallo, J. A., Serrano-Gotarredona, T. & Linares-Barranco, B. A 3.6 μ s Latency Asynchronous Frame-Free Event-Driven Dynamic-Vision-Sensor. *IEEE J. Solid-State Circuits* **46**, 1443-1455 (2011).
4. Zhang, Z. *et al.* In-sensor reservoir computing system for latent fingerprint recognition with deep ultraviolet photo-synapses and memristor array. *Nat. Commun.* **13**, 6590 (2022).
5. Ji, X. *et al.* Mimicking associative learning using an ion-trapping non-volatile synaptic organic electrochemical transistor. *Nat. Commun.* **12**, 2480 (2021).
6. Ren, Y. *et al.* Synaptic plasticity in self-powered artificial striate cortex for binocular orientation selectivity. *Nat. Commun.* **13**, 5585 (2022).
7. Lee, T. J. *et al.* Realization of an Artificial Visual Nervous System using an Integrated Optoelectronic Device Array. *Adv. Mater.* **33**, e2105485 (2021).
8. Kwon, S. M. *et al.* Large-Area Pixelized Optoelectronic Neuromorphic Devices with Multispectral Light-Modulated Bidirectional Synaptic Circuits. *Adv. Mater.* **33**, e2105017 (2021).

REVIEWERS' COMMENTS

Reviewer #1 (Remarks to the Author):

The manuscript has been improved by adding more experiments according to the suggestions in the second review. Although the new experimental results are provided as supplementary materials, not as the main results or figures, those results strengthen the main results and figures. Thus, the revised manuscript is considered improved to be acceptable for publication in a certain journal. However, it is still hard for this reviewer to believe that this manuscript is suitable for publication in the journal, Nature Communications since this study appears to contribute little to fundamental developments in the field of neuro-inspired optical sensors, but rather primarily to its application to image pre-processing. In addition, some of the essential experiments/results are still missing from the revised manuscript.

Nevertheless, again, this reviewer provided further comments in the pdf file of the manuscript, hoping the authors may further improve the quality of their study and manuscript for the future.

Reviewer #2 (Remarks to the Author):

The paper presents a new array of neuroinspired photosensitive devices that present photopersistent effects. They demonstrate that the preprocessing of their devices improve the object recognition. The paper has improved over previous version and author have clarified previous imprecise comments. The writing of the paper has also been improved.

Responses to reviewers' comments (NCOMMS-22-41687C)

We are grateful for the reviewers' comments on our work. We also thank very much for the reviewers' time and efforts to provide us with detailed and valuable comments. These suggestions are greatly helpful to us for improving the quality of this work. We have tried our best to address all the concerns raised by the reviewer point-by-point as follows, and highlighted the revised parts in red color.

Our responses to reviewer 1's comments are from **page R1** to **page R18**, and the reviewers' comments by the built-in comment function in pdf were cited as screen capture images.

Reviewer #1:

The manuscript has been improved by adding more experiments according to the suggestions in the second review. Although the new experimental results are provided as supplementary materials, not as the main results or figures, those results strengthen the main results and figures. Thus, the revised manuscript is considered improved to be acceptable for publication in a certain journal. However, it is still hard for this reviewer to believe that this manuscript is suitable for publication in the journal, Nature Communications since this study appears to contribute little to fundamental developments in the field of neuro-inspired optical sensors, but rather primarily to its application to image pre-processing. In addition, some of the essential experiments/results are still missing from the revised manuscript.

Nevertheless, again, this reviewer provided further comments in the pdf file of the manuscript, hoping the authors may further improve the quality of their study and manuscript for the future.

Rely: Thanks very much for the reviewer's comments. It is a pity that the Reviewer has obtained this impression. The work aims to construct a front-end sensor array monolithically integrated with both image sensing and several pre-processing capabilities (e.g., contrast enhancement and trajectory registration) using highly

simplified circuits. We wish that the study could provide inspiration for the design of multi-functional optical sensors with highly integrated density and low circuit complexity, aiming to facilitate the full implementation of in-sensor processing for both static and dynamic scenes in neuro-inspired devices based on emerging materials.

In aspect of architecture design, we demonstrate a promising strategy of multi-functional integration at device level. The pre-processing functionality of proposed sensor array originates from the unique and non-linear time-dependent current relaxation dynamics in each sensor cell (corresponding to each pixel), which is distinguishable from the conventional photodiode cell of a CMOS image sensor. As a result, we could achieve multi-functional integration at the device level instead of conventional system-on-chip (SoC) integration or heterogeneous unit integration at the chip level. It can largely improve the integration density and simplify the circuit design complexity for future high-density and low-power applications. Especially, for power-limited and size-limited applications such as drones, our array exhibits great advantages in terms of density, low power, and low latency since data transfer can be considerably reduced between the sensor and processing units.

For practical applications, our sensor array expands the scope of front-end pre-processing functions from static images to dynamic trajectories. The contrast enhancement between the target letter and the background can be achieved by our sensor array itself, relying on the non-linear intensity-dependent photocurrent relaxation rates. Moreover, in contrast to previous work that only reported certain pre-processing functions for static images, we proposed an algorithm at the device level based on the kinetics of photocurrent in NbS₂/MoS₂ optoelectronic devices and deployed it for motion scenarios successfully. Benefiting from the optically induced current response and time-dependent relaxation characteristics, the sensor array can register the sequence of stimuli independently according to the different conductance states in trajectory pixels without any external circuit integration. The conception of in-sensor trajectory registration provides a highly efficient method to process the motion scenes, which is expected to greatly facilitate the development of neuro-inspired optical sensors in field of sensing and pre-processing dynamic scenes.

To guarantee an accurate description in the manuscript, we have revised some figures (Fig. 1a, Fig. 2c, Fig. 3d, Supplementary Fig. S20, S23-S26) and updated the results of CNN recognition accuracy with dependence on the number of training epochs according to the reviewer's suggestions. The new criticisms raised by the Reviewer are point-by-point responded as below:

(1)

Neuro-inspired optical sensor array with high-accuracy static image recognition and dynamic trace extraction

1. Regardless of the English editing, its is considered that the "with" must be inconsistent with the claim of this study. Rather than "with", "for"/"useful for"/"usable for" is suitable for this study. This is not just an English title name, but a title that conveys the claim/message of the research.

Reply: Thanks very much for the reviewer's comments. The term “with” has been replaced with “for” according to the reviewer's suggestion.

(2)

Neuro-inspired optical sensor array with high-accuracy static image recognition and dynamic trace extraction

2. If the authors wish to keep "with", and make the title to represent the claim as accurately as possible, then 1)"high-accuracy" should be deleted, 2) "image recognition" should be replaced with, for example, "contrast enhancement", and 3) "trace extraction" should be replaced with, for example, "trajectory registration".

Reply: Thanks very much for the reviewer's comments. For more accuracy, the term “with” has been replaced with “for”. The revised title of the manuscript as below:

“Neuro-inspired optical sensor array for high-accuracy static image recognition and dynamic trace extraction.”

The human visual system **mainly consists of the** retina, optic nerve, lateral geniculate nucleus (LGN) and visual cortex. Light signals are **initially** converted by photoreceptor

(3)

3. Please use "cortices" instead of "cortex"

Reply: Thanks very much for the reviewer's comments. The term "**cortex**" has been replaced with "**cortices**" according to the reviewer's suggestion.

redundant and unstructured visual data^{11,45,46}. **Subsequently, the** LGN receives and transmits information from **the** retina to **the brain** cortex for the cognition of visual

(4)

4. please delete "brain"

Reply: The term "**brain**" has been deleted according to the reviewer's suggestion.

stimuli⁴⁷⁻⁵¹. The functions in terms of noise filtering, edge enhancement, and motion detection of **the** retina **provide the** brain cortex with high efficiency for high-level

(5)

5. please delete "brain"

Reply: Thanks very much for the reviewer's comments. The term "**brain**" has been deleted according to the reviewer's suggestion.

(6)

As shown in **Fig. 1a**, inspired by the pre-processing functions of the retina, we

6. In Fig. 1a left-most, red/green color code is opposite; the red cell in the illustration is rod-type and green one is the cone-type photoreceptors.

Reply: Thanks very much for the reviewer's comments. As the reviewer discussed, the color code of rod-type and cone-type photoreceptors in **Fig. 1a** is opposite. Therefore, we revised **Fig. 1a** with **Fig. R1** according to the suggestion, in which the mistake of color code has been corrected.

Fig. R1. Implementation of a neuro-inspired optical sensor array for the pre-processing of both static and moving targets, in which the mistake of color code has been corrected.

Fig. R1 is updated as part of **Fig. 1** in main text.

(7)

Notably, the device shows a pronounced PPC effect at a negative gate voltage (**Fig.**

2c). The photocurrent rapidly increases when the light is turned on, showing a fast

7. In Fig. 2c upper-most, the time axis for "Laser pulse" is rightward shifted.

Reply: Thanks very much for the reviewer's comments. As the reviewer discussed, the time axis between "laser pulse" and "photocurrent" is shifted in **Fig. 2c** in the manuscript. As shown in **Fig. R2**, we revised **Fig. 2c** to ensure that the time axes of all variables are aligned.

Fig. R2. Photoreponse of the device to four light ON/OFF cycles at $V_{gs}=-50$ V, in which the time axes of all variables are revised to align.

Fig. R2 is updated as part of **Fig. 2** in main text.

respectively (**Supplementary Figure S11**). τ_1 is an order of magnitude larger than τ_2 ,
(8)
indicating a successful stimuli process, consistent with previously reported neuro-
inspired optically stimulated devices (**Supplementary Table S1**)⁶⁰⁻⁶⁴. The dynamic

8. Based on Table S1, the time constants in the present sensor array are almost the same as those in Reference [63] and [64]. It is hard to believe that those time constants are superior to previous studies despite of the authors statement in their rebuttal letter (quote; "As shown in Table R2, the time constants of our device show superiority to those of previously reported neuro-inspired optoelectronic devices."). Besides the definition of the "superiority" depends on the purposes of those sensors.

Reply: Thanks very much for the reviewer's comments. We are sorry for the inaccurate statement regarding the time constants of the sensor array in the last response letter. It is routine to use time constants for evaluating basic characteristic of "pulse-interval dependent plasticity" in emerging neuro-inspired optical devices. As mentioned in main text of manuscript, the calculated time constants of NbS₂/MoS₂ optical sensor are equal to those of most previously reported devices, indicating its great potential in neuro-inspired applications for various purposes.

In addition, we acknowledge reviewer's claim that definition of the "superiority" depends on the purposes of those sensors. **The superiority of our proposed sensor lies in its expanded functionality, which integrates processing functions for both static and motion objects in a highly simplified optical sensor array without external processing circuits.** In contrast to previous neuro-inspired optical sensors that only reported on pre-processing static images, we have innovatively proposed an algorithm at the device level based on the kinetics of photocurrent in an NbS₂/MoS₂ optical device and successfully deployed it for motion scenarios.

(9)
enhancement, an artificial vision system including a NbS₂/MoS₂ sensor array and a
CNN were constructed, as shown in **Fig. 3d**. A 24000-sized image training set with 10

9. Although this reviewer would appreciate that the authors have added more results in the CNN training experiments, yet the authors have not answered to the suggestion in the second review (quote): "The authors need to show quantitative relationships between the contrast value and the recognition accuracy by using different combinations of laser light intensities for the target and background (and/or by using the image sets obtained at different time points of reading the preprocessed images out of the device). "

First of all, the data should be the real measured photocurrent maps, but not ones simulated based on the measured property of the device (Fig. S20). This could be done by, for example, dividing the noisy raw image (28 by 28 pixels) into 3 by 3 blocks (i.e. each block has 10 by 10 pixels or less), applying the blocked image to the device, wait for the decay for different time durations to readout the photo-current from the 10 by 10 sensors. And after repeating this procedure 9 times, those 9 of the blocked 10-by-10 photo-current maps are combined into a single "Enhanced" image.

Second, the real "Enhanced" image should be used as a test data (not a training data) to make the feed-forward inference with the CNN model that is pre-trained with a noise-less dataset.

Finally, based on the CNN inference experiment, relationships between the contrast value of the real "Enhanced" image and the recognition accuracy can be plotted. In this case, the definition of the contrast must be different from the one used in Fig. S18.

Then, it might be expected that both the contrast value and the recognition accuracy improve along with the relaxation time. Showing such a time-dependent improvement in the contrast enhancement of static images, and in the resulting recognition accuracy for those enhanced images, through a real (not simulated) experiment is considered to be essential to demonstrate the pre-processing functionality of the present device.

Reply: Thanks very much for the reviewer's comments. First of all, **we have to emphasize again that the Fig. 3c in the main text is exactly measured by mapping the real photocurrent of each pixel**, and the original **Supplementary Fig. 20**

(**Supplementary Fig. 21** in the updated version) is derived from the statistic values across a large number of real devices (**Fig. R3**). It strongly demonstrated the function of image contrast enhancement by our 10×10 NbS₂/MoS₂ sensor array.

In our study, the detection of the sensor array for static images is achieved with the assistance of the certain shaped photomask, in which the illumination with certain intensity would project the transparent area of the mask onto the sensor array. It is a routine strategy and widely used in state-of-the-art neuro-inspired optical sensors (*In-sensor image memorization and encoding via optical neurons for bio-stimulus domain reduction toward visual cognitive processing. Nature Communications, 2022, 13: 5223; A flexible ultrasensitive optoelectronic sensor array for neuromorphic vision systems. Nature Communications, 2021, 12: 1798; Retina-Inspired Color-Cognitive Learning via Chromatically Controllable Mixed Quantum Dot Synaptic Transistor Arrays. Advanced Materials, 2022, 34: 2108979*). Although the strategy reviewer mentioned, in which the 28×28 images are divided into several sub-blocks for measurement, is theoretically feasible. We have to note that up to 9 photomasks need to be fabricated for target letter and background letter, respectively, for each 28×28 image in EMNIST data set. In this process, the background noise with random light intensity cannot be totally overlapped onto the sensor array, which would cause tremendous challenges to experimentally evaluate the functions of filtering and contrast enhancement. Moreover, the development of corresponding neural networks is also immature. Indeed, the strategy in our study is widely utilized to accurately evaluate the performance of the sensor array for processing a large number of images with complex light intensity features, and we believe it is enough to demonstrate the feasibility of our device to “enhance” the contrast of target image (*Integration of synaptic phototransistors and quantum dot light-emitting diodes for visualization and recognition of UV patterns. Science Advances, 2022, 8: eabq3101; Photo-induced non-volatile VO₂ phase transition for neuromorphic ultraviolet sensors. Nature Communications, 2022, 13: 1729*).

Fig. R3. Statistics of photocurrent relaxation behavior of devices under different light intensities.

Secondly, as the reviewer mentioned, conventional approach employs a CNN model pre-trained with a noise-less dataset for feed-forward inference. However, we would like to clarify that our research mainly focuses on the real-world situations only with noisy unenhanced images, and there is no access to the completely clean and noise-free images. Using NbS₂/MoS₂ sensor array, these noisy images can be processed for better clarity. In other words, the "enhanced" dataset is expected to be the cleanest version of the dataset we can obtain in practical, real-world applications. Training our CNN model with these enhanced images allows us to understand its performance in conditions similar to real-life situations, making our findings more practical and applicable to real-world applications.

Thirdly, regarding the concern about the definition of contrast and the relationships between the contrast value and recognition accuracy, we have to emphasize the definition of contrast applied in this context as the ratio of the average intensity of informative letters to the average intensity of the noise. The noise component is extracted by taking the absolute difference between the noisy image and the expected clean informative letters image. The contrast can be expressed as:

$$\text{Contrast} = \frac{\text{Average}(\text{intensity of the informative letter})}{\text{Average}(\text{intensity of the noise})} \quad (1)$$

Fig. R4 and **R5** show the contrast enhancement performance of sensor array model and related recognition accuracy of CNN for 4 different combinations of laser light intensities for the target and background. The first two combinations involved unenhanced original input images with contrasts of 2.56 (Combination 1) and 2.29

(Combination 2), each enhanced with a relaxation time of 150s, depicted in **Fig. R4**. The next two combinations used unenhanced original input images with a contrast of 2.83. These images were enhanced with two different relaxation times: 0s (Combination 3) and 150s (Combination 4), as shown in **Fig. R5**.

Fig. R4. Contrast enhancement performances of sensor array and recognition accuracy of CNN for input images with different contrast of 2.29 and 2.56. a-b Comparison of the images before and after pre-processing. **c-d** Recognition accuracies of CNN with and without pre-processing.

Fig. R5. Contrast enhancement performances of sensor array and recognition accuracy of CNN for relaxation time of 0 s and 150 s. a-b Comparison of the images before and after pre-processing. **c-d** Recognition accuracies of CNN with and without pre-processing.

Fig. R6. Relationship between the contrast value and recognition accuracy of 4 different combinations of laser light intensities for the target and background and different device relaxation times. a Recognition accuracy of the CNNs with 800 training epochs under the 4 different combinations. **b** Relationship between the image contrast value and recognition accuracy.

Fig. R6 illustrates the relationship between the contrast value and recognition accuracy using 4 different combinations of laser light intensities for the target and background. It is clear that an increase in the original contrast of the input image, while maintaining the same NbS₂/MoS₂ optical sensor relaxation time, leads to an increased contrast in the output image, as well as enhancement of CNN recognition accuracy. Similarly, with a fixed original input image contrast, a longer NbS₂/MoS₂ optical sensor relaxation time would result into the increase of output image contrast and the accuracy of the CNN.

In summary, since the development of neuro-inspired vision sensors based on emerging materials and emerging in-sensor computing is still in its early stages, the scale of vision arrays with in-sensor computing capabilities are usually limited by the current fabrication technology and the uniformity of large-scale materials. They also lack standard ancillary components comparable to those found in commercial CMOS vision chips. These limitations make the prototyping of sensor arrays very challenging for capturing and processing massive real visual information. As described above, we

believe that the simulated relationship between contrast value and recognition accuracy, as obtained through this constructed sensor array model, could be an alternative method to evaluate the relationship in real-life scenarios. Although there is still much work to be done before neuro-inspired vision sensors based on emerging devices can work for real visual environments, the integration of multiple processing functions into the sensor cell is a breakthrough in the recent years, which offers a revolutionary strategy in data transfer reduction and system structure simplification.

Fig. R3 is added in supplementary information as **Supplementary Fig. 20**.

Fig. R4 and **R5** are updated as **Supplementary Figs. 24** and **25** in supplementary information, respectively.

Fig. R6 is added in supplementary information as **Supplementary Fig. 26**.

The related description of contrast in supplementary information has been revised as:

“The unenhanced dataset is created by overlapping a dark background letter and additional random background noise to informative letter images for closing to real scenarios. The light intensity ratio of bright informative letter and background letter is fixed at 1:0.5. The light intensity ratio of bright informative letter and background noise are randomly generated from range 1:0 to 1:0.6. The contrast of images in dataset is defined as the ratio of the average intensity of informative letters to the average intensity of the noise. The noise component is extracted by taking the absolute difference between the noisy image and the expected clean informative letters image. The contrast can be expressed as:

$$\text{Contrast} = \frac{\text{Average}(\text{intensity of the informative letter})}{\text{Average}(\text{intensity of the noise})} \quad (1)$$

”

(10)

enhancement, an artificial vision system including a NbS₂/MoS₂ sensor array and a

CNN were constructed, as shown in **Fig. 3d**. A 24000-sized image training set with 10

10. please insert "numerical model simulating a 28 × 28-pixel".

Reply: Thanks very much for the reviewer’s comments. The related sentence has been revised as below:

“To evaluate the recognition accuracy of the images before and after contrast enhancement, an artificial vision system including a numerical model simulating 28×28 -pixel $\text{NbS}_2/\text{MoS}_2$ sensor array and a CNN were constructed, as shown in Fig. 3d.”

enhancement, an artificial vision system including a $\text{NbS}_2/\text{MoS}_2$ sensor array and a
(11)
 CNN were constructed, as shown in Fig. 3d. A 24000-sized image training set with 10

11. In the Fig. 3d, change "Sensor Array" to "Sensor Array Model"

Reply: Thanks very much for the reviewer’s comments. As shown in Fig. R7, the term “Sensor Array” in Fig. 3d of manuscript has been revised with “Sensor Array Model”.

Fig. R7 Schematic diagram of constructed artificial vision system, in which the term “Sensor Array” has been revised with “Sensor Array Model”.

Fig. R7 is updated as part of **Fig. 3** in main text.

background letter, and background white noise, which were pre-processed by the sensor
(12)
 array and then inputted into the CNN for further high-level recognition

12. please insert "model"

Reply: Thanks very much for the reviewer’s comments. The related sentence has been revised as below:

“Each image with 28×28 pixels consisted of a bright informative letter, a dark background letter, and background white noise, which were pre-processed by the sensor array model and then inputted into the CNN for further high-level recognition (Supplementary Figs. 19-23).”

(13)

(Supplementary Figures S19-S22). Fig. 3e and **Supplementary Figures S23-S24**

show the performance of the artificial vision system for different combinations of input light intensity and relaxation time. The results of the output image dataset indicated that

13. Regarding the results shown in Figs. 3f, S22, S23c-d, S24c-d, the number of epoch should be increased beyond 500, since the accuracies (especially those for the "Unenhanced" ones in Figs. S23c-d and S24c-d) appear not to reach steady/plateau states with such a small epoch size.

Reply: Thanks very much for the reviewer's comments. We've increased the number of training epochs to 800 and updated the related **Fig. 3f** in main text and **Supplementary Figs. 22-24** in Supplementary Information, as shown in **Fig. R8**, **Fig. R9**, **Fig. R4**, **Fig. R5**, respectively.

Fig. R8. Recognition accuracies of the vision system with and without NbS₂/MoS₂ sensor array for image pre-processing.

Fig. R9. Recognition accuracies of two types neural network. **a** CNN and **b** ANN on two datasets.

Fig. R8 is updated as Fig. 3f in main text, and Fig. R9 is updated as Supplementary Fig. 23 in supplementary information.

array also showed stable trajectory registration performance for the moving light spots
(14)
 with different combinations of velocity and light intensity (Supplementary Figure
 S26).

14. The unit of light intensity is written incorrectly on the top of the left panels.

Reply: Thanks very much for the reviewer's comments. As shown in Fig. R10, the unit of light intensity in original Supplementary Fig. 26 has been revised.

Fig. R10 Performance of in-sensor trajectory registration for different velocity and light intensity of moving light spot. **a-b** NbS₂/MoS₂ sensor array for slower and faster trajectory registration. **c-d** NbS₂/MoS₂ sensor array for lower light intensity and higher light intensity trajectory registration (left panel shows real-time photocurrent of pixels located at trajectory, right panel shows current mapping of trajectory registration). The mistake of unit of light intensity has been corrected.

Fig. R10 is updated as **Supplementary Fig. 28** in supplementary information.

current data of two parts are collected and map-plotted into the same image. For motion
(15)
detection, a continuous laser (532 nm, 3.1 $\mu\text{W } \mu\text{m}^{-2}$) moving with a uniform speed along

15. The value should be updated accordingly to the new experiments (Figs. S26).

Reply: Thanks very much for the reviewer's comments. The related sentence has been revised as below:

"For motion detection, a 532 nm continuous laser with different light intensities (1.6, 3.1 or 4.1 $\mu\text{W } \mu\text{m}^{-2}$) moving with a uniform speed along the pre-defined trace was utilized to simulate the movement of the target."

(16)
The moving time between adjacent pixels was approximately 2.5 s, and the effective
illuminating time of each pixel was approximately 500 ms.

16. These values should be updated accordingly to the new experiments (Figs. S26).

Reply: Thanks very much for the reviewer's comments. The related sentence has been revised as below:

"The moving time between adjacent pixels and effective illuminating time of each pixel for motion velocity of 20, 40 and 80 $\mu\text{m } \text{s}^{-1}$ were approximately 5 s/1 s, 2.5s/0.5s and 1.25 s/0.25 s, respectively."

Data availability

(17)

All data are available in the main text or the supplementary materials.

17. The python codes/programs should be made available on a Github, since the machine learning-based evaluation is extended by the revisions, but yet that the details of the CNN and ANN (network structures, activation function, optimization, batch size,

dropouts, padding, pooling, loss function and etc.) are not described at all in the manuscript as well as the supplementary materials.

Reply: Thanks very much for the reviewer's comments. The network structures of CNN and ANN are shown in **Supplementary Fig. 22** in revised supplementary information. ANN uses ReLU as the activation function for all fully connected layers except the last layer, which utilizes Softmax function. It is optimized using the Adam function, configured with beta1 equal to 0.90, beta2 equal to 0.999, and a learning rate of 0.001. The ANN is trained with a batch size of 40 and uses the categorical crossentropy loss function to calculate errors. CNN also employs the same ReLU activation function for all convolution and fully connected layers except the last layer, which adopts Softmax function. CNN is optimized with the identical Adam function parameters as ANN (beta1=0.9, beta2=0.999, learning rate=0.001). The training batch size is also 40. The CNN's convolutional layers are configured with "same" padding to ensure consistent spatial dimensions between input and output feature maps. The categorical crossentropy loss function is also used in the CNN for error calculation.

Since our code is not a general Python code but a newly developed code, particularly for our emerging device array, we prefer not to open it for publicity as the open-access code on Github at the current stage, which involves the copyright issue. However, we have stated the code availability in the manuscript, and we would be ready to send it to the readers under reasonable requests.

The related description in Supplementary Information has been revised as:

"ANN uses ReLU as the activation function for all fully connected layers except the last layer, which utilizes Softmax function. It is optimized using the Adam function, configured with beta1 equal to 0.9, beta2 equal to 0.999, and a learning rate of 0.001. The ANN is trained with a batch size of 40 and uses the categorical crossentropy loss function to calculate errors. CNN also employs the same ReLU activation function for all convolution and fully connected layers except the last layer, which adopts Softmax function. CNN is optimized with the identical Adam function parameters as ANN (beta1=0.9, beta2=0.999, learning rate=0.001). The training batch size is also 40. The CNN's convolutional layers are configured with "same" padding to ensure consistent

spatial dimensions between input and output feature maps. The categorical crossentropy loss function is also used in the CNN for error calculation.”